# Vascular FLRT2 regulates venous-mediated angiogenic expansion and CNS barriergenesis

C. Llaó-Cid [1,5], B. Peguera[1,5], P. Kobialka[1], L. Decker [1], J. Vogenstahl[1,2], N. Alivodej[1,2], S. Srivastava[1], J. Jin[1], B. C. Kirchmaier[1], C. Milla[1], H. Schlierbach[3], A. Schänzer[3], T. Acker [3], M. Segarra [1,4] ✉ & A. Acker-Palmer [1,2,4] ✉

Veins have emerged as the origin of all other endothelial cell subtypes needed to expand vascular networks during developmental and pathological neoangiogenesis. Here, we uncover the role of the angioneurin Fibronectin Leucine Rich Transmembrane protein (FLRT) 2 in central nervous system (CNS) vascular development in the mouse. Early postnatal FLRT2 deletion reveals specific defects in retinal veins, impacting endothelial cell proliferation, sprouting and polarity that result in reduced tip cells at the vascular front. FLRT2 interacts with VE-cadherin and together with the endocytic adaptor protein Numb contribute to the modulation of adherens junction morphology in both retina and cerebral cortex in vivo. Utilizing expansion microscopy, we visualize the altered dynamic distribution of VE-cadherin in tissue of FLRT2 endothelial mutants. Additionally, FLRT2 in cortical vessels regulates the crosstalk between adherens and tight junctions, influencing blood-brain barrier development. Our findings position FLRT2 as a vein-specific regulator of CNS vascular development.

Venous endothelial cells (ECs) are the primary cellular source of angiogenesis, the expansion of a preexisting vascular network necessary for the vascularization of all tissues. Angiogenic sprouts from the venous plexus drive the formation of the capillary and arterial network as shown in the development of several tissues, such as the heart[1], brain[2] and retina[3,4]. Rapidly proliferating venous ECs migrate in the opposite direction of the bloodstream, undergo differentiation into various endothelial subtypes (capillaries, tip cells and arteries)[5], and act as the main supplier of ECs for the expansion of the vascular network (reviewed in ref. 6). Notably, quiescent venous endothelium gets reactivated upon injury and acquires a proliferative and migratory behavior[3], suggesting the presence of venous-specific molecular control mechanisms. Despite significant advancements in comprehending the role of venous endothelium in various developmental and pathological contexts, our understanding of the molecular mechanisms responsible for angiogenic expansion from the venous endothelium is still unexplored.

Fibronectin leucine rich transmembrane (FLRT) proteins, a versatile family of molecules comprising three members (FLRT1-3), have been recently shown to have dual roles during the development and function of neuronal and vascular systems. Since their identification[7], multiple functions of FLRTs in early embryonic and neuronal development have been characterized[8–14]. For instance, FLRTs regulate cell repulsion and attraction during neuronal migration in the cerebral cortex and can interact with a myriad of other signaling partners[15,16]. We have previously identified FLRT signaling complexes as novel angioneurins[17], by showing that FLRT3 is mostly expressed in the neuronal compartment of the retina and regulates developmental

[1]Buchmann Institute for Molecular Life Sciences (BMLS), Institute of Cell Biology and Neuroscience, Goethe University Frankfurt, Max-von-Laue-Str. 15, D-60438 Frankfurt am Main, Germany. [2]Max Planck Institute for Brain Research, Max-von-Laue-Str. 4, 60438 Frankfurt am Main, Germany. [3]Institute of Neuropathology, Justus Liebig University Giessen, D-35392 Giessen, Germany. [4]Cardio-Pulmonary Institute (CPI), Max-von-Laue-Str. 15, D-60438 Frankfurt am Main, Germany. [5]These authors contributed equally: C. Llaó-Cid, B. Peguera. ✉e-mail: Segarra@bio.uni-frankfurt.de; Acker-Palmer@bio.uni-frankfurt.de

retina vascularization by exerting repulsive responses in ECs expressing Uncoordinated-5 receptor B (Unc5B). Few studies have pointed out the role of FLRT2 in ECs; early expression of FLRT2 in placental vessels contributes to placental labyrinth formation[18], and tumor-specific inter-endothelial adhesion mediated by FLRT2 controls pathological angiogenesis and cancer aggressiveness[19]. However, FLRT2 is also expressed in vessels of the central nervous system (CNS) under physiological conditions, suggesting a potential role in CNS vascularization that has yet to be explored.

Here, we show that FLRT2 is expressed in the CNS vasculature, specifically in veins, and deletion of FLRT2 early postnatal leads to specific proliferation and sprouting defects in the venous plexus within the retina. Such defects lead to an impaired EC polarity and a reduction in tip cells at the vascular front. Mechanistically, we demonstrate that FLRT2 interacts with VE-cadherin and controls its recycling at EC junctions. FLRT2 forms a complex with the endocytic adaptor protein Numb, allowing it to modulate VE-cadherin dynamics. Deletion of FLRT2 in ECs disrupts the morphology of adherens junctions in vivo, both in the retina and the cerebral cortex. Employing expansion microscopy, we directly show in tissue the subcellular localization of VE-cadherin at tip cell endothelial sprouts versus ECs within the mother vessel and how the distribution is affected in FLRT2 endothelial-specific mutants. Moreover, in cortical vessels, FLRT2 regulates the crosstalk between adherens and tight junctions important for the development of the blood-brain barrier (BBB). In summary, FLRT2 emerges as a regulator of CNS vascular development, influencing venous-mediated angiogenesis, ECs junctional organization, VE-cadherin dynamics and BBB function.

## Results

### FLRT2 is expressed in vessels and controls postnatal CNS vascular development

We aimed to study the expression of FLRT2 at postnatal ages, when the CNS vasculature undergoes active growth and remodeling crucial for proper organ development. Fluorescence in situ hybridization (FISH) was employed to investigate the expression patterns of FLRT2 in different regions of the CNS, such as the murine postnatal retina and cerebral cortex. Besides the well-known expression of FLRT2 in neurons[8,20-22], we demonstrated that FLRT2 is expressed in the CNS vasculature during the postnatal period, as illustrated in Fig. 1a. Furthermore, immunostaining analysis conducted on the cerebral cortex highlighted a pronounced presence of FLRT2 in specific large vessels (indicated by arrows in Fig. 1b). The observed expression pattern of FLRT2 in the CNS vasculature suggests a potentially significant role of this protein in the process of CNS vascularization.

To investigate the specific role of FLRT2 in ECs, a vascular-specific FLRT2 loss-of-function mouse model (referred to as $Flrt2^{i\Delta EC}$) was generated. This was achieved by crossing tamoxifen-inducible EC-specific Cre-deleter Cdh5(PAC)-CreERT2 mice[23] with $Flrt2^{lox/lox}$ mice[8]. The resulting mice were subjected to administration of 4-OH-tamoxifen (Tmx) from postnatal day (P) 1 to P3 to suppress the expression of FLRT2 exclusively in ECs (Supplementary Fig. 1a, b). The effectiveness of Tmx treatment was confirmed by evaluating Cre-mediated recombination, which resulted in tdTomato signal in approximately 80% of the vasculature in the retina of Cdh5(PAC)-CreERT2 mice crossed with the Rosa26-Stop$^{lox/lox}$-tdTomato reporter line[24] (Supplementary Fig. 1c, d). Similarly, the vasculature of the cerebral cortex also showed a highly efficient Cre-recombination (96,4% of tdTomato signal in the vasculature) (Supplementary Fig. 1e, f). To verify the successful reduction of FLRT2 protein expression in ECs, western blot analysis was performed in primary mouse lung ECs, demonstrating an efficient reduction in ECs isolated from mutant ($Flrt2^{i\Delta EC}$) mice compared to control littermates ($Flrt2^{lox/lox}$) (Supplementary Fig. 1g). To demonstrate the reduction of FLRT2 in the CNS vasculature, we isolated primary mouse brain ECs at P5 from $Flrt2^{i\Delta EC}$

and control littermates after Tmx treatment from P1-P3. RT-qPCR showed that $Flrt2$ gene expression was significantly reduced, and this reduction did not result in a compensatory upregulation of $Flrt3$, another FLRT family member with reported endothelial expression[17,25] (Supplementary Fig. 1h). The reduction of FLRT2 protein levels was confirmed by immunodetection of FLRT2 protein in primary mouse brain ECs seeded on coverslips (Supplementary Fig. 1i, j). Additionally, the loss of FLRT2 expression in the vasculature did not cause growth retardation in the whole body of either female or male mice, nor did it result in changes in brain size in the postnatal animals (Supplementary Fig. 1k–n).

We next analyzed the effects of endothelial FLRT2 deletion in CNS postnatal angiogenesis. We isolated mouse retinas at P7-P8 and utilized isolectinB4 (IB4) staining to visualize the blood vessels. The radial length of the retinal vascularization was significantly reduced in $Flrt2^{i\Delta EC}$ mice compared to control littermates, such that the vascular plexus covered a smaller extent of the retinal surface at P7-P8 (Fig. 1c, d). Additionally, the total vessel length and the number of vascular branch points were also significantly reduced in $Flrt2^{i\Delta EC}$ retinas (Fig. 1c, e, f). Furthermore, the capillary network between arteries and veins exhibited reduced density in $Flrt2^{i\Delta EC}$ animals compared to their littermate controls (Fig. 1g, h). However, there were no differences in vessel regression, as determined by the presence of Collagen IV empty sleeves (Supplementary Fig. 2a, b). This indicates that the vascular defects triggered by the absence of FLRT2 are due to defective vessel formation rather than increased vessel loss. Overall, these findings provide compelling evidence that FLRT2 plays a crucial role in postnatal angiogenesis within the CNS.

To determine whether the effects observed were specific to retinal vasculature or applicable to the entire CNS vasculature, we extended our investigation to the cerebral cortex vasculature. We examined cortical sections from P7-P8 $Flrt2^{i\Delta EC}$ mice and control littermates that underwent Tmx treatment from P1 to P3. To visualize the blood vessels, we stained the sections with an anti-Glut1 antibody. Consistent with our previous findings in the retina, we found that $Flrt2^{i\Delta EC}$ animals exhibited a reduction in blood vessels covering the cortical area (Fig. 1i, j). Additionally, the total length of vessels (Fig. 1i, k) and the number of vessel branch points (Fig. 1i, l) were significantly decreased in $Flrt2^{i\Delta EC}$ mice compared to the control group. These results indicate that FLRT2 is essential for postnatal angiogenesis not only in the retina but also throughout the CNS, including the cerebral cortex.

### FLRT2 deletion causes vein-specific defects in the retina and reduced sprouting angiogenesis at the vascular front

To gain further insights into the underlying cause of the capillary density defect observed in $Flrt2^{i\Delta EC}$ mice, we conducted a quantitative analysis focusing on the branch points along the arteries and veins in the retina. Our analysis revealed an intriguing pattern: while the number of branches from arteries remained unaffected by FLRT2 deletion, there was a significant reduction in the number of branches originating from veins (Fig. 2a, b). These results strongly indicate that the absence of vascular FLRT2 specifically impacts the sprouting of branches from veins, which ultimately contributes to the observed capillary density defect. Postnatal endothelial FLRT2 deletion did not have an impact on the total number of arteries and veins present in the retina (Supplementary Fig. 2c, d).

Single-cell RNA sequencing data from brain samples[26,27] indicates a prevalent expression of $Flrt2$ in the venous compartment. To prove the presence of FLRT2 protein in a vein-specific manner, we conducted experiments utilizing expansion microscopy, a recently developed technique that enhances imaging resolution by physically enlarging the sample size[28]. Additionally, we leveraged the morphological distinction between arteries and veins in retinal vasculature for our analysis[29]. Retinal tissue was immunolabeled for FLRT2 and Collagen IV

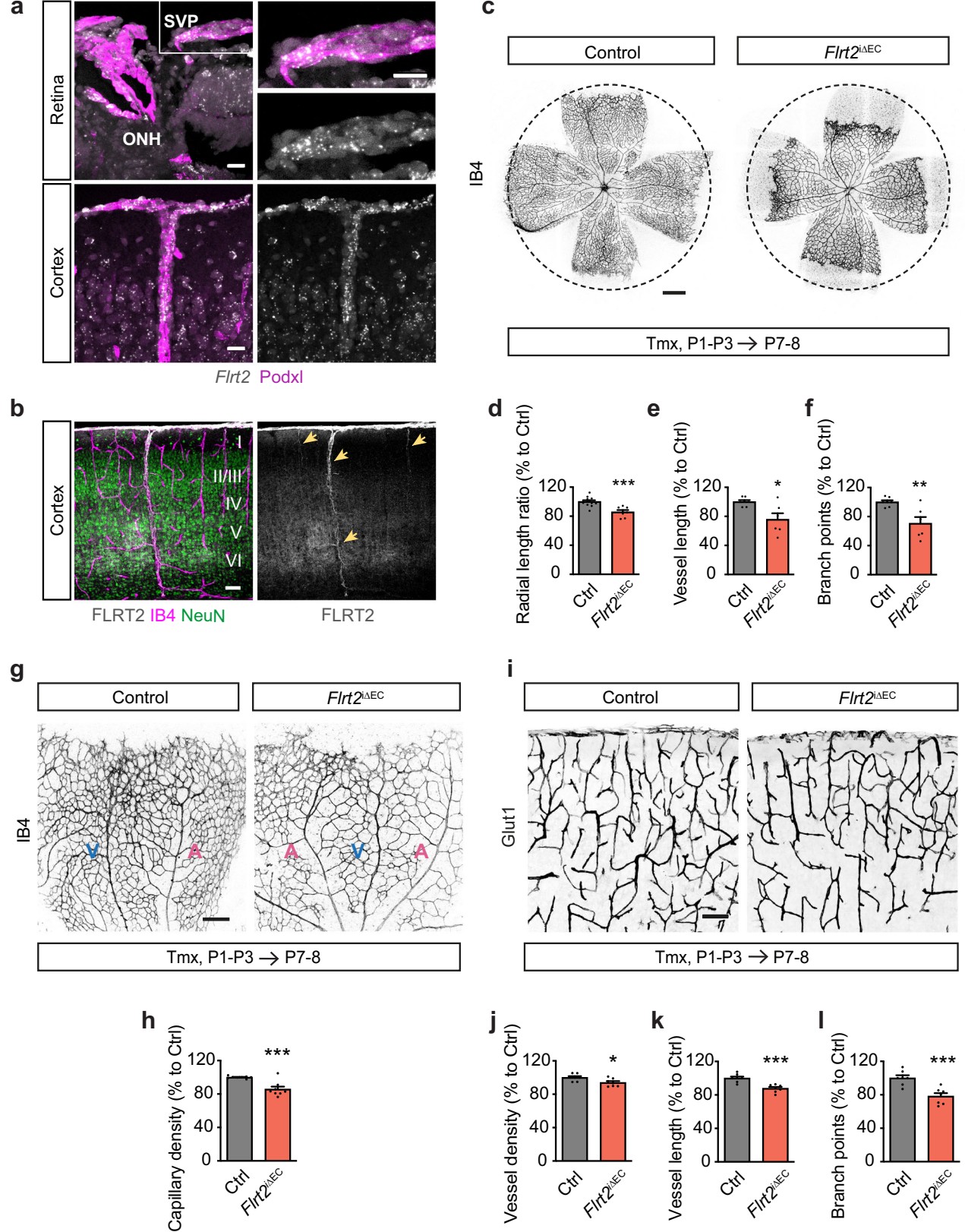

to visualize the vasculature. Subsequently, the tissue was embedded in a swellable polymer gel. After tissue digestion, the staining information was solely retained within the gel, which could be expanded by immersion in water, resulting in a four-fold isotropic increase in sample size. The expanded retinas were then examined, revealing that the FLRT2 signal was minimal in arteries but prominently localized to the

EC membrane of veins, indicating that FLRT2 expression exhibited preferential venous zonation (Fig. 2c). To corroborate whether FLRT2 venous zonation is also prevalent in the vasculature of the cerebral cortex, we performed fluorescent in situ hybridization combining *Flrt2* mRNA detection with in situ probes for arterial or venous markers (*Gkn3* and *Slc38a5*, respectively) based on published single-cell RNA

**Fig. 1 | CNS vascularization is compromised after FLRT2 deletion in ECs. a** *Flrt2* mRNA detection by fluorescence in situ hybridization (FISH) in P8 wild-type mouse retina in the optic nerve head (ONH) area, where the central artery and vein enter the retina, and in the superficial vascular plexus (SVP) (upper panel); and in P8 wild-type cerebral cortex (upper cortical layers and pial vasculature) (lower panel). Blood vessels were detected by immunostaining with podocalyxin (Podxl) (general vessel marker). **b** Coronal section of the cerebral cortex from P6 wild-type mouse stained for FLRT2 and NeuN as neuronal marker (neuronal layers I-VI annotated). Blood vessels were visualized with isolectin-B4 (IB4) staining. Arrows show FLRT2 positive blood vessels (right). **c** Flat-mounted P7-P8 retinas from control and *Flrt2*[iΔEC] littermate mice injected with 4-hydroxytamoxifen (Tmx) from P1 to P3 and stained with IB4. Quantification of radial vascular length

ratio (**d**), total retinal vessel length (**e**), and total number of branch points (**f**) per retina. *n* = 12 control and 7 mutant (**d**), 7 control and 6 mutant (**e**, **f**) animals. Two-tailed unpaired *t*-test, *p* = 0.0004 (**d**), 0.013 (**e**), 0.005 (**f**). **g** Representative images of P7-P8 control and *Flrt2*[iΔEC] flat-mounted retinas stained for IB4. Veins (V) and arteries (A) are indicated. **h** Quantification of capillary network density between veins and arteries. *n* = 8 animals per genotype. Two-tailed unpaired *t*-test, *p* = 0.0005. **i** Glut1 staining of the vasculature in control and *Flrt2*[iΔEC] brain cortices from P7-P8 mice after Tmx administration from P1 to P3. Quantification of vessel density (**j**), vessel length (**k**) and number of branch points (**l**). *n* = 7 animals per genotype. Two-tailed unpaired *t*-test, *p* = 0.028 (**j**), 0.001 (**k**), 0.001 (**l**). Scale bars: 20 μm (**a**), 50 μm (**b**), 500 μm (**c**), 200 μm (**g**), 100 μm (**i**). Data are shown as mean ± SEM. *$P < 0.05$, **$P < 0.01$, ***$P < 0.001$.

sequencing data[27]. Podocalyxin immunostaining was used as a general vascular marker. Consistent with our observations in retinal vessels, we detected *Flrt2* mRNA in veins, whereas *Flrt2* signal in arteries was undetectable (Supplementary Fig. 2e).

Veins have been identified as the primary source of ECs for angiogenesis, and recent studies have demonstrated that venous ECs give rise to the endothelial tip cells, which are located at the leading edge of retinal vasculature and subsequently migrate into the arterial plexus[2,3,5]. Considering this, we examined the vasculature at the retinal front and found that the number of vascular sprouts, which are characterized by protruding cellular structures along the vascular front, was significantly reduced in *Flrt2*[iΔEC] animals (Fig. 2d, e). The fewer tip cells in the *Flrt2*[iΔEC] animals did not show defects in the number of filopodia extensions per cell (Supplementary Fig. 2f, g). Vascular sprouts are formed by leading endothelial tip cells followed by stalk cells that proliferate and elongate the developing vascular tubes[30]. Consistent with the decreased number of vascular sprouts, we also observed a decreased proportion of tip cells relative to stalk cells at the vascular front of *Flrt2*[iΔEC] mice (Fig. 2f, g). In addition, the number of proliferating ECs labeled with EdU at the angiogenic front was significantly reduced in the retinal vasculature of *Flrt2*[iΔEC] animals (Supplementary Fig. 2h, i). The deficient angiogenic sprouting observed in *Flrt2*[iΔEC] animals was not limited to the retinal plexus but was also observed in other regions of the CNS vasculature. We found a reduced number of tip cells in the cerebral cortex at P7-8, further supporting the generalized impairment of sprouting angiogenesis in *Flrt2*[iΔEC] mice (Fig. 2h, i).

To determine the underlying cause of the reduced vascularization in *Flrt2*[iΔEC] mice, we examined the development of the vasculature at an earlier time point (P4-P5). Interestingly, even though the deletion of *Flrt2* was efficient at P5 (Supplementary Fig. 1h), we did not observe any differences in previously analyzed vascular growth parameters between mutant and control samples in the retina (Supplementary Fig. 3a–c) or the cerebral cortex (Supplementary Fig. 3d–g). However, we found that the number of branches from the retinal veins (Supplementary Fig. 3h, i) and the sprouts at the vascular front (Supplementary Fig. 3j, k) were already significantly reduced in *Flrt2*[iΔEC] mouse retinas at this early time point. Similarly, deficient sprouting was evident in the cortical vessels (Supplementary Fig. 3l, m), preceding the manifestation of other vascular defects. These findings strongly suggest that the absence of FLRT2 impairs the early stages of angiogenic sprouting and such impairment in sprouting appears to be the triggering event that ultimately leads to the subsequent vascular defects observed at later time points.

Next, we examined the cellular-level effects derived from FLRT2 absence at P5, potentially causing a loss of vascular density and length at P7-P8. First, we quantified ECs expressing the apoptotic marker cleaved caspase-3 in the retina and cerebellar cortex. Both control and mutant vessels showed similarly low levels of apoptosis at P5 (Supplementary Fig. 4a–d), aligning with the absence of vessel regression defects at P7 (Supplementary Fig. 2a, b). Next, we investigated cell cycle disruption by measuring the expression of the cell cycle inhibitor

p21 (*Cdkna1*), which induces cell cycle arrest[31]. While p21 was primarily detected in tip cells at vascular front of control retinas as expected[32], *Flrt2*[iΔEC] mice showed a significant increase in p21 expression, also localized in stalk cells (Supplementary Fig. 4e, f), indicating that proliferative endothelium has entered cell cycle arrest. Supporting these results, the cerebral cortex vasculature also showed a significant increase in ECs expressing p21 (Supplementary Fig. 4g, h). These findings are consistent with decreased cell proliferation at P7-P8, indicated by fewer cells entering the S phase, as shown by the thymidine analog EdU assay (Supplementary Fig. 2h, i).

### FLRT2 controls sprouting angiogenesis by interacting with VE-cadherin to regulate its recycling at EC junctions

VE-cadherin is a transmembrane protein that plays a critical role in establishing and maintaining adherens junctions between ECs, which are essential for the integrity of blood vessels[33–36]. The regulation of VE-cadherin levels and turnover at the cell surface is crucial for the formation of endothelial sprouts during angiogenesis. Lateral diffusion, clustering at cell-cell junctions, and internalization through various endocytic pathways have been identified as mechanisms governing VE-cadherin dynamics[33–35,37,38]. To investigate a potential interaction between FLRT2 and VE-cadherin in ECs we employed proximity ligation assays (PLA). PLA is a technique that enables the detection of protein-protein interactions in situ by visualizing fluorescent puncta that indicate close proximity between two proteins. PLA assay in human umbilical vein ECs (HUVECs) revealed an interaction between FLRT2 and VE-cadherin (Fig. 3a, b). To confirm such interaction, we conducted co-immunoprecipitation experiments. Lysates from cultured HUVECs were incubated with either a VE-cadherin antibody or an IgG control, and the resulting immunoprecipitates were subjected to western blot analysis. Detection with FLRT2-specific antibodies confirmed the interaction between FLRT2 and VE-cadherin in ECs (Supplementary Fig. 5a). To determine if a similar interaction occurs in vivo, we performed immunoprecipitation in total brain lysates using a VE-cadherin antibody as bait to selectively isolate FLRT2 coupled to VE-cadherin in the CNS vasculature. Our results revealed that VE-cadherin binds to FLRT2 in the murine vasculature (Fig. 3c). Importantly, this interaction was reproduced using PLA in primary mouse brain ECs from control animals and significantly diminished in primary mouse brain ECs isolated from *Flrt2*[iΔEC] mice (Supplementary Fig. 5b, c). Based on these findings, we postulate that FLRT2 functions as a partner of VE-cadherin to control sprouting angiogenesis.

Following our observation of the association between FLRT2 and VE-cadherin, we aimed to understand the functional significance of this interaction and its potential role in regulating the dynamic turnover of VE-cadherin at cell junctions. The expression of VE-cadherin at the cell surface is tightly regulated, with rapid endocytosis and subsequent degradation or recycling back to the cell surface[39]. This fast recycling provides a dynamic intercellular adhesion, necessary for EC rearrangement during sprouting angiogenesis[33,34,40]. To investigate whether FLRT2 is involved in VE-cadherin endocytosis and recycling, we utilized small interfering RNA (siRNA) to knock-down FLRT2

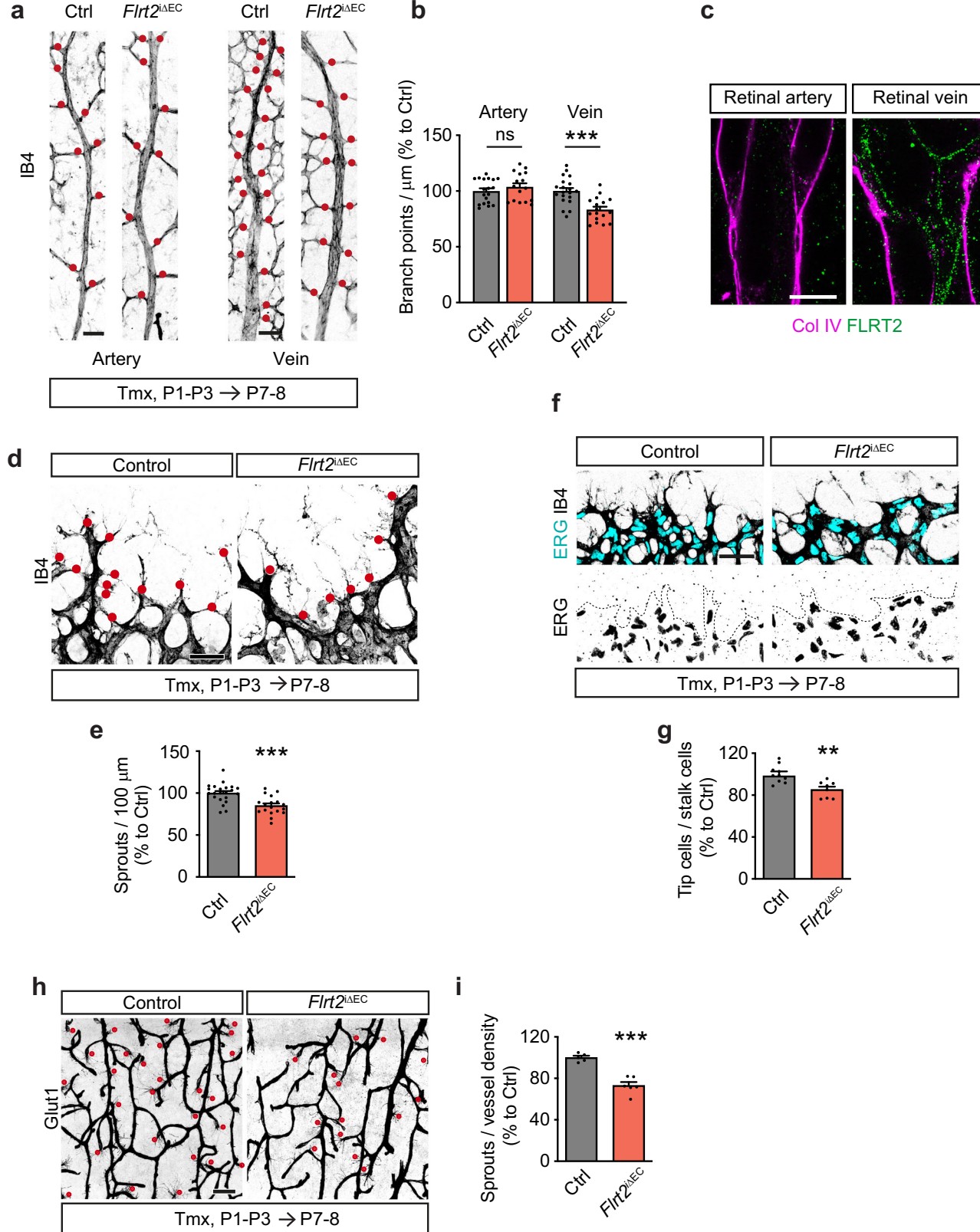

expression in HUVECs. Following a successful reduction of FLRT2 protein levels (Fig. 3d, e), we performed an antibody-feeding assay in HUVECs to track the internalization and vesicular trafficking of VE-cadherin. HUVECs were incubated with an antibody specific to the extracellular domain of VE-cadherin to next identify the internalized protein. To visualize VE-cadherin internalization in cells before its degradation, we added chloroquine, an inhibitor of lysosomal degradation[39,41]. This step allowed us to trap VE-cadherin that is being destined to lysosomal degradation. By performing an acidic wash before fixation, only the labeling of internalized protein remained. Afterwards, the cells were immunolabeled with a different species-specific VE-cadherin antibody to detect the total amount of VE-cadherin in the cells. The staining for endocytosed VE-cadherin revealed a significant increase in VE-cadherin internalization in ECs

**Fig. 2 | Endothelial FLRT2 regulates specifically venous sprouting and CNS angiogenic expansion. a** Representative images of main arteries and veins (identified by their morphology) from control and *Flrt2*<sup>iΔEC</sup> P7-P8 retinas stained with IB4. Red dots indicate branch points from the corresponding mother vessel. **b** Quantification of the number of branch points in arteries (left) and veins (right) per vessel length. *n* = 19 control and 16 mutant animals (arteries); 20 control and 17 mutant animals (veins). Two-tailed Mann-Whitney test, *p* = 0.333 (arteries); Two-tailed unpaired *t*-test, *p* = 0.0001 (veins). **c** Expanded P7 retina stained for FLRT2 and collagen IV (Col IV) showing FLRT2 expression in retinal vessels. Note the expression of FLRT2 at the EC membrane in the vein but its absence in the artery. **d** Representative images of retinal vascular front from control and *Flrt2*<sup>iΔEC</sup> mice stained with IB4. Red dots indicate cellular protrusions identified as angiogenic sprouts. **e** Quantification of number of sprouts per 100 μm of retinal vascular front. *n* = 22 control and 18 mutant mice. Two-tailed unpaired *t*-test, *p* = 0.0002. **f** Vascular fronts from control and *Flrt2*<sup>iΔEC</sup> P7-P8 retinas showing blood vessels labelled with IB4 and EC nuclei stained for ERG. **g** Quantification of the number of tip cells per stalk cells at the vascular front. *n* = 10 control and 8 mutant mice. Two-tailed unpaired *t*-test, *p* = 0.002. **h** Glut1 staining visualizing vessel sprouts in P7-P8 control and *Flrt2*<sup>iΔEC</sup> cerebral cortices. **i** Quantification of the number of sprouts per vessel density in the cerebral cortex. *n* = 5 control and 6 mutant mice. Two-tailed unpaired *t*-test, *p* < 0.0001. Scale bars: 50 μm (**a**, **f**), 90 μm (**c**), 40 μm (**d**), 100 μm (**h**). Data are shown as mean ± SEM. \*\*P < 0.01, \*\*\*P < 0.001, ns = not significant.

lacking FLRT2 (Fig. 3f, g). Interestingly, we detected an increased amount of VE-cadherin accumulated at the EC junctions in the absence of FLRT2 (Fig. 3f, h and Supplementary Fig. 5d, e) that was accompanied by a significant upregulation of the VE-cadherin gene (*Cdh5*) expression analyzed by quantitative RT-PCR (Supplementary Fig. 5f). These observations align with previously reported compensatory mechanisms, which involve increased VE-cadherin synthesis to uphold EC adhesion under conditions of accelerated protein turnover[42].

Studies have shown that upon endocytosis, a 95 kDa VE-cadherin fragment is generated by Calpain-1-mediated proteolytic cleavage at the C-terminal tail. This cleavage directs the cleaved VE-cadherin fragment towards degradation in the lysosomal compartment rather than recycling to the cell membrane[43]. Based on this knowledge, we hypothesized that FLRT2 may regulate the degradation/recycling dynamics of VE-cadherin by affecting its cleavage. Supporting this hypothesis, western blot analysis showed an increase in the cleaved 95 kDa VE-cadherin fragment in the absence of FLRT2 (Fig. 3i, j). Calpain-1 and Calpain-2 are the two Calpain isoforms described in ECs[44]. In vitro studies have shown that Calpain-1 can generate VE-cadherin cleaved fragment[43], and consistently, we detected an elevated Calpain-1 signal in *Flrt2*-silenced ECs (Supplementary Fig. 5g, h). To determine if this mechanism occurs in vivo, we analyzed Calpain expression in tissue. Focusing on Calpain-2, the most abundant in mouse brain ECs (see transcriptomic data[27,45]), we found significantly increased protein levels in primary mouse brain ECs isolated from *Flrt2*<sup>iΔEC</sup> mice compared to control littermates (Fig. 3k, l). Further analysis of tissue samples revealed enhanced Calpain-2 expression in FLRT2-depleted vasculature (Fig. 3m, n), corroborating that FLRT2 regulates VE-cadherin degradation via Calpain-mediated cleavage. These results suggest that FLRT2 plays a crucial role in regulating the internalization/recycling dynamics of VE-cadherin by affecting its cleavage by Calpains and its consequent degradation. Note that the increased expression and degradation of VE-cadherin in FLRT2 knockdown cells rendered unaffected levels of total VE-cadherin protein (Fig. 3i, j). These results provide further support for the compensatory mechanisms that occur in *Flrt2*-silenced ECs, involving the upregulation of VE-cadherin de novo synthesis and increased VE-cadherin signal along the cell junctions to maintain EC adhesion.

## FLRT2 controls VE-cadherin dynamics through the endocytic adaptor protein Numb

To unravel the molecular mechanisms underlying the regulation of VE-cadherin by FLRT2, our focus turned to Numb, an intracellular adaptor protein pivotal for the endocytic recycling of transmembrane proteins, including cadherins[46–49]. We therefore investigated the potential interaction between FLRT2 and Numb in ECs. Proximity ligation assays unveiled interactions between Numb, FLRT2, and VE-cadherin (Fig. 4a), indicating the formation of a functional complex. Subcellular localization analysis of FLRT2/Numb complexes, using expansion microscopy in sparse ECs in culture to enhance visualization of cell-cell interfaces, revealed the accumulation of Numb/FLRT2 complexes at junctions between ECs and along EC filopodia extensions (Fig. 4b). Furthermore, ECs deficient in FLRT2 exhibited a decrease in *Numb*

mRNA levels, as confirmed by quantitative RT-PCR analysis (Fig. 4c), coupled with a reduction in Numb protein levels compared to control-silenced cells (Fig. 4d, e). To verify whether FLRT2 also regulates Numb expression in vivo, we isolated primary mouse brain ECs from control and *Flrt2*<sup>iΔEC</sup> mice and performed RT-qPCR. Importantly, the expression level of *Numb* was significantly reduced upon FLRT2 abrogation in the brain vasculature (Fig. 4f). These findings provide additional support for the functional interaction between FLRT2 and Numb, underscoring their collaborative role in sprouting angiogenesis.

Numb has also been shown to play a pivotal role in front-rear polarity[50] which is essential for sprouting angiogenesis[51]. Importantly, it has been described that EC polarity at the front is a response to angiogenic stimuli regulating sprouting angiogenesis independently of the blood flow[52]. Since deletion of FLRT2 in ECs led to a reduction of Numb protein, we postulated that vascular FLRT2 mutants will display defects in EC polarity at the sprouting front. EC polarity is characterized by the position of the Golgi apparatus in front of the nucleus pointing toward the directional angiogenic cues. To assess whether front-rear cell polarization was impaired in *Flrt2*-deficient EC, we used siRNA to knock-down FLRT2 expression in HUVECs and performed a scratch wound assay upon their confluent growth in culture. We then measured the proportion of properly polarized cells (considering those in which the Golgi orientation, detected with GM130 antibody, was positioned within ± 60 degrees perpendicular to the scratch[53]) at the three first front layers of the wound (Fig. 4g). These experiments revealed that *Flrt2*-knockdown ECs failed to properly polarize at the migrating front (Fig. 4h, i). These results were supported in vivo by the loss of EC polarity at the sprouting front in endothelial-specific FLRT2 deficient mutants. The proportion of cells polarized within the first 200 μm of the angiogenic front, which depends on chemoattractant cues rather than blood-flow-induced shear-stress[52], was significantly reduced in *Flrt2*-deficient retinas (Fig. 4j, k). This indicates that endothelial FLRT2 regulates front-rear cell polarization necessary for vascular sprouting independent from mechanical forces.

These results suggest that FLRT2 modulates the recycling of VE-cadherin through Numb and this association is important for EC polarity at the vascular front and, consequently, for sprouting angiogenesis.

## Deletion of FLRT2 affects the morphology of adherens junctions in vivo

To elucidate whether the altered degradation/recycling levels of VE-cadherin in FLRT2 deficient animals and the defective cell polarity have functional consequences for the organization of adherens junctions in the CNS vasculature in vivo, we analyzed the pattern of VE-cadherin distribution in the vasculature of control and *Flrt2*<sup>iΔEC</sup> mice at P5. We focused on the sprouting front of the retina and the remodeling capillaries in the cerebral cortex and applied a categorization method previously described[33,34]. In this analysis, VE-cadherin staining images were segmented into regions of interest (ROIs), and each ROI was blindly assigned to one of three categories based on the VE-cadherin distribution pattern: smooth continuous pattern corresponding to inactive endothelium, irregular pattern corresponding to medium

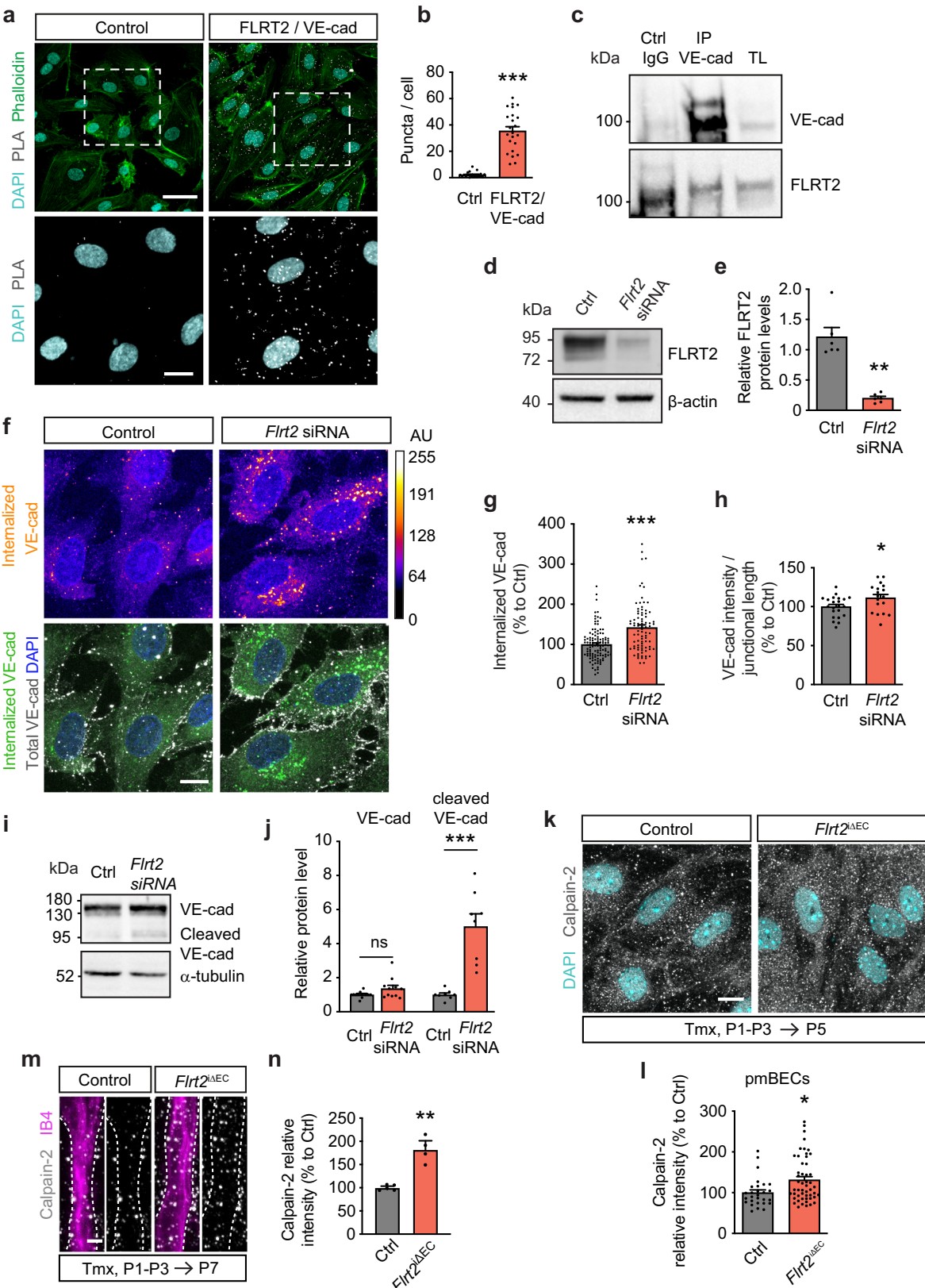

activity, and rough discontinuous pattern corresponding to active endothelium. Our analysis revealed a significant increase in inactive junctions in the vessels of *Flrt2*[iΔEC] animals, both in the retina (Fig. 5a, b) and in the capillaries of the cerebral cortex (Fig. 5c, d). Additionally, the medium activity was significantly reduced in the cerebral cortex vessels (Fig. 5c, d). These results indicate that the reduced sprouting

activity observed in FLRT2 mutant mice (see Fig. 2 and Supplementary Fig. 3) is a consequence of dysregulated dynamics of VE-cadherin at the adherens junctions in vivo. The altered distribution patterns of VE-cadherin suggest impaired EC activation and remodeling processes, further supporting the important role of FLRT2 in maintaining the proper organization and functionality of the CNS vasculature.

**Fig. 3 | FLRT2 controls VE-cadherin recycling at EC junctions. a** Proximity ligation assay (PLA) in HUVEC. Cells were stained with phalloidin (actin filaments) and DAPI (nuclei). **b** PLA signal quantification as puncta per cell number (DAPI) in the field. $n = 20$ and 23 pictures per condition from 3 experiments. Two-tailed Mann-Whitney test, $p < 0.0001$. **c** Immunoprecipitation (IP) of VE-cadherin and immunodetection of FLRT2 and VE-cadherin from mouse brain lysates. TL, total lysate. **d** FLRT2 immunoblot from HUVECs treated with *Flrt2* and control siRNA. Loading control: β-actin. **e** FLRT2 protein quantification in control and *Flrt2*-siRNA treated HUVECs. $n = 6$ independent experiments. Two-tailed unpaired *t*-test, $p = 0.001$. **f** Antibody feeding assay in HUVEC transfected with control and *Flrt2*-siRNA treated with chloroquine. Cells immunostained for internalized VE-cadherin, total VE-cadherin and DAPI. Intensity of internalized VE-cadherin shown in arbitrary units (AU, upper panels). **g** Fluorescence intensity quantification of internalized VE-cadherin per cell. $n = 84$ control, 115 *Flrt2*-siRNA cells from 3 experiments. Two-tailed Mann-Whitney test, $p < 0.0001$. **h** Quantification of VE-cadherin intensity per

cell-junction length. $n = 22$ control, 18 *Flrt2* siRNA-treated pictures from 3 experiments. Two-tailed unpaired *t*-test, $p = 0.029$. **i** VE-cadherin and α-tubulin (loading control) immunoblots in HUVEC transfected with control- and *Flrt2*-siRNAs. **j** Quantification of VE-cadherin/loading control and cleaved VE-cadherin/loading control ratios. $n = 11$ (total), 8 (cleaved) experiments. Two-tailed unpaired *t*-test, $p = 0.074$ (total), $p = 0.0001$ (cleaved). **k** Primary mouse brain ECs (pmBEC) from control and *Flrt2*$^{i\Delta EC}$ littermates stained for Calpain-2 and DAPI. **l** Calpain-2 fluorescence intensity quantification per cell. $n = 28$ and 51 cells per condition, from 1 control and 1 *Flrt2*$^{i\Delta EC}$ littermate. Two-tailed Mann-Whitney test, $p = 0.011$. **m** Vessels stained for Calpain-2 and IB4 from control and *Flrt2*$^{i\Delta EC}$ littermates. **n** Calpain-2 fluorescence intensity quantification in vessels. $n = 5$ controls, 4 mutants. Two-tailed unpaired *t*-test, $p = 0.002$. Scale bars: 40 μm (**a**, upper panels), 15 μm (**a**, lower panels), 10 μm (**f**, **k**), 5 μm (**m**). Data are shown as mean ± SEM. *$P < 0.05$, **$P < 0.01$, ***$P < 0.001$, ns = not significant.

## High resolution imaging of VE-cadherin junctions in tissue

To gain a better understanding of the relationship between the dynamics of VE-cadherin at the adherens junction and endothelial sprouting activity, we aimed to visualize the subcellular molecular arrangement of VE-cadherin in its native tissue environment at high resolution. For this purpose, we employed the technique of expanding cerebral cortical slices, as described above for the retina (see Fig. 2c), and analyzed and reconstructed capillaries corresponding to layer II/III of the cortex. By employing expansion microscopy, we were able to visualize the association of FLRT2, VE-cadherin, and Numb in adherens junctions at the interface between ECs in the native tissue (Fig. 6a). Likewise, we also captured the close association of FLRT2 and Numb with VE-cadherin at the retinal vasculature using expansion microscopy (Fig. 6b, c). Our 3D visualization of VE-cadherin arrangement in control mice (Fig. 6d) revealed an active pattern. VE-cadherin was localized to the adherens junctions between ECs and, interestingly, we also observed VE-cadherin in vesicles close to the membrane (highlighted in boxes 1 and 2 in Fig. 6d). This pattern suggests that VE-cadherin undergoes dynamic recycling in stalk cells, which constitute the ECs of the mother vessels. Notably, in tip cells, VE-cadherin was predominantly found in vesicles occasionally present within filopodia (highlighted in box 3 in Fig. 6d). The deletion of FLRT2 in vessels led to changes in the distribution of VE-cadherin in the mother vessels. Filopodia extension in the capillaries was infrequent, and this was accompanied by a continuous distribution of VE-cadherin at the endothelial adherens junctions. Furthermore, there was a clear reduction in the presence of VE-cadherin within intracellular vesicles, consistent with our findings in cell culture where the absence of FLRT2 led to increased internalization and rapid degradation (see Fig. 3).

## FLRT2 deficiency disrupts tight junctions and the blood-brain barrier

In CNS blood vessels, the dynamic formation of new angiogenic sprouts and growth of the vasculature need to be coupled to the assembly of tight intercellular junctions to acquire BBB properties. Interestingly, BBB properties reside in both capillaries and venules[54], where we have most of FLRT2 expression in the neocortex. Stable adherens junctions are considered necessary for the formation and regulation of tight junctions. This junctional crosstalk involves VE-cadherin, which can regulate the expression of the tight junction protein claudin-5, the most abundant tight junction in the brain vasculature[55], through β-catenin nuclear localization and its association with the transcription factor FOXO1[56]. Interestingly, Calpain-mediated cleavage of VE-cadherin occurs between the β-catenin and p120-binding domains within the cadherin cytoplasmic tail, and the cleavage of N-cadherin cytoplasmic domain binding to β-catenin has been shown to modulate cell-cell adhesion and nuclear accumulation of β-catenin[43,57].

Based on these observations, we hypothesized that FLRT2, as a modulator of VE-cadherin trafficking, may also contribute to the regulation of β-catenin nuclear transport and thereby control the expression of junctional proteins and the tightness of the BBB. To investigate this hypothesis, we silenced *Flrt2* expression in brain ECs and analyzed the pathway in both nuclear and cytoplasmic protein fractions. Consistent with our hypothesis, we observed increased levels of both β-catenin and FOXO1 proteins in the nuclear fraction in the absence of FLRT2 (Fig. 7a–c). Analysis of mRNA in control and *Flrt2*-silenced ECs revealed reduced expression of the gene encoding Claudin-5 (*Cldn5*) (Supplementary Fig. 6a). This result was further confirmed at protein level (Fig. 7d, e). Importantly, we found that *Cldn5* is downregulated in the whole-brain tissue of *Flrt2*$^{i\Delta EC}$ mice compared to control littermates at both the mRNA and protein levels (Supplementary Figs. 6b and 7f, g). To examine the interplay between VE-cadherin and Claudin-5 in the absence of FLRT2, we co-labelled cortical vasculature with both junctional markers. Interestingly, while in control animals both junctional proteins overlap delineating the cell membrane, in FLRT2-deficient animals Claudin-5 frequently appeared split in two lanes encasing VE-cadherin (Fig. 7h). Quantification showed a significant increase in this split pattern in FLRT2-deficient mutants (Fig. 7i). These results indicate that FLRT2 deletion modulates both adherens and tight junctions and their interaction.

Next, we examined if other junctions were altered in the absence of FLRT2 in the vasculature. We analyzed zona occludens-1 (ZO-1), a scaffolding protein linking Claudin-5 to the cytoskeleton[58], and the junctional adhesion molecule-A (JAM-A), which regulates Claudin-5 expression and BBB integrity[59]. The expression of ZO-1 and JAM-A was not affected in *Flrt2*$^{i\Delta EC}$ mice compared to control littermates (Supplementary Fig. 6c–h), suggesting that FLRT2-mediated changes in VE-cadherin expression specifically regulate Claudin-5.

The tight junction Claudin-5 has a particular role in regulating the paracellular passage of small molecules through the brain endothelium, since Claudin-5 full abrogation[60] or mosaic endothelial specific deletion[61] causes size-selective BBB permeability. To investigate the role of FLRT2 in modulating barrier permeability in vivo, we performed experiments using a small-sized tracer, AlexaFluor555-cadaverine (~950 Da), injected into control and *Flrt2*$^{i\Delta EC}$ mice at P7-P8. After allowing the tracer to circulate for 90 minutes[62,63], we observed a significant increase in AlexaFluor555-cadaverine tracer extravasation into the brain parenchyma of *Flrt2*$^{i\Delta EC}$ mice (Fig. 7j, k). Extensive areas of leakage were visualized in the cerebral cortex of mutant mice (Fig. 7l). Next, to investigate the size-selectivity of BBB disruption, we injected control and mutant animals with AlexaFluor555-ovalbumin, a larger tracer (45 kDa). Our findings indicated that the BBB integrity remained intact in both control and mutant animals (Supplementary Fig. 6i, j), highlighting the size-selective function of Claudin-5 in BBB tightness regulated by endothelial FLRT2. In addition, when the AlexaFluor555-cadaverine tracer was injected at an earlier timepoint (P5), there was

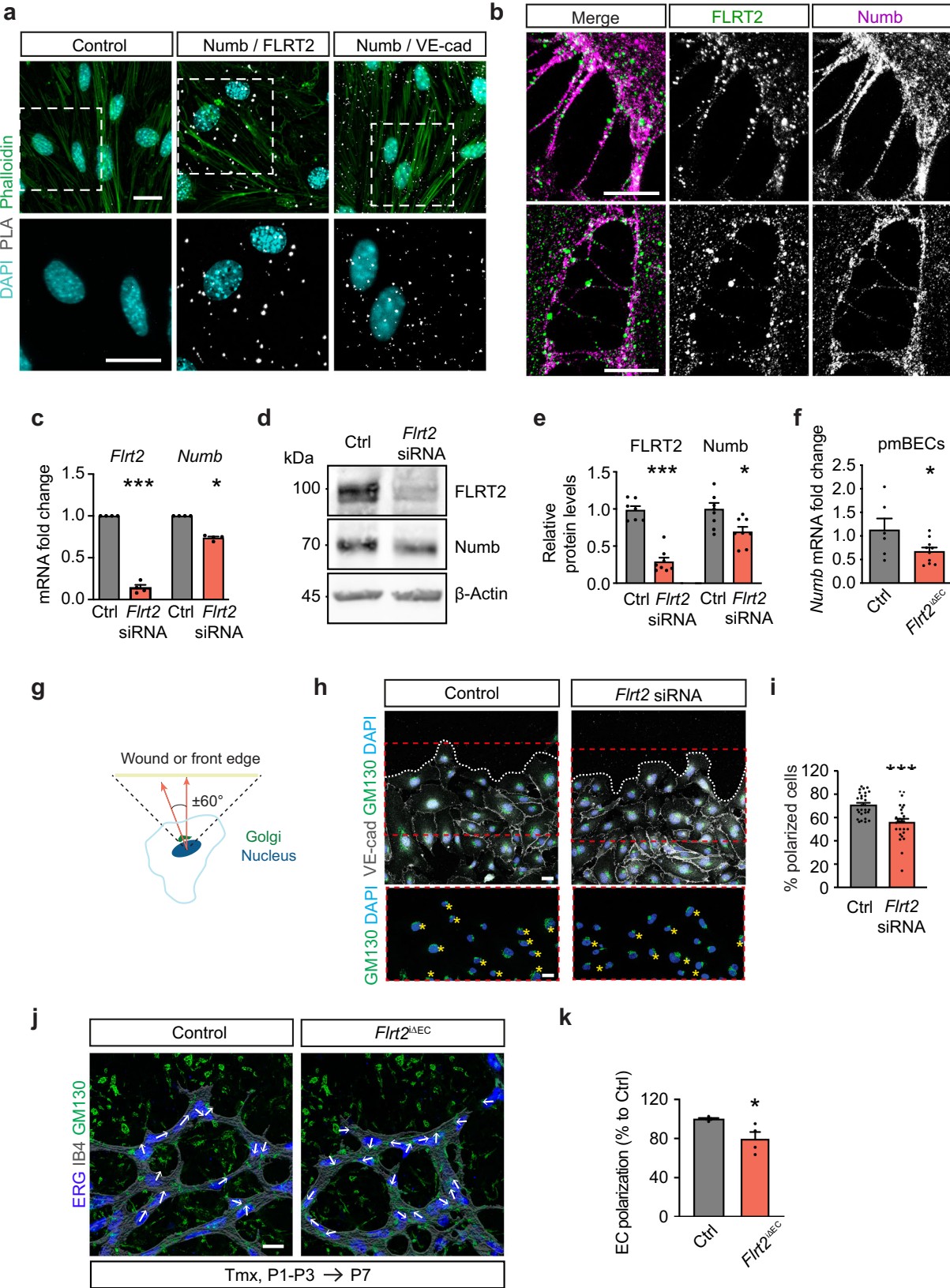

no increased extravasation observed in *Flrt2*^iΔEC brains (Supplementary Fig. 6k, l), indicating that the defective sprouting phenotype observed at this time point (see Supplementary Fig. 3) is not a consequence of a disrupted barrier.

To further confirm that BBB leakiness could solely be attributed to altered tight junctions, we assessed whether other components of the neurovascular unit were affected by FLRT2 absence in the vasculature. Pericyte coverage of vessels, detected by PDGFRβ staining, showed no differences between control and mutant brains (Supplementary Fig. 7a, b). Similarly, the coverage of vessels by astrocytic end-feet labeled with the water channel marker Aquaporin4 was comparable between both genotypes (Supplementary Fig. 7c, d). It's important to

**Fig. 4 | FLRT2 interacts with Numb and modulates EC polarization. a** Proximity ligation assay (PLA) in EC cultures. Cells were stained with phalloidin and DAPI to visualize actin filaments and nuclei, respectively. **b** Expansion microscopy on HUVEC cultures stained for FLRT2 and Numb. **c** mRNA expression of *Flrt2* and *Numb* in ECs transfected with control and *Flrt2*-specific siRNAs. *n* = 4 independent experiments. Two-tailed unpaired *t*-test, *p* < 0.0001 (*Flrt2*), *p* = 0.031 (*Numb*). **d** Immunoblot showing FLRT2 and Numb protein levels in ECs transfected with control and *Flrt2*-specific siRNAs. **e** FLRT2 and Numb protein levels quantification in ECs transfected with control and *Flrt2*-specific siRNAs. *n* = 7 (control) and 8 (*Flrt2*-siRNA) independent experiments. Two-tailed unpaired *t*-test, *p* < 0.0001 (*Flrt2*), *p* = 0.012 (*Numb*). **f** mRNA expression quantification of *Numb* from primary mouse brain ECs (pmBECs) from control and *Flrt2*$^{i\Delta EC}$ littermates. *n* = 6 control and 10 mutant animals. Two-tailed unpaired *t*-test, *p* = 0.0499. **g** Scheme showing the quantification of Golgi orientation in migrating ECs. Cells were classified as polarized if the angle formed between the scratch or vascular front and Golgi located within 120°. **h** Scratch assay on HUVECs stained for VE-cadherin, Golgi apparatus (GM130) and cell nuclei (DAPI). Stars indicate cells polarized towards the wound. The 3 first cell rows were considered for quantification. **i** Quantification of the percentage of cells per image polarized towards the wound. *n* = 28 control and 27 *Flrt2* siRNA images from 3 independent experiments. Two-tailed Mann-Whitney test, *p* = 0.0001. **j** Representative images of retinal vascular front from control and *Flrt2*$^{i\Delta EC}$ littermates stained for blood vessels (IB4), EC nuclei (ERG) and Golgi apparatus (GM130). Arrows indicate cellular orientation identified with GM130 position relative to ERG staining. **k** Quantification of the percentage of cells polarized towards the vascular front. The 3 first cell rows were considered for quantification. *n* = 4 animals per genotype. Two-tailed Mann-Whitney test, *p* = 0.029. Scale bars: 20 μm (**a**, **b**, **h**, **j**). Data are shown as mean ± SEM. *\*P* > 0.05, \*\*\**P* < 0.001.

note that during the first postnatal week, the vasculature is not yet fully ensheathed by astrocytic processes[64]. Additionally, the basal membrane protein Collagen IV showed similar coverage of the vasculature in both control and *Flrt2*$^{i\Delta EC}$ animals (Supplementary Fig. 7e, f).

To get an anatomical insight into the compromised barrier permeability, we performed transmission electron microscopy (TEM) after AlexaFluor555-cadaverine application. Punched samples from specific leakage areas (Supplementary Fig. 8) were processed for TEM and images from round capillaries (5.5–8.5 μm diameter) were acquired. High magnification images allowed the identification of abnormal EC junctions with tortuous appearance as well as the presence of large vacuoles ( > 100 nm diameter) close to the junction (Fig. 7m). The analysis of tight junctions revealed a significant increase in the incidence of abnormal junctions and vacuoles in mutant mice compared to controls (Fig. 7n). In all, these findings indicate that the absence of FLRT2 in the vasculature results in compromised junctions, both adherens and tight, leading to increased BBB permeability.

Therefore, our results unveiled FLRT2 as a regulatory hub of the junction dynamics in ECs during angiogenesis. FLRT2 expression is crucial for the development of an active sprouting angiogenesis as well as a tight assembly of the BBB.

## Discussion

Our study sheds light on the function of FLRT2 in modulating endothelial junctional complexes during CNS postnatal angiogenesis. FLRT2 is expressed selectively in veins and capillaries and enriched at intercellular junctions. Our findings indicate that vascular deletion of FLRT2 initially causes cell cycle arrest, diminishes EC sprouting, and leads to morphological changes in adherens junctions. Consequently, this results in reduced EC proliferation, decreased vascular density, and altered morphology of both adherens and tight junctions, ultimately contributing to BBB disruption (Fig. 8a). Importantly, we identified the adherens junction protein VE-cadherin as an interacting partner of FLRT2. FLRT2 forms a complex with the endocytic adaptor protein Numb to fine-tune the internalization and turnover of VE-cadherin at the cell membrane (Fig. 8b). Deletion of FLRT2 specifically in ECs induces the proteolytic cleavage of VE-cadherin by Calpains, leading to its lysosomal degradation and repression of Claudin-5 transcription. Conversely, as compensatory mechanism, FLRT2 deficiency promotes the de novo biosynthesis of VE-cadherin and its accumulation at the cell membrane (Fig. 8b). Overall, FLRT2 plays a relevant role in regulating vein sprouting angiogenesis for vascular growth and in cross-regulating tight junctions to establish BBB properties.

FLRTs have traditionally been studied primarily in the context of neuronal development, with a focus on cell adhesion or repulsion in neurons to regulate neuronal migration during mammalian cerebral cortex morphogenesis (reviewed in ref. 15). Additionally, they play roles in axon guidance and synapse formation[8,9,14,65]. Our research, alongside other studies[13,18,19], has unveiled unknown functions of FLRT2 in the vascular system, being identified as part of the group of proteins known as angioneurins. Angioneurins play crucial roles in mediating crosstalk and coordination between the vascular and nervous systems, facilitating the development and functioning of both systems[66]. By identifying the involvement of FLRT2 in vascular morphogenesis, EC cell cycle, EC dynamics, and BBB integrity, our findings underscore the emerging significance of FLRT2 as an angioneurin. Further exploration of the functions of FLRT2 and other angioneurins holds promise for deeper insights into the interconnectedness of nervous and vascular systems and their implications in various physiological and pathological processes.

Angiogenic sprouting is tightly linked to the cell cycle in physiological conditions: tip cells exhibit cell cycle arrest characterized by elevated p21 expression, while stalk cells actively proliferate to expand the vascular plexus[32]. Our findings demonstrate that FLRT2 not only impairs EC sprouting but also halts EC cell cycle progression beyond the vascular front, stalling CNS vascular growth. Importantly, vascular senescence is marked by increased p21 expression in ECs, irrespective of the stressors causing cellular aging[67]. Recent studies have revealed that FLRT2 expression diminishes with age in human vessels and in aortas of aged rats compared to young counterparts, coinciding with an increased p21 expression in older tissues, indicating that loss of FLRT2 in ECs is associated with vascular senescence[68]. This data is consistent with our observations and enhances the relevance of FLRT2 in vascular development and its maintenance. Furthermore, considering the link between cell cycle states and arterio-venous specification[69], it is plausible that FLRT2's venous zonation and its regulation of cell cycle progression also influence arterio-venous fate.

Until now, investigations into FLRT signaling have primarily centered on its heterotypic binding with other membrane proteins, either in cis or trans configurations. Notable examples include interactions with Latrophilin, Unc5, and Teneurin[17,18,70–72] as well as homotypic binding to other FLRTs[73]. The equilibrium between adhesive and repulsive signals is intricately regulated through the formation of complex clusters involving the extracellular domains of these membrane-bound molecules[15]. In contrast, the understanding of intracellular signaling pathways mediated by the C-terminal tail of FLRTs remains limited. The C-terminus domain in FLRT2 was shown to interact with the intracellular domain of FGFR2 constituting a positive feedback regulatory loop of FLRT2 on FGF signaling in craniofacial tissues[74]. Here we show the formation of a complex FLRT2/Numb/VE-cadherin essential for sprouting angiogenesis and barriergenesis. We provide evidence that FLRT2 loss of function in ECs leads to increased VE-cadherin internalization and protein cleavage by Calpains which fates cadherins for a degradative rather than recycling pathway as it has been shown for N-cadherin and E-cadherin in other systems[57,75,76]. Interestingly, C-cadherin endocytosis has also been associated with the binding of FLRT3 to the small GTPase Rnd1[77], suggesting that FLRT cytoplasmic tails emerge as important signaling hubs that might have a general function in modulating endocytosis of cadherins.

**a**

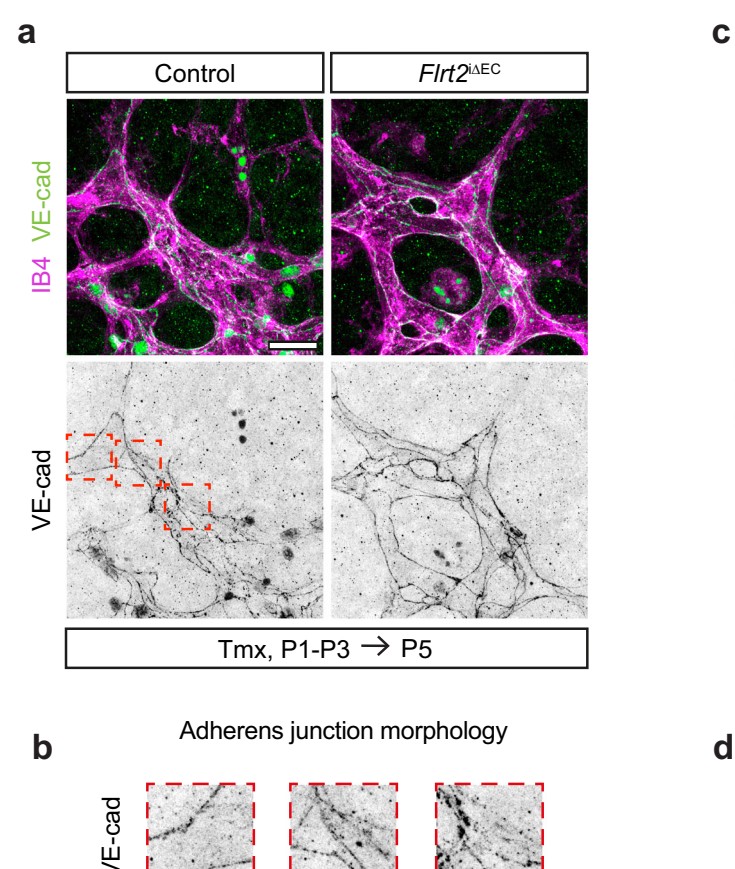

**c**

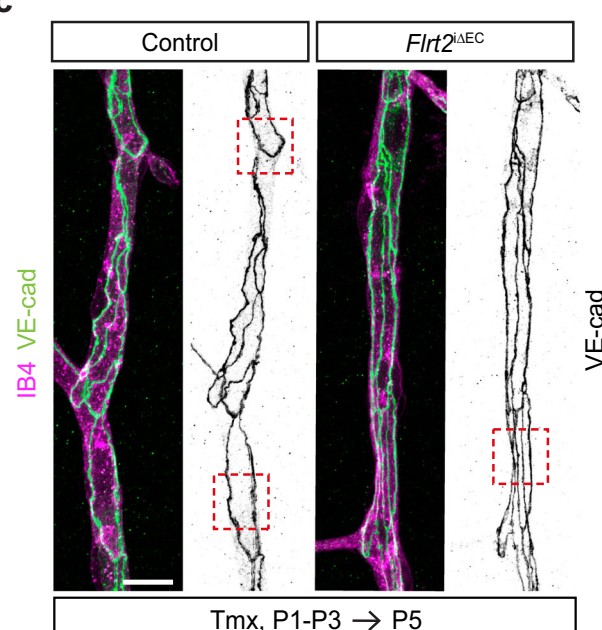

**b**  Adherens junction morphology

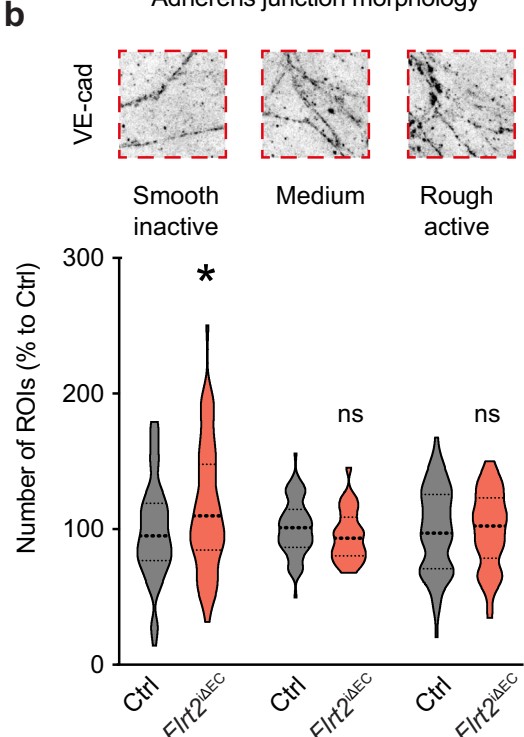

**d**  Adherens junction morphology

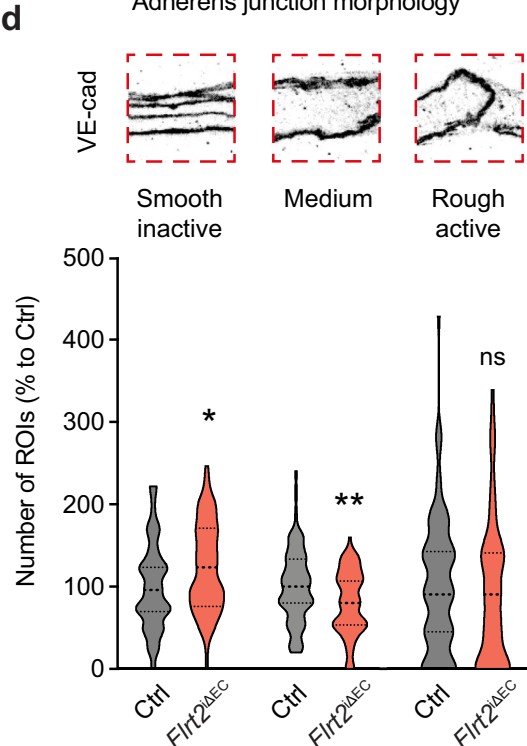

**Fig. 5 | Deletion of FLRT2 affects the morphology of adherens junctions in vivo.** Vascular front of P5 retinas (**a**) and remodeling capillaries in cerebral cortex (**c**) from control and *Flrt2*^iΔEC littermates stained for VE-cadherin antibody. Blood vessels visualized with IB4. Quantification of VE-cadherin activity in retina (**b**) and cerebral cortex (**d**) blood vessels. 15 × 15 µm regions of interest (ROIs) were blindly classified to a VE-cadherin activity category: low (smooth pattern), medium (irregular pattern), high (rough pattern). Example images of the three VE-cadherin activity categories in retina (**b**) and cortex (**d**) vessels are shown in correlation with the graph. $n = 66$ control and 56 mutant images from three litters (**b**), 78 control and 80 *Flrt2* mutant images from two litters (**d**). Two-way ANOVA, $p = 0.012$ (low), 0.693 (medium), 0.999 (high) for **b**; 0.023 (low), 0.009 (medium) and 0.854 (high) for **d**. Scale bars: 20 µm (**a**), 15 µm (**c**). Data are shown as median (thick dashed line) and quartiles (thin dashed lines). *$P > 0.05$, **$P < 0.01$, ns = not significant.

Our findings emphasize the crucial role of FLRT2 in regulating the subcellular dynamic arrangement of VE-cadherin. Visualizing differential distribution of proteins in the different subcellular compartments within cells in their physiological environment represents a challenging higher level of complexity in sample preparation, immunolabeling and imaging. To increase spatial resolution in our samples, we used expansion microscopy[78,79]. Expansion microscopy aims to enhance resolution by increasing sample size and allowed us to

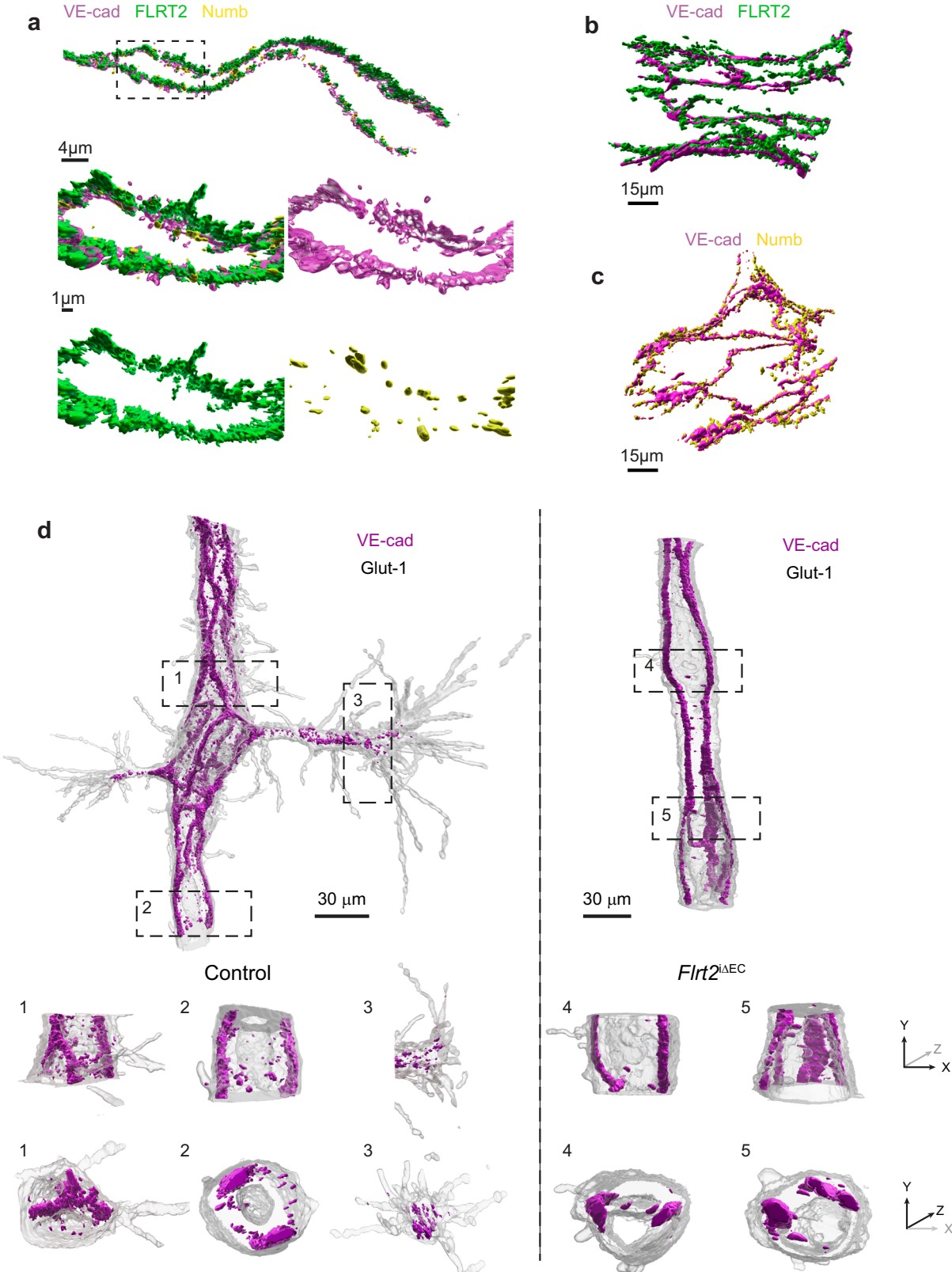

**Fig. 6 | High resolution imaging of VE-cadherin junctions in tissue. a** Expansion microscopy 3D-visualization of a large vessel stained for VE-cadherin, FLRT2 and Numb in the cerebral cortex. Higher magnification (lower panels) showing the colocalization of the three proteins at EC junction. Expansion microscopy 3D-visualization of a large vessel stained for VE-cadherin and FLRT2 (**b**) and VE-cadherin and Numb (**c**) in the retina. **d** Expansion microscopy 3D-visualization of cerebral cortex capillaries stained for VE-cadherin and Glut1 in control (left) and *Flrt2*[iΔEC] (right) mice. Higher magnifications (1-5) showing x-y planes (first row) and y-z planes (second row) exposing VE-cadherin pattern in a control capillary (1-2), a control tip cell (3), and a *Flrt2*[iΔEC] capillary (4-5). Scale bars: 4 µm (an upper panel), 1 µm (a lower pannels), 15 µm (**b**, **c**), 30 µm (**d**).

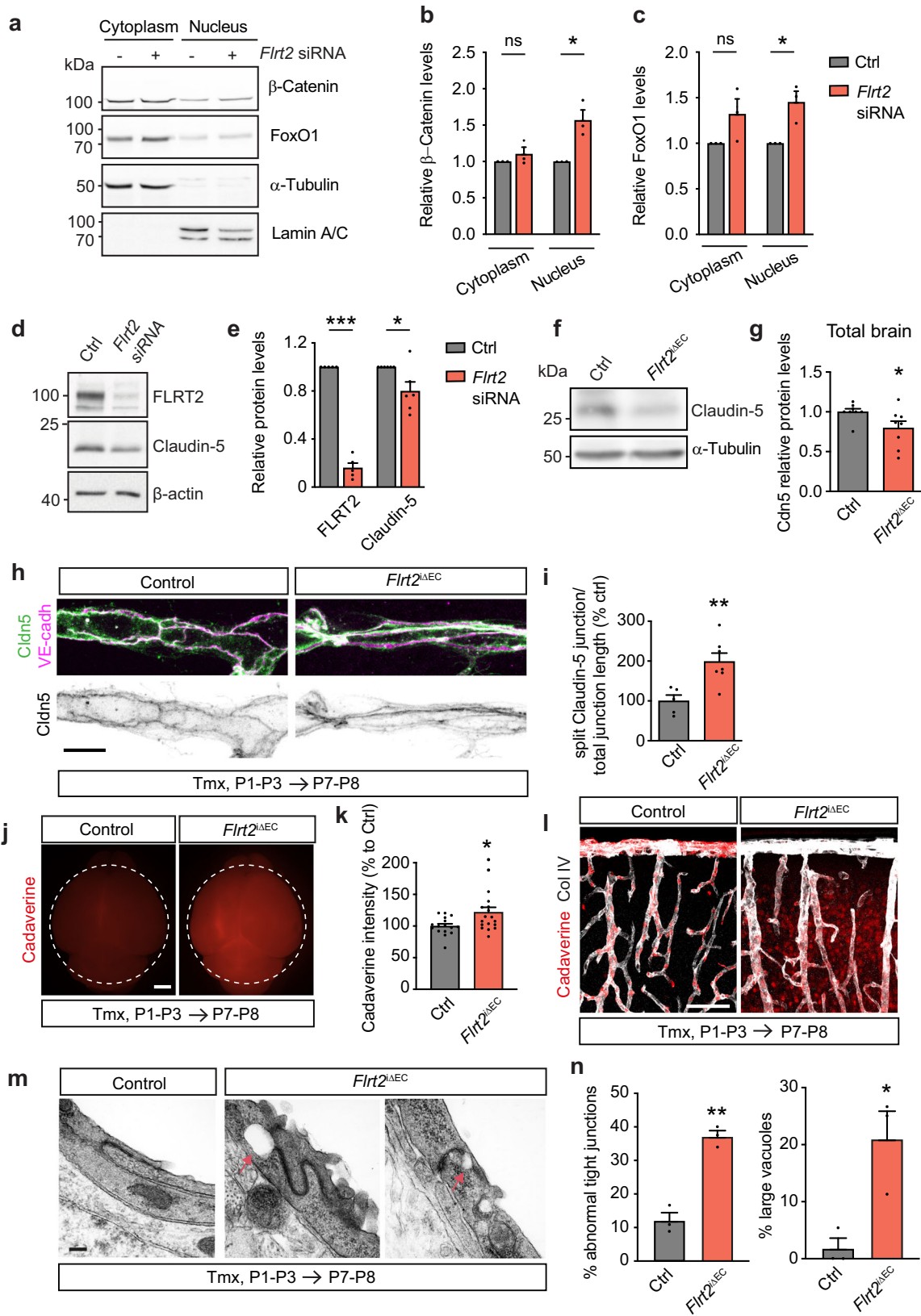

uncover FLRT2 proteins specifically at veins in retinal tissue, to outline the tripartite complex of VE-cadherin/FLRT2/Numb at the cell junctions, to localize FLRT2/Numb clusters at ECs filopodia interfaces and more importantly to directly visualize the distribution of VE-cadherin in different subcellular compartments in the tissue and correlate those to the activity patterns of the vasculature in vivo.

The interplay between adherens junctions and tight junctions is pivotal for maintaining BBB tightness (reviewed in refs. 80,81). In FLRT2-deficient ECs Calpain cleavage of VE-cadherin releases the C-terminal fragment that binds to β-catenin. This results in the translocation of β-catenin and FOXO1 to the nucleus and the downregulation of *Claudin-5*. Moreover, the structural arrangement of

**Fig. 7 | FLRT2 deficiency disrupts tight junctions and blood-brain barrier.**
**a** Immunoblot of cytosolic and nuclear fractions obtained from bEnd.3 cells treated with control and *Flrt2*-siRNA showing protein levels of β-catenin and FoxO1, and α-tubulin (cytosolic) and Lamin A/C (nuclear) as controls. β-catenin (**b**) and FoxO1 (**c**) protein levels relative to loading controls. $n = 3$ independent experiments. Two-tailed unpaired t-tests, $p = 0.342$ (cytoplasm), $p = 0.016$ (nucleus) (**b**); $p = 0.126$ (cytoplasm), $p = 0.020$ (nucleus) (**c**). **d** Immunoblot showing FLRT2 and Claudin-5, and β-actin (loading control) in control and *Flrt2* siRNA-treated bEnd.3 cells. **e** Quantification of FLRT2 and Claudin-5 protein levels. $n = 5$ (FLRT2) and 6 (Claudin-5) independent experiments. Two-tailed unpaired t-test, $p < 0.0001$ and $p = 0.026$. **f** Immunoblot showing Claudin-5, and α-tubulin as loading control in total brain lysates from control and *Flrt2*[iΔEC] littermates. **g** Quantification of Claudin-5 protein levels. $n = 8$ animals per genotype. Two-tailed Mann-Whitney test, $p$ value = 0.034. **h** Neocortical blood vessels stained for Claudin-5 and VE-cadherin.

**i** Quantification of the ratio of split Claudin-5 junction length to the total junctional length. $n = 5$ control and 7 mutant mice. Two-tailed unpaired t-test, $p = 0.006$.
**j** Representative fluorescent whole-brain images of control and *Flrt2*[iΔEC] littermates injected with AlexaFluor™ 555-conjugated cadaverine at P7-P8. **k** Cadaverine whole-brain intensity quantification in control and *Flrt2*[iΔEC] littermates. $n = 16$ control, 18 mutant animals. Two-tailed unpaired t-test, $p = 0.018$. **l** Brain cortices from cadaverine-injected control and *Flrt2*[iΔEC] littermates immunostained for Collagen IV (Col IV). **m** TEM representative images of EC junctions in brain capillaries in control and *Flrt2*[iΔEC] mice. Note the abnormal junctions often associated with the presence of vacuoles (red arrows) in *Flrt2*[iΔEC] vessels. **n** Incidence of abnormal junctions (left) and junctions with vacuoles (right) in control and *Flrt2*[iΔEC] brain capillaries. $n = 3$ animals per genotype. Two-tailed unpaired t-test, $p = 0.001$ (junctions), $p = 0.021$ (vacuoles). Scale bars: 10 μm (**h**), 1 mm (**j**), 100 μm (**l**), 200 nm (**m**). Data are shown as mean ± SEM. *$P < 0.05$, **$P < 0.01$, ***$P > 0.001$.

adherens and tight junctions in *Flrt2*[iΔEC] vessel walls displays an abnormal phenotype, characterized by the unusual splitting of Claudin-5 ensheathing VE-cadherin. This suggests a potential misplacement of Claudin-5 within the polarized structure of the basolateral membrane. While unprecedent, this observation bears some resemblance to the mislocalization of tight junctions seen in hepatocytes following JAM-A silencing, which leads to overexpression of E-cadherin[82]. Therefore, FLRT2 likely fine-tunes the dynamic recycling of VE-cadherin coupled to Claudin-5 expression, crucial for sprouting angiogenesis while preserving barrier integrity. Additionally, *Flrt2*[iΔEC] mutants exhibit selective increased BBB permeability to small-size molecules, similar to mice lacking Claudin-5[60,61]. Recent studies have intriguingly demonstrated that the brain venous compartment exhibits increased susceptibility to leukocyte extravasation[83] and enhanced CNS drug delivery through EC transcytosis[84]. However, it remains unclear whether tight junctions in the vein-venule zone are regulated differently than those in arterioles and arteries, and FLRT2 might contribute to this distinctive regulation.

Several studies highlight the significance of the venous plexus as a crucial source of ECs for the formation of other blood vessel types and the expansion of an angiogenic vascular network[3,85]. Moreover, venous cells are the main vascular subtype responsible for cerebral cavernous malformations (CCMs)[86] and pathological neoangiogenesis[5] and therefore unraveling venous-specific targets could lead to improve therapeutic strategies for such angiogenesis-related diseases. FLRT2 has been associated with tumor progression in breast[87] and bladder cancer[88,89]. In this line, aberrant expression of FLRT2 has also been observed in blood vessels of colorectal cancer samples, particularly in areas associated with tumor progression and an increased presence of sprouting-like structures[19]. Furthermore, FLRT2 has been implicated in neurological diseases involving a vascular component, such as ischemic stroke[90], multiple sclerosis[91] and spinal cord injury[22], although the molecular mechanisms remain poorly understood. Our findings underscore the importance of investigating FLRT2 and its related signaling pathways in various pathological conditions, with the potential to pave the way for alternative treatment approaches for such angiogenesis-related diseases.

## Methods

### Genetically modified mice and treatments

All animal experiments were approved by the Regierungspräsidium of Darmstadt and the Veterinäramt of Frankfurt am Main and performed under the permits V54-19 c 20/15 – FR/1016, V54-19 c 20/15 – FR/2006, and V54-19 c 20/15 – FR/2010. Individually ventilated cages were used to host the mice, maintaining them under specific germ-free conditions, in 12 hours day/night light cycles, temperature 22 °C, humidity 55%, Aspen bedding, and with food and water ab libitum. To generate EC-specific *Flrt2* knockout mice (*Flrt2*[iΔEC]), *Flrt2*[lox/lox] mice (kindly provided by R. Klein) were crossed with Cdh5(PAC)-CreER[T2] (kindly provided by R. Adams). Both males and females were used for all

experiments indistinctively. The entire postnatal litter was injected with tamoxifen and mice were equally treated. Mice were allocated into different groups based on their genotype, being Cre-negative littermate animals assigned as controls and Cre-positive assigned as mutants. Cdh5(PAC)-CreER[T2] and *Flrt2*[iΔEC] mice were additionally crossed with the (Rosa26)tdTomato reporter line (Jax strain 007914) to visualize Cre activity. Cre activity was induced in newborn pups by intraperitoneal injections of 50 μl of 4-hydroxytamoxifen (Tmx) (1 mg/ml) in peanut oil for 3 consecutive days (P1 to P3). All mice were crossed with a C57BL/6 N genetic background. Mice were genotyped by PCR. Protocols and primer sequences were used as described by the distributor or donating investigator. Wild-type C57BL/6 N animals (Jackson Laboratories) were used for fluorescent in situ hybridization (FISH), immunostaining, and expansion microscopy.

Body weight from male and female animals was measured just before anesthetizing the pups for perfusion. For intracardiac perfusion and tissue fixation, mice were deeply anesthetized by intraperitoneal injection of anesthesia (ketamine, 180 mg/kg body weight, Ketavet; xylazine 10 mg/kg body weight, Rompun). Animals were intracardially perfused with cold PBS, followed by a cold fixing solution, 4% paraformaldehyde (PFA) (w/v) in PBS or 10% trichloroacetic acid (TCA) (w/v) in dH$_2$O. Mice used for FISH were directly perfused with ice-cold Rnase-free 4% PFA.

5-ethynyl-2′-deoxyuridine (EdU)-incorporation assay was performed using Click-iT Plus EdU Alexa Fluor 488 Imaging Kit (Invitrogen, C10637). Animals were injected intraperitoneally with EdU reagent at a dose of 10 μl per 1 g animal (stock solution 20 mM solution in saline), and were sacrificed after 2 hours, followed by retinas isolation. EdU-incorporation was detected following manufacturer's instructions.

For tracer injections, animals were injected intraperitoneally with 950 Da AlexaFluor555-cadaverine 30 mg/kg body weight (A30677, ThermoFisher) diluted in PBS or intracardially with 45 kDa AlexaFluor555-Ovalbumin 150 mg/kg body weight (34782, Invitrogen) diluted in saline. Intracardiac injection was performed under anesthetized mice (ketamine 100 mg/kg body weight, Ketavet; xylazine 10 mg/kg, Rompun). After 90 minutes of Cadaverine or 30 minutes of Ovalbumin circulation, animals were deeply anesthetized and directly perfused with ice-cold 4% PFA.

For transmission electron microscopy (TEM), mice were perfused with ice-cold 0.3 M HEPES, 1.5% PFA, and 1.5% glutaraldehyde (GDA) after cadaverine injection and circulation (described above). Isolated brains were postfixed overnight at 4 °C. Tracer extravasation into the brain parenchyma was localized in 80 μm thick coronal sections. Punched samples (2 mm diameter) were obtained from leakage and control areas and processed for TEM at the Institute of Neuropathology, Justus-Liebig-Universität Gießen as previously described[63]. Shortly, punched samples were post-fixed with 6% glutaraldehyde/0.4 M PBS at RT for 24 hours, washed 5 times in 0.1 M Epon-PBS and processed with a tissue processor with 1% osmium tetroxide (Leica EM

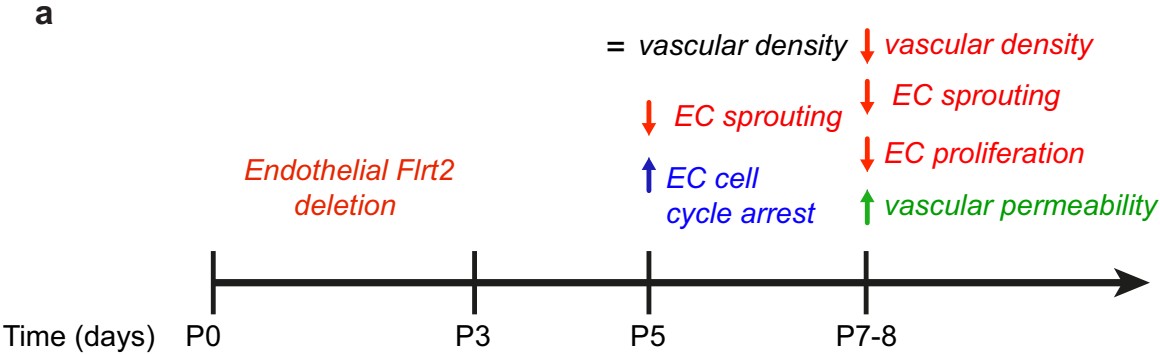

**a**

**b**

**Fig. 8 | FLRT2 regulates vascular sprouting and CNS barriergenesis by interacting with VE-cadherin. a** Schematic timeline representation of the vascular events derived from FLRT2 deletion in ECs. **b** Schematic representation of the molecular mechanisms regulated by FLRT2 in postnatal CNS vasculature. In control conditions, FLRT2 forms a complex with VE-cadherin and the endocytic adaptor Numb which allows the dynamic turnover of VE-cadherin necessary for angiogenic sprouting. By contrast, in FLRT2-deficient cells, VE-cadherin cytoplasmic tail is cleaved by Calpains and subsequently fated to lysosomal degradation, while a compensatory biosynthesis accumulates VE-cadherin at the cell membrane impairing vascular sprouts. In addition, FLRT2 deletion facilitates the nuclear translocation of β-catenin, repressing Claudin-5 expression and triggering increased size-selective BBB permeability.

TP). Samples were dehydrated by immersion in increasing ethanol concentrations (25%, 35%, 50%, 70%, 75%, 85%, 100%), and after that infiltrated with a propylene oxide and a propylene oxide/resin mixture (Agar 100 Resin Kit). Polymerization of the resin was accomplished at 60°C for 24 h. Semi-thin sections (990 nm thickness) were prepared and stained with 1% Richardson to assess overall tissue morphology. Ultrathin sections (190 nm thickness) were prepared and placed at 200 mesh copper grids (3.05 mm). Ultrathin sections were subsequently contrasted with EM AC20 (0.5% uranyl acetate/Ultrostain I and 3% lead citrate/Ultrostain II). Samples were examined and photographed using a transmission electron Zeiss microscope, type EM 109 using a 2K-CCD-Camera from TRS.

## Cell culture and treatment

Primary mouse lung EC isolation and culture was performed as previously described[92]. In short, lungs were freshly isolated from P8 mice, collected in cold dissection buffer (HBSS supplemented with 10% FBS) and treated with collagenase type II. Filtered tissue pellets were resuspended in ice-cold dissection buffer and incubated with magnetic Dynabeads (11035, ThermoFisher) coupled with rat anti-mouse CD31 (BD BioscienceS, cat 553370). Cells were resuspended in EC medium composed of high-glucose DMEM GlutaMAX-I, 1% penicillin-streptomycin (P/S), 20% FBS, 0.4% EC growth supplement with heparin, and plated onto 1% gelatin-coated plates. After 3 days, cells were passaged and they were subsequently used for further experiments once they reached confluency.

For primary mouse brain ECs (pmBECs) isolation, brains were freshly collected in dissection buffer (DMEM GlutaMAX-I, 2% bovine serum albumin, 5% penicillin/streptomycin). Tissue was minced into small pieces and digested with 1 mg/mL collagenase II, 1 mg/mL dispase II and 0.5 mg/mL DNAse 1 in dissection buffer under light agitation at 37 °C for 40 minutes. Digested tissue was then lightly triturated in dissection buffer to create a single-cell suspension. Myelin removal was performed by layering cells on top of a solution containing 22% Percoll in $Ca^{2+}/Mg^{2+}$-free D-PBS followed by centrifugation at 560 g for 10 minutes at 4 °C. Cells were strained using a 40 μm Flowmi cell strainer. Endothelial cells were enriched using Magnetic Activated Cell Sorting (MACS) using PECAM1-conjugated magnetic microbeads (130-097-418, Miltenyi Biotec) as per the manufacturer's instructions using dissection buffer throughout. Collected Pecam1+ cells were centrifuged and processed immediately for RNA isolation or seeded on glass coverslips previously coated with 1% gelatin.

Human umbilical vein ECs (HUVECs) (Lonza, CC-2519) were cultured with EGM™-2 (Lonza, CC-3202) supplemented with 1% P/S on gelatin-coated dishes. bEnd.3 cells (ATCC, CRL-2299) were cultured in high-glucose DMEM GlutaMAX-I, supplemented with 10% FBS and 1% P/S. Both cell types were authenticated by vendor or donor.

For siRNA-mediated *Flrt2* knockdown, 50–70% confluent HUVECs or bEnd.3 cells were prewashed once with OptiMEM and subsequently treated with siRNA probes at a final concentration of 67 nM mixed with Lipofectamine RNAiMax transfection reagent (13778939, Invitrogen) in OptiMEM for 4 hours at 37 °C. siRNA probes used: *Flrt2*: SASI_Hs01_00071383 (Merck), *Flrt2*: SASI_Mm01_00075874 (Merck), Firefly luciferase (customized): CGUACGCGGAAUACUUCGA[dT][dT] (sense) and UCGAAGUAUUCCGCGUACG (antisense). Knockdown efficiency was assessed by RT-qPCR and western blot 48 hours and 72 hours after transfection, respectively. Experiments on siRNA treated cells were performed 72 hours post-transfection.

For immunostaining and proximity ligation assay (PLA), HUVECs, bEnd.3 and pmBECs were cultured on glass coverslips previously coated with 1% gelatin. Subsequently, cells were fixed with 4% PFA for 15 minutes at room temperature.

For the VE-cadherin feeding assay, HUVECs were seeded onto gelatin-coated coverslips to obtain a confluent culture the next day. Cells were incubated with 10 μg/ml extracellular domain-specific VE-cadherin antibody (clone BV6, MABT134, Merck Millipore) for 1 hour at 4 °C in EBM2 basal medium with 3% BSA. Unbound antibodies were washed with ice-cold EBM2 basal medium and the cells were further incubated with EGM™-2 cell culture medium containing 150 μM chloroquine for 4 hours at 37 °C. Afterwards, cells were washed for 8 minutes with PBS containing $Mg^{2+}$ and $Ca^{2+}$ ions and supplemented with 25 mM glycine and 3% BSA (pH 2.7) and fixed with 2% PFA for 10 minutes at room temperature. Coverslips were immunostained for internalized VE-cadherin using anti-mouse-Alexa Fluor-488 (A-21202, ThermoFisher). Total VE-cadherin was detected with an antibody produced in another species (goat, AF1002, R&D Systems).

## Immunostaining and FISH

All samples from control and mutant littermates were treated and processed under the same conditions. After perfusion, brains and eyes were collected and further incubated in fixative solution for 2 hours at RT and then kept in PBS – 0.001% azide at 4 °C until further processed. Fixed brains were embedded in 4% low melting point agarose in PBS and cut coronally in 80 μm thick sections with a vibratome; fixed retinas were isolated and cut. For immunofluorescence staining, floating brain sections and retinas were permeabilized and blocked with PBS – 0.5% Triton X-100 and 10% normal donkey serum (NDS) for 1 hour at RT under gentle agitation. Tissue samples were incubated with primary antibodies in PBS – 0.3% Triton X-100 and 2% or 5% NDS for brains and retinas, respectively, overnight at 4 °C. The following primary antibodies were used: goat anti-podocalyxin 1:200 (AF1556, R&D), mouse anti-NeuN 1:200 (Millipore; MAB377), goat anti-FLRT2 1:50 (AF2877, R&D), rabbit anti-Glut1 1:200 (07-1401, Millipore), rabbit anti-collagen IV 1:200 (2150-1470, Bio-rad), rabbit anti-Calpain-2 1:200 (ab39165, Abcam), rabbit anti-ERG 1:200 (ab92513, Abcam), rat anti-VE-cadherin 1:100 (550274, BD Pharmingen), mouse anti-GM130 1:100 (610823, BD Biosciences), rabbit anti-Numb 1:100 (2756, Cell Signalling), mouse anti-Claudin-5 1:100 (35-2500, Invitrogen), rabbit anti-RCFP 1:200 (632475, Takara Bio), rabbit anti-cleaved caspase-3 1:100 (9661, NEB), rat anti-p21 1:100 (ab107099, Abcam), rat anti-JAM-A 1:500 (MABT128, Millipore), mouse anti-ZO-1 1:200 (3309100, Invitrogen), goat anti-PDGFRβ 1:100 (GT 15065, Neuromics), and rabbit anti-Aquaporin 4 1:100 (AB2218, Millipore). Next, tissue samples were washed 3 times with PBS – 0.1% Tween®20 or TBS – 0.1% Tween®20 for 5 – 10 minutes and incubated with secondary antibodies in PBS and 2% and 3% NDS for brains and retinas, respectively, for 2 hours at RT or overnight at 4 °C. The following secondary antibodies conjugated to Alexa Fluor −488, −555, −568, −594, and −647 were used: donkey anti-mouse/rabbit/goat/rat 1:200 (Life Technologies). Additional staining with Isolectin GS-IB4 conjugated to Alexa −488, −568, and −647 (Invitrogen) 1:200, and 4 ',6-diamidino-2-fenilindol (DAPI) 1:500 was performed. Stained brain sections and retinas were flat mounted with mounting medium (S302380-2, Dako) using microscope slides and glass coverslips.

ECs cultured on coverslips were fixed with 10% TCA or 4% PFA for 10 minutes at RT. Cells were permeabilized 10 minutes at RT with PBS – 0.3% triton X-100, washed with PBS and blocked with 5% NDS in PBS for 30 minutes at RT. Cells were incubated with primary antibodies in PBS with 5% NDS overnight at 4 °C. The following primary antibodies were used: mouse anti-VE-cadherin 1:100 (sc-9989, Santa Cruz), mouse anti-VE-cadherin 1:100 (MABT134, Millipore), goat anti-VE-cadherin 1:100 (AF1002, R&D), rabbit anti-Calpain-2 1:100 (ab39165, Abcam), rabbit anti-Calpain-1 1:100 (2556, Cell signaling), goat anti-FLRT2 1:50 (AF2877, R&D), rabbit anti-Numb 1:100 (2756, Cell Signaling), mouse anti-GM130 1:100 (610823, BD Biosciences) and rat anti-Pecam1 1:100 (550274, BD Pharmingen). Next, cells were washed 3 times with PBS for 10 minutes each and incubated with secondary antibodies in PBS and 2% NDS for 2 hours at RT or overnight at 4 °C. Additional staining with Phalloidin conjugated to FITC 1:200 (P5282, Sigma), and 4 ',6-diami-dino-2-fenilindol (DAPI) 1:500 was performed. The secondary

antibodies listed above were used. Last, cells were washed 3 times with PBS for 10 min, and mounted with mounting medium (S302380-2, Dako) on microscope slides.

For fluorescence in situ hybridization (FISH), samples were cryopreserved by immersion in 15% and 30% sucrose-PBS solution at 4 °C, embedded in Tissue-Tek optimal cutting temperature (OCT) compound (4583, Sakura) and sliced at 16 μm with a cryostat, serially mounted on SuperFrost slides and stored at −80 °C. mRNA detection and immunostaining were performed as previously described in ref. 93 as follows: Tissue samples were first fixed with 4% PFA washed with PBS and subsequently permeabilized with proteinase K (12 μg/ml), followed by a second fixation step in 4% PFA. Next, samples were incubated in 0.2 M hydrochloric acid treatment followed by an acetylation process with 0.1 M triethanol amine-HCl/ acetic anhydrate at pH 8.0. Finally, tissue was dehydrated using ethanol solutions (60%, 80%, 95% and 100%) and air dried. Samples were incubated overnight at 65 °C with Digoxigenin-labeled anti-sense riboprobes dissolved in hybridization solution (50% Formamide, 10 mM Tris-Cl, pH8.0, 200 ug/ml yeast tRNA, 10% Dextran sulfate, 1x Denhardt's solution, 600 mM NaCl, 0.25% SDS, 1 mM EDTA, pH8.0). Next, washes were performed in 2x saline-sodium citrate buffer (SSC)/50% formamide at 65 °C during 30 min, followed by TNE (1 M Tris-hydrochloride pH 7.5, 5 M Sodium chloride, 0.5 M EDTA, pH 8.0) containing 20 μg/ml RNase A for 10 min at 37 °C, and in 2x SSC, 0.2x SSC and 0.1x SSC at 65 °C 20-min each. Prior to riboprobe detection samples were washed in TN buffer (5 M Sodium chloride, 1 M Tris-hydrochloride pH 7.5) and blocked with TNB buffer (TN buffer with 0.5% NEN Blocking Reagent). Riboprobe detection was done with Anti-digoxigenin-AP (alkaline phosphatase) antibody (Fab fragments, Roche #11093274910) in TNB overnight at 4 °C. Next, samples were washed with TNT buffer (TN buffer, 20% Tween) and incubated with the secondary antibodies in TNB for 2 hours at RT. Alkaline phosphatase activity was detected with HNPP/ Fast Red solution (Roche #11758888001) in detection buffer (100 mM Tris-Cl pH 8.0, 100 mM NaCl, 10 mM $MgCl_2$), samples were washed in TNT and TN adding DAPI in one of the washing steps. Slides were imaged using an upright laser scanning confocal spectral microscope (SP5) equipped with a water immersion objective 25x/0.95 NA or stored in PBS for a maximum of 3 days.

The plasmid to generate the *Flrt2* riboprobe was kindly provided by R. Klein. The following primers were used to generate the additional plasmids and corresponding riboprobes: *Gkn3*-Fw CAGTTATCCTCT GGATGGCTCT, *Gkn3*-Rv CAGAGGCTAGATGAGGTCTGTC, *Slc38a5*-Fw GCCTCATCTTCATCCTTCCCAG, *Slc38a5*-Rv AGCGCCAGGGTAACCC TAACA.

## Expansion microscopy (ExM)

Cerebral cortex, retinal samples and HUVEC cultured cells for ExM were fixed and stained as described above. Tissue samples were incubated with primary antibodies for 3 days and with secondary antibodies for 2 days. ExM was performed as previously described[28,94]. Briefly, stained samples were transferred to 35 mm MatTek dish (P35G-1.0-14-C, MatTek) and incubated with AcX/DMSO (1:100) in PBS overnight at RT. Next, samples were washed twice in PBS for 15 min. Gelation solution (47:1:1:1 Stock X (in g/100 ml solution: 8.6 sodium acrylate, 2.5 acrylamide, 0.15 N,N-methylenebisacrylamide, 11.7 NaCl, 1x PBS, in water): tetramethylethylenediamine (TEMED): ammonium persulfate (APS): distilled water) was prepared. A pre-incubation step with the gelling solution for 30 minutes at 4 °C was performed. Then, samples were incubated with freshly prepared gelling solution and covered with a glass coverslip (15 mm Ø). Total embedding of the sample into the gelling solution, as well as sample flatness was ensured before incubation at 37 °C for 2 hours. After gel polymerized, the coverslips were removed and 2 ml of digestion buffer (in 100 ml solution: 0.5 g Triton X-100, 0.2 ml EDTA disodium (pH 0.8), 5 ml Triton-Cl (1 M, pH 8.0), 4.67 g NaCl, in water) containing Proteinase K

(800 U/ml) were added and incubated overnight at RT. Next, digestion buffer was removed and gel samples were incubated with distilled $H_2O$ 6 times for 20 minutes, which ensured maximum expansion efficiency. Samples were embedded in 2% low-melting point agarose in $H_2O$ in a 3 cm glass Petri dish. Tissue samples were imaged with an upright SP8 Leica confocal microscope using water immersion objectives, and cultured cells were imaged with an inverted Zeiss LSM 980 confocal microscope using a 40x water immersion objective.

## Proximity ligation assay (PLA)

Goat anti-FLRT2 antibody (AF2877, R&D), and rabbit anti-Numb antibody (2756, Cell Signaling) were conjugated to PLA oligonucleotides PLUS (DUO92009, Merck) or MINUS, (DUO92010, Merck), following manufacturer's instructions. PLA was performed using the Duolink In Situ Detection Reagents Orange kit (DUO92007, Sigma-Aldrich). Fixed HUVECs, bEnd.3 cells and pmBEC were permeabilized 4 to 10 minutes at 4 °C with PBS – 0.1% Triton X-100, washed for 4 minutes with PBS and then blocked with blocking solution for 1 hour at RT. Phalloidin-FITC 1:100 (P5282, Sigma), primary antibody mouse anti-VE-cadherin 1:100 (sc-9989, Santa Cruz), goat anti-VE-cadherin 1:100 (AF1002, R&D), FLRT2 antibody conjugated with PLA oligonucleotides 1:70 and Numb antibody conjugated with PLA oligonucleotides 1:100 were incubated overnight at 4 °C in PLA probe diluent solution. Then, cells were washed twice with Buffer A (0.15 M NaCl, 0.01 M Tris base, 0.05% Tween®20, pH 7.4) for 3 minutes and incubated with PLA probe anti-mouse-MINUS (1:5) (DUO92004, Sigma-Aldrich) or anti-goat-MINUS (1:5) (DUO92006) in blocking solution at 37 °C for 1 h. Next, cells were washed twice with buffer A for 3 min. Ligation buffer was diluted 1:5 and ligase 1:40 in MilliQ-treated $H_2O$ and incubated on samples for 30 minutes at 37 °C, followed by 2 washes with Buffer A for 2 minutes at RT. Orange Amplification Buffer was diluted 1:5 and polymerase 1:80 in MilliQ-treated $H_2O$. Cells were incubated in this solution for 100 minutes at 37 °C. Then, cells were washed with Buffer B (0.1 M NaCl, 0.2 M Tris Base, 26 g Tris-HCl, pH 7.5) for 3 min, incubated with DAPI (1:1000) in buffer B for 10 min, washed again with buffer B for 5 min, and finally washed with MilliQ-treated $H_2O$. Samples were mounted on microscope slides with Dako mounting medium and imaged with an SP5 confocal microscope. For VE-cadherin and FLRT2 and Numb and VE-cadherin interaction, signal specificity was ensured using as a control condition the addition of probe-conjugated goat anti-FLRT-2 antibody or probe-conjugated rabbit anti-Numb and PLA probe anti-mouse-MINUS, without VE-cadherin primary antibody. For Numb and FLRT2 interactions, signal specificity was ensured using as a control condition the addition of probe-conjugated Numb and PLA probe anti-goat MINUS, without probe-conjugated goat anti-FLRT2 antibody.

## mRNA extraction and quantitative PCR

For gene expression analysis in postnatal brains, samples were lysed with TRIzol® reagent (15596-018, LifeTechnologies) and mRNA was extracted following manufacturer's protocol. For cell culture and primary isolated cell samples, RNA was extracted using QIAGEN Rneasy Plus Mini Kit (74134, Qiagen) and Rneasy Micro Kit (74004, Qiagen), respectively, following manufacturer's instructions.

For reverse transcription PCR, 100 – 500 ng were used as input amount of total RNA. RNA was reverse-transcribed into cDNA using a High-Capacity cDNA Reverse Transcription Kit (4368814, Thermo-Fisher) following the manufacturer's instructions. For pmBECs samples, RNA was reversed-transcribed using $RT^2$ First Strand Kit (330401, Qiagen) following manufacturer's instructions. Quantitative PCR assays were performed using TaqMan Fast Universal PCR master mix (4304437, ThermoFisher) and TaqMan Gene Expression probes for human *FLRT2* (Hs00544171_s1), human *Cdh5* (Hs00901470_m1), mouse *Flrt2* (Mm03809571_m1), mouse *Flrt3* (Mm01328142_m1), mouse *Numb* (Mm00477927_m1), and mouse *Cldn5* (Mm00727012_s1). Human *B2M*

(Hs00187842_m1) and mouse *B2m* (Mm00437762_m1) served as endogenous controls. qPCR was performed using a Stop One Plus Real-Time PCR System (Applied Biosystems).

## Protein extraction, co-immunoprecipitation, and western blot

For brain protein extraction, neocortical samples were mechanically disaggregated with autoclaved plastic micropistilles in RIPA lysis buffer (150 mM sodium chloride; 1% Triton X-100; 0.5% sodium deoxycholate; 0.1% SDS; 50 mM Tris, pH 8.0 and 1%) complemented with complete protease inhibitor cocktail (PIC) (Complete EDTA-free Proteinase inhibitor cocktail tablets (11836170001, Roche)), 0.08% sodium fluoride, 0.44% sodium pyrophosphate and 10% vanadate. Cell cultures were washed with PBS and lysed with cold RIPA lysis buffer. Next, all samples were incubated at 4 °C rotating for 30 minutes and centrifuged at 18000 g for 10 minutes at 4 °C. Supernatant was collected and protein concentration was determined using DC Protein assay kit (500-116, Bio-rad) according to manufacturer's instructions.

For VE-cadherin and FLRT2 co-immunoprecipitation from cell culture samples, three 10 cm dishes of confluent HUVECs were washed with ice-cold PBS, scraped, collected and centrifuged at 300 g for 5 minutes at 4 °C. Cells were lysed with freshly prepared IP buffer (50 mM Tris/HCl, pH 7.4, 150 mM NaCl, 1% NP − 40, 0.5% Sodium deoxycholate, 1 mM EDTA, 10 mM NaF, 20% PIC, 0.5 mM DTT and 10% vanadate) and incubated for 1 hour at 4 °C with rotation. Protein sample was centrifuged for 20 minutes at 16300 g at 4 °C and 35 µl were taken for total lysate sample. The rest of the sample was divided into three tubes and incubated overnight rotating at 4 °C with 3 µl of mouse anti-VE-cadherin antibody (MABT134, Millipore) or donkey anti-mouse IgG (A16013, ThermoFisher) as control. Next, samples were incubated with protein G-agarose beads (11719416001, Merck) in IP buffer overnight at 4 °C with rotation. Beads were collected by centrifugation and washed three times with ice-cold IP wash buffer (PIC 1:50, 0.5 mM DTT, 10% vanadate in PBS).

For VE-cadherin co-immunoprecipitation from WT brain lysates, brains were mechanically disaggregated in Co-IP lysis buffer (25 mM Tris HCl (pH 7.4); 150 mM NaCl, 5 mM EDTA, 1% NP-40 complemented with PIC (Complete EDTA-free Proteinase inhibitor cocktail tablets (11836170001, Roche). The lysates were incubated for 1 hour at 4 °C with rotation and then centrifuged for 20 min at 16300 g at 4 °C. 25 µl of the supernatant was taken for total lysate sample and the rest of the lysate was divided into two tubes and incubated overnight at 4 °C with 3 µl of goat anti-VE-cadherin antibody (AF1002, R&D), or anti-goat IgG (AB-108-C, R&D) as control. The samples were incubated with 30 µl Dynabeads Protein G (10003D, Invitrogen) for 2 hours at 4 °C with rotation. The beads were collected on the magnetic chamber, washed four times with ice-cold Co-IP lysis buffer and finally the proteins were eluted with 2X Laemmli sample buffer.

For cytoplasmic and nuclear fractions, bEnd3 cultures 48 hours after siRNA transfection were starved overnight in DMEM GlutaMAX-I, supplemented with 2% FBS and 1% P/S, and in the presence of LiCl (30 mM) to stabilize endogenous β-catenin. NE-PER Nuclear and Cytoplasmic Extraction Reagents were used according to manufacturer's instructions (78833, Thermo Scientific) to separate the nuclear and cytoplasmic fractions prior to western blot.

For western blot, samples were resuspended in loading buffer (8% SDS, 200 mM Tris-HCl pH 6.8, 400 mM DTT, 0.4% Bromophenol blue, 40% Glycerol), boiled, and loaded in acrylamide gels for SDS-PAGE protein separation and then transferred onto nitrocellulose membranes. Membranes were blocked in TBS − 0.1% Tween®20 with 5% skimmed milk powder or 5% BSA, depending on the antibody manufacturer's instructions. Membranes were incubated with primary antibodies in blocking solution overnight at 4 °C. The following primary antibodies were used: goat anti-VE-cadherin 1:2000 (AF1002, R&D), goat anti-FLRT2 1:500 (AF2877, R&D), mouse anti-Numb 1 1:1000 (sc-136554, Santa Cruz), rabbit anti-β-catenin 1:2000 (8480, Cell

Signaling), rabbit anti-FoxO1 (2880, Cell Signaling), mouse anti-Claudin-5 1:1000 (35-2500, Invitrogen), rat anti-JAM-A 1:1000 (MABT128, Millipore), and rabbit anti-ZO-1 1:1000 (61-7300, Invitrogen). Mouse anti-α-tubulin 1:2000 (A-11126, Invitrogen), mouse anti-β-actin 1:5000 (sc-47778, Santa Cruz), mouse anti-pan-cadherin 1:2000 (C1821, Sigma), and rabbit anti-lamin A/C (2032, Cell Signaling) were used as loading controls. Membranes were washed with TBS- 0.1% Tween®20 and incubated in blocking solution for 2 hours at RT with the following HRP-conjugated secondary antibodies: donkey anti-goat 1:3000 (705-035-147, Jackson Immuno Research Laboratories), goat anti-mouse 1:3000 (115-035-146, Jackson Immuno Research Laboratories), donkey anti-rabbit (10379664, ThermoFisher) and goat anti-rat (NA935V, GE Healthcare). After washing with TBS-0.1% Tween®20, HRP activity was detected using enhanced ECL western blotting detection reagents (RPN2106, GE Healthcare) and the ImageQuant™ LAS 4000 system. WB signal was measured using Image Studio™ Lite software (version 5.2.5).

## Imaging analysis and quantifications

All in vivo experiments include animals from at least two different litters unless otherwise specified. Cell culture experiments were repeated at least three times unless otherwise specified. For image quantification, at least 5 maximal intensity projection images of a 1024 ×1024-pixel size per animal or condition were used unless stated otherwise. For all brain quantifications, coronal sections of the intermediate cerebral cortex were selected for image acquisition, being the lateral ventricles visible in the more rostral sections and the anterior hippocampus in the more caudal sections. Images were acquired in the medial part of the cortex, between the interhemispheric fissure and the lateral part of the cortical hemispheres.

Total vessel length and total number of branch points were quantified using AngioTool software (version 0.6)[95]. ImageJ/Fiji (version 2.14/1.54 f) was used for the rest of image analysis. Capillary densities were quantified as % area covered by vessel maker. In retinas, capillary areas between arteries and veins were manually drawn, and the number of arteries and veins was manually counted. Vein and artery branch points, vessel sprouts, filopodia number in retinal fronts, vessel sprouts in cortical vessels, retinal empty Collagen IV sleeves, cleaved caspase-3 and p21+ ECs were manually quantified using the Cell counter plugin from Fiji, and related to the artery and vein length, vascular front length, tdTomato⁺ tip cell, vascular density, image field, % area covered by vessel maker (retina) and total vessel length (brain cortices) respectively. For p21 and caspase-3 staining, masks using the IB4 signal were used to restrict the staining to the vascular bed. Tip/stalk cell ratio was quantified as described before[96]. Briefly, ERG+ nuclei positioned at sprout structures were manually counted as tip cells with Cell Counter plugin in Fiji, while directly neighboring cells not in sprout positions were counted as stalk cells. Tip / stalk cell ratio was normalized to the vascular front length, measured with a manually drawn segmented line. Cre mediated recombination in Cdh5(PAC)-CreERT2 mice crossed with Rosa26-Stop^{lox/lox}-tdTomato mice was evaluated by dividing tdTomato % covered area by IB4 or Podxl % covered area. PDGFRβ and collagen IV coverage was quantified as % area covered per % vascular covered area. Aquaporin-4 coverage was quantified manually tracing the area around blood vessels and calculating % area covered. For cortical blood vessel Calpain-2 quantification, the IB4 signal was processed using thresholding to generate a blood vessel mask and applied stack-wise automatically. This mask was subsequently employed to delineate regions of interest for quantifying integrated intensities of Calpain-2 as well as measuring the area covered by the blood vessel mask. Simultaneously, mean grey values of background signals in Calpain-2 channels were quantified in at least five distinct locations. The data were then processed using Python to compute the corrected total fluorescence (CTF) values (integrated intensity / (mean background x area)) for each image. Claudin-5

junctional quantification was performed using SNT ImageJ's framework for tracing Claudin-5 split junctional length and referred to VE-cadherin total junctional length.

For PLA puncta quantification in HUVEC stainings, phalloidin staining was used to create a mask for cell surfaces and PLA puncta were counted only within this mask. Total number of PLA dots were then referred to total number of DAPI nuclei per image. For PLA puncta quantification in pmBEC stainings, PLA puncta within whole cells were manually counted using Cell counter plugin from Fiji. For VE-cadherin internalization, Calpain-1 and −2, and FLRT2 quantifications in HUVECs and pmBECs, individual cell surfaces from confocal images were manually traced and the total intensity (Integrated Density) of the staining was quantified. Three small circular ROIs per image were used to calculate the background (Gray Value). Corrected Total Fluorescence (CTF) was calculated with the formula: Integrated density − (Area × Background Mean Gray Value). For VE-cadherin junctional intensity, cell junctions were manually traced with the freehand line tool from Fiji, creating ROIs with a 25-pixel width along cell-cell contacts. Signal intensity (Raw Integrated Density) was measured in the ROIs and divided by the ROI length.

Cell polarity assessment was performed as previously described[53]. In short, the Golgi apparatus orientation was measured by connecting the position of GM130 signal to the center of the mass of the cell nuclei, stained with DAPI or ERG. Cells were considered polarized when the angle towards the wound or sprouting front was within ± 60 degrees.

For the analysis of VE-cadherin activity levels in tissue samples, a Fiji macro script was written based on a similar analysis included in ref. 34. Confocal images of VE-cadherin and IB4 as blood vessel marker (BVM) at 63x magnification and 1.6x zoom were acquired in the vascular front area of P5 mouse retinas and in capillaries in upper and lower cortical layers of the cerebral cortex. In Fiji, an automatic threshold was applied to the BVM channel with the Li method and then converted into a mask. Within the area outlining the BVM mask, small 100 ×100-pixel ROIs were created (15 μm × 15 μm). Only ROIs with a minimum overlap of 300 pixels with the BVM mask were included, minimizing the number of "empty" patches without VE-cadherin signal. The acquired ROIs from the BVM channel were used as a template to divide the respective VE-cadherin images into patches. These patches were then presented blindly and in a random order to the user who manually assigned an activity category to each patch. To evaluate the results, the patches were unblinded and grouped by animal. For each animal the percentage of patches falling into each category was determined and the resulting percentages of categories were normalized to the average of the control animals in the respective litter.

To measure the intensity of the emitted fluorescence of Alexa fluorophore coupled to cadaverine from isolated brains, images were obtained with a fluorescent lamp (Texas red filter, TRX) and a binocular microscope. For analysis of the images, the mean gray value measurement in the Fiji software was used. At least 3 independent round ROIs with the same area were applied to obtain the average fluorescent intensity in one brain. Subsequently, the average value per brain was divided for the average value obtained from the control animals of the respective litters.

For the TEM analysis, images of tight junction in round capillaries (5.5–8.5 mm diameter) were acquired in punches obtained from 3 control and 3 *Flrt2*[iΔEC] brains. The incidence of abnormal tight junctions in control and *Flrt2*[iΔEC] brain samples was calculated, being considered abnormal junctions the ones exposing a tortuous morphology or with large vacuoles proximal to the junction.

## Statistics

Data was obtained with measurements taken from distinct samples. Statistical analysis was performed using GraphPad Prism (version 8). First, normal distribution of the samples was tested. When samples followed a normal distribution, 2-tailed unpaired Student's *t*-tests, or

two-way analysis of variance (ANOVA) were used when comparing two variables or when comparing multiple measurements, respectively. For samples not following a normal distribution, a Mann-Whitney test was used. Statistical significance was defined as $P < 0.05$ (*), $P < 0.01$ (**) and $P < 0.001$ (***). All graph values indicate mean ± Standard Error of the Mean (SEM), unless otherwise specified.

## Reporting summary

Further information on research design is available in the Nature Portfolio Reporting Summary linked to this article.

## Data availability

Source data are provided with this paper.

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

## Acknowledgements

We thank D. Schmelzer and T. Belefikh for animal care, M. Parrilla, U. Bauer, P. Brendel, and S. Junek and C. Molenda from MPIBR imaging facility for technical support and R. Klein and R. Adams for sharing resources. We are grateful to the members of the Acker-Palmer lab. Funding: Deutsche Forschungsgemeinschaft: SFB 834 (AAP), 1080 (AAP), 1507 (AAP), 1531(MS & AAP); FOR2325 (AAP), RTG2566 (AAP), EXC 2026 (AAP); European Research Council: ERC_AdG_Neurovessel (Project Number: 669742) (AAP); Schwiete Foundation (AAP) and the Max Planck Fellow Program (AAP).

## Author contributions

C.L.-C. designed, performed and analyzed retina and BBB experiments and expansion microscopy; B.P. designed, performed and analyzed cerebral cortex and BBB experiments and expansion microscopy; P.K. contributed to retina experiments and culture experiments; L.D. performed experiments for VE-cad analysis; J.V. performed PLA experiments, immunohistochemistry and expansion microscopy; S.S., J.J. and C.M. contributed with immunoblots; N.A. performed primary brain endothelial cell isolation and data analysis; B.K. supervised animal experimentation and prepared documentation; H.S., A.S. and T.A. performed and supervised TEM data acquisition; M.S. and A.A-P. designed and supervised all stages of the project and are co-last authors. C.L-C., B.P., M.S. and A.A.P. wrote the manuscript. All authors discussed and interpreted the data and gave input to the written manuscript.

## Funding

## Competing interests

The authors declare no competing interests.
