## [Transparent Peer Review file · Nature Communications]

Vascular FLRT2 regulates venous-mediated angiogenic expansion and CNS barrierogenesis

Corresponding Author: Professor Amparo Acker-Palmer

Version 0:

Reviewer comments:

Reviewer #1

(Remarks to the Author)

Peguera et al discusses the role of FLRT2 in regulating venous-mediated angiogenic expansion and CNS barrierogenesis. The study utilized genetically modified mice to examine the impact of FLRT2 on vascular development and barrier formation in the central nervous system. The results indicate that FLRT2 plays a significant role in venous zonation and angiogenesis, with preferential localization to the endothelial cell membrane of veins. Additionally, the study suggests that venous endothelial cells contribute to angiogenesis and the formation of the arterial plexus. The research provides insights into the molecular mechanisms underlying vascular development and barrier formation in the CNS, with potential implications for understanding and potentially treating related neurological conditions. The presented study is important and well-conceived but I have identified a few points that, in my opinion, need to be address to enhance its scientific rigor.

Major

Introduction:

- 1) The introduction exhibits a certain degree of generality and would benefit from improved coherence between paragraphs. Furthermore, the authors have relied on a few reviews to substantiate their descriptions, overlooking crucial recent studies on venous vasculature. Notably, the work by Coelho-Santos et al. PNAS 2021 (PMID: 34172585) demonstrates that veins play a pivotal role in establishing the capillary bed during postnatal brain development. Moreover, findings from Lee et al. Circulation. 2021 (PMID: 34474596) highlight the significance of veins in retinal development and pathophysiology. These important and recent studies should be incorporated into the manuscript instead of relying solely on general reviews.
- 2) The reference 2 cited does not really describe the statement: 'quiescent venous endothelium can be reprogrammed to activate migratory behavior in response to angiogenic and mechanical cues'
- 3) The hypothesis or the main objective of the study is unclear.

Methods:

- 1) The rationale for housing the animals individually in ventilated cages remains unspecified, this can impact their behavior. The pups were also alone?
- 2) In this study, fluorescence was quantified using Integrated Density - (Area x Background Mean Gray Value). The primary variable for fluorescence in integrated density is the area. It is suggested to employ Mean Gray Value instead of Integrated Density, as was done for cadaverine. Is there a specific explanation for this change in the approach for measuring fluorescence?

Results:

- 1) One of the primary findings asserted by this study is the selective expression of FLRT2 in veins and capillaries. Although this can be truth for retina does not seems translated for brain vascular system. Looking carefully to the images I did not observe a clear reflection of this claim in the presented results. Specifically, in Figure 1b (brain), it is evident that Flrt2 is expressed in arteries (by the morphology) as well. The representation of capillaries was challenging to identify in the provided figures. It is crucial to distinguish arteries by labelling with alpha-smooth muscle actin and veins with EphB4 to accurately dissect this expression pattern.
- 2) In the figure, Flrt2 is not only staining vessels; it is evidently labelling other cells in both retinas and the brain. Could the author provide explanations for this observation?
- 3) The Western blot presented in the manuscript to demonstrate FLRT2 EC-specific deletion was conducted using EC-lung

isolates (Extended data Fig. 1e). Why was a similar analysis not performed in the retina or brain tissues?

4) 'The effectiveness of Tmx treatment was confirmed by evaluating Cre-mediated recombination, which resulted in tdTomato signal in approximately 80% of the vasculature in the retina of Cdh5(PAC)-CreERT2 mice crossed with the Rosa26-Stoplox/lox-tdTomato reporter line'. What about the brain?

1) In Figure 2d of the Flrt2 knockout (KO), I have observed additional sprouts that were not included in the count. I am uncertain whether this discrepancy is due to issues with the selection of the representative image, but I can identify three more sprouts upon closer examination. The same discrepancy is noticed in Figure 2h; although the cropped images appear distinctly different, upon a comprehensive assessment and counting of sprouts over the representative figures, they seem comparable between the control (ctr) and knockout (KO) groups. This should be clarified.

2) Did the animals survive beyond postnatal day 8 (P8)? Were there any observed behavioral alterations, particularly in relation to developmental milestones and the age at which eyes opened?

3) The authors did not measure blood flow, but can infer how blood flow is impacted by FLRT2 based on the EGR results?

4) Since FLRT2 was deleted in EC but not specifically in veins, what outcome does the authors anticipate for the experiment?

5) Regarding the sentence: 'Perinatal endothelial FLRT2 deletion did not have an impact on the total number of arteries and veins present in the retina (Extended data Fig. 2a, b)'. The tamoxifen-induced deletion of FLRT2 was consistently performed after birth. "Perinatal" typically refers to events occurring before birth. Was this a typographical error, or if not, could you please provide clarification?

6) In this manuscript, experiments were conducted to knock down FLRT2 in EC during postnatal CNS development to elucidate the functional role of FLRT2 in vascular development and barrier formation. What potential outcomes could arise if genetic manipulation were employed to overexpress FLRT2 in EC?

7) Was sex dimorphism considered in the analysis?

8) It would be interesting to investigate the effects of pharmacological agents that modulate FLRT2 activity, specifically employing FLRT2 agonists, to assess their potential in rescuing the impacts on vascular development, particularly in angiogenesis and blood-brain barrier properties.

9) It was mentioned that single-cell RNA sequencing was conducted to profile expression within the vascular zone, but the results were not presented. This information is crucial for the study. Additionally, revealing the gene expression patterns of endothelial cells in the presence or absence of FLRT2 could offer valuable insights into the molecular pathways and signaling networks regulated by FLRT2 during vascular development in a more unbiased manner.

10) When examining the BBB at an earlier time point (P4-P5), no alterations were observed. Could this be due to FLRT2-EC deletion not having occurred yet? The results for FLRT2-EC deletion were only presented at P7-P8, and it needs clarification whether, at the time of the analysis at P4-P5, FLRT2 was deleted in EC.

11) Consistent with this, Extended Figure 3 demonstrates that FLRT2-EC deletion does not result in capillary density differences at P4-P5; however, there are fewer vein branches and reduced angiogenesis. I tried to comprehend how this discrepancy might arise and looking into the text for some theories, but the authors did not provide an explanation. I posit that this may be crucial to understand and could be linked to the fact that FLRT2 might not be deleted yet or is less expressed at these early time points. Could the issue potentially be related to cell death, or arrested proliferation rather than an angiogenesis problem?

12) Proximity ligation assays for FLRT2 and VE-cadherin were exclusively conducted in vitro, which is essential for initial screenings but does not capture the in vivo complexity, including factors like shear stress from blood flow. Therefore, these assays should be further validated in the retina and brain to account for the physiological conditions in vivo.

13) VE-Cadherin is increased, but the BBB is leaking at P8. The authors referred to internalization and compensation. However, it appears that VE-cadherin, despite the increase, remains at the membrane, and no 'holes' were observed in the monolayer. Can the authors provide more clarification on these observations? Additionally, if there are fewer new vessels and a high expression of VE-cadherin, were the VE-cadherin quantifications normalized to the number of vessels or the length of the vasculature?

14) The observed effect of FLRT2 on VE-cadherin may represent a compensatory mechanism related to other tight junctions, particularly the decreased expression of Claudin-5, as demonstrated in the study. While it is evident how FLRT2 modulates VE-cadherin, the involvement of FLRT2 in Claudin-5 expression is less clear. It would be interesting to demonstrate the organization of Claudin-5 by immunohistochemistry and assess whether it correlates with the increased expression of VE-cadherin.

15) Considering the observed increase in BBB leakage over time following FLRT2-EC deletion, it prompts speculation that its expression is changing over time. Therefore, it is important to investigate the expression profile of FLRT2 during postnatal development in both the retina and the brain, with a particular focus on the cortical region, as the study was centered on this area.

16) How does FLRT2 regulate venous-mediated angiogenic expansion and CNS barrier genesis? What type of genes can be regulating?

17) Cadaverine (950 Da) is relatively small, and it was allowed to circulate for an excessive 90 minutes before fixation. The BBB exhibits permeability to small dyes until postnatal day 12 (P12), and according to previous research (Coelho-Santos et al., PNAS 2021 - PMID: 34172585), a mere 10 minutes after circulation is sufficient for a 10 kDa molecule to exit the circulation. Therefore, the sentence 'indicating that the defective sprouting phenotype observed at this time point is not a consequence of a disrupted barrier' might not be accurate. It could be a conclusion drawn from the experimental time frame used, as all the main vascular angiogenesis processes were already in place. This should be considered.

18) Pericytes are essential cells for the stability of the vascular system and work in synergy during angiogenesis. Were any alterations after FLRT2 deletion observed in these cells?

19) After thoroughly reviewing the results and scrutinizing the data, it appears that they do not substantiate the title 'Vascular FLRT2 regulates venous-mediated angiogenic expansion and CNS Barrier genesis.' It would be prudent to consider substituting "CNS" with "retina" in the title. Given the presented data, it is challenging to draw conclusive parallels between the impact of FLRT2 on vascular development and barrierogenesis in the brain through veins. Although recent papers have

highlighted veins as sources of angiogenesis, the labeling of FLRT2 appears to extend across other vascular cells and non-vascular cells in the cortical region.

20) The references 2,21 and 22 used do not reflect the statements and should be substituted by (PMID: 34172585) and (PMID: 34474596): Veins have been identified as the primary source of ECs for angiogenesis, and recent studies have demonstrated that venous ECs give rise to the endothelial tip cells, which are located at the leading edge of retinal vasculature and subsequently migrate into the arterial plexus 2,21-23.

21) This sentence needs to be clearer about the type of vasculature; it begins with "mutant vessels compared to control capillaries." Are the authors comparing capillaries with capillaries, or is the comparison inclusive of all types of vasculature (arteries, veins, and capillary bed)?

22) In Figure 7I, the term "abnormal junctions" is used; could the authors provide clarification regarding the criteria used for defining these junctions?

Discussion:

Jumping to conclusions without robust supporting results should be avoided. It is crucial to meticulously discuss the actual data presented in the paper. The author should delve further into the spatiotemporal events identified in this important study, emphasizing their significance for capillary formation and maturation, rather than engaging in speculation about unexplored diseases in this story. Are there any disorders where FLRT2 plays a particular role? If so, discussing those instances could be important.

Minor

1) Introduction:

- a. CNS was never described before.
- b. The authors referred 'endothelial subtypes' should state what subtypes.

2) Methods:

- a. When the authors refer to "1024 x 1024-pixel size maximal intensity projection images (with a minimum of 5 images per animal or condition)," clarification is needed regarding whether this pertains to depth, such as a z-stack, or specific Regions of Interest (ROIs). The methods section should provide explicit details.

3) Results

- a. In the figures, what is black should be white or grey – just an aesthetic note.
- b. It would be highly beneficial for readers to have a schematic depicting the timeline of events, facilitating a clearer understanding of the complexity and findings of the work.

If possible, the manuscript would benefit from some English editing to achieve a more fluid and harmonious text for improved readability.

Throughout the text, some acronyms are explained while others are not. "Blood-brain barrier" and 'endothelial cells' are sometimes abbreviated and sometimes written in full. Please standardize the usage for consistency.

Reviewer #2

(Remarks to the Author)

This interesting manuscript from Peguera and coworkers characterizes the function of FLRT2 in the vessels of the retina and the cerebral cortex. Previous studies from the same group had shown that FLRT3 interacts with Unc5 receptors to regulate retinal vascularization (Seiradake et al., 2014). However, little is known about the functions of other FLRT members in this process or about potential novel interacting partners. In this study, the authors extend this concept to FLRT2, showing that it forms a complex with Numb and VE-cadherin, modulating the recycling of VE-cadherin. This contributes to the modulation of adherens junctions in both the retina and cerebral cortex, and tight junctions that are crucial to blood-brain barrier development.

This manuscript is well-organized, reporting interesting results with an appropriate experimental design and high-quality figures. The data presented in Figures 1, 2, and 4, which show reduced overall retinal vascularization and blood vessel density in the cerebral cortex by P7-P8, reduced branching originating from veins but not arteries, reduced angiogenic sprouts, and altered adherens junctions, are very clear and convincing. The evidence supporting a model in which FLRT2 forms a complex with Numb and VE-cadherin, thereby regulating the dynamics of the latter, relies mainly on in vitro assays using HUVEC and FLRT2 siRNA (as shown in Figure 3 and most of Figure 4; although it is interesting the colocalization between FLRT2 and VE-cadherin in a cerebral cortex vessel in Fig. 6). The FLRT2-dependent regulation of junctional proteins, such as Claudin 5, is primarily demonstrated in bEnd.3 cells using FLRT2 siRNA (Figure 7). These two mechanisms are very intriguing; however, without further in vivo experiments to complement the in vitro work, questions are raised regarding their role as the major contributors to the interesting phenotype reported. For example, can the FLRT2-Numb-VE-cadherin complex be detected in vivo? Could Unc5 receptors be participating to some extent in this phenotype? Indeed, in a previous study, the authors showed that FLRT3 interacts with Unc5 receptors present in blood vessels, thereby regulating retinal vascularization

Below are some questions that should be addressed to further solidify the main conclusions of this study.

1. The authors state that FLRT2 forms a complex with Numb and VE-cadherin; however, most of the data are from in vitro studies. The authors should demonstrate that this complex forms in vivo. For example, they could perform pull-down assays from retina or cerebral cortex lysates using antibodies or recombinant proteins as bait. Another option could be to use

expansion microscopy on tissue samples, labeling all three proteins and checking that they are present at the cell surface. Regarding this, even though the colocalization between FLRT2 and VE-cadherin in 3D reconstructed vessels from the cortex is interesting (Fig. 6), I find that the VE-cadherin staining is very strong, covering almost the entire surface. Would it be possible to obtain images of the three proteins, similar to Figure 4b for FLRT2 and Numb, but with higher resolution?

2. The authors should perform a rescue experiment in their vascular-specific FLRT2 mutant to support their findings. For example, would only the intracellular domain of FLRT2 attached to the cell surface of vessels be enough to rescue the phenotype, or is the full-length version required? This could indicate whether a FLRT2-ligand is necessary to regulate this process. For example, the authors mention in the discussion that the FLRT3-Rnd1 complex regulates cadherin endocytosis. It has also been shown that Unc5B binds to this complex in cis, thereby regulating cell adhesion (Karaulanov, 2009). Would it be possible to use a mutant version of FLRT2 that does not bind to Unc5 and see whether it can rescue any phenotype?

3. The finding that calpain 1 levels increase in FLRT2 knock-down cells is very interesting. Given that the authors link this protein with both the 95 kDa VE-cadherin being targeted for degradation, and β -catenin nuclear transport and the tightness of the BBB, they should show that the levels of this protein also increase in vivo. One option could be to perform a western blot on FLRT2 conditional mutants, or to use immunofluorescence as demonstrated in Fig. 3K, to avoid any underestimation of the effect.

4. In line with previous question, Is the 95 kDa VE-cadherin form also increased in the FLRT2-mutant mice? Regarding Claudin-5 (Cldn5), the authors show a slight mRNA reduction in the total brain of mutant mice. What about the protein levels? Perhaps performing immunofluorescence could reveal a stronger effect than analyzing mRNA from the entire brain, as the latter approach may dilute the effect

5. A previous study has shown that FLRT2 is required for the proper integrity of the basement membrane during heart development (Müller P et al., 2011). Is the basement membrane of FLRT2-mutant blood vessels affected? Maybe comparing control and mutant sections stained with collagen IV could help to address this question.

6. During heart development, it has been shown that other members, like FLRT3, can compensate for the loss of FLRT2. Is FLRT3 expressed in these blood vessels, and if so, are its levels affected by FLRT2 knockdown?

Minor questions:

1. Figure 1b displays FLRT2 immunofluorescence in a blood vessel of the cerebral cortex. Is FLRT2 expressed homogeneously across the entire blood vessel throughout the cortical layers? And in rostro-caudal sections? Considering that both FLRTs (including FLRT2) and several of their ligands are specifically expressed in different cortical layers, it would be interesting to determine whether blood vessels show a homogeneous distribution of this receptor, or if there are enrichments specific to certain cortical layers.

2. Figures 3 and 4 show PLA experiments indicating close proximity between FLRT2-VE-cadherin, Numb-FLRT2, and Numb-VE-cadherin. What percentage of the puncta is present at the cell surface? From the images, it appears that a significant proportion is located in the cytosol. Labeling the plasma membrane of the cells with a specific dye could be useful in measuring this.

3. In the discussion there is a small typo. "GTPase Rdn1" should be "GTPase Rnd1".

Reviewer #3

(Remarks to the Author)

In this new manuscript by Peguera et al., authors describe the role of protein FLRT2 in the regulation of murine CNS angiogenesis and barrier genesis, with a focus on the venous origin of vascular expansion and maturation during the first postnatal week in the retina and cerebral cortex (up to P8).

The manuscript is very well written, data clearly presented, and the model is interesting. While this study overall is well performed, important concerns remain about the lack of depth in the descriptive and mechanistic aspects, as well as the lack of functional relevance of the pathway beyond the first postnatal week, thus hindering the enthusiasm of the reader. To bring this study to the next level, authors may wish to consider the following points:

Major points:

1- An important weakness of the manuscript in its present form is the lack of functional relevance to older ages (adulthood). To address this, authors could consider measuring CNS vascular structure and function in adult *Flrt2*^{DEC} mice. Even more interesting would be to measure neuronal function at P8 and in adults. What does the lack of FLRT2 in vessels cause in terms of cortical lamination? Retinal function? Mouse behaviour? Answers to the questions would bring a unique dimension to this study.

2- The effectiveness of Tmx treatment was assessed using *Cdh5-CreERT2;Rosa26StopFlox-tdTomato* mice. This raises concerns, and a preferable way would be to use a reporter that turns ON upon excision of the floxed *Flrt2* allele. I do not think the approach chosen by authors is reliable in assessing the efficiency of Cre to remove *Flrt2* from *Cdh5*-expressing cells. It is more testing Cre activity per se (which is quite different from assessing its action on the target gene).

3- GLUT-1 and Collagen IV were used to label CNS vessels. However, the expression of GLUT-1 is highly sensitive to metabolic changes, and col. IV labels the vascular basement membrane (for instance, a vascular tract without endothelium may still appear as labeled by Col.IV). Hence, these are not the best options. To address this, authors could add immunostaining and new quantifications using a structural marker like PECAM-1 (and/or a fluorescent perfusion marker injected in the systemic circulation).

4- In vitro experiments (PLA) were done using HUVECs. This raises concerns about the generalizability of mechanisms across species. To address this, authors could use primary mouse CNS endothelial cells (ECs) isolated and cultured. Many protocols have been published to perform such experiments. Authors could then compare primary mouse CNS ECs from their control and mutant mice for qualitative and quantitative assessments.

5- While it may seem a minor point, the quality of the EM images in Figure 7k are below standards. Sample preparation is far from optimal and images not convincing. If authors wish to add EM to the study, they should revisit this part by performing new experiments with better tissue fixation and a deeper level of quantitative analysis.

6- Here, the focus is on the endothelium, however it is well known that developing CNS vessels closely interact with glial cells, particularly in the retina where astrocytes form a carpet guiding vascular growth. Authors could investigate whether lack of FLRT2 in endothelial cells affects glial coverage for instance (using GFAP in retinas and ALDH1L1 in the cerebral cortex).

Minor points:

1- Please define FLRT2 abbreviation when first mentioned in the Introduction.

2- In the Results section (first paragraph) please justify the use of Podocalyxin in Figure 1. Is it considered a vein marker? Why other known vein markers were not used in this study?

3- Differences measured in claudin-5 expression are small. Authors could measure protein levels and distribution of other EC junction for a more thorough description (e.g. ZO-1, occluding). A better description of both junction types (tight and adherent) would solidify the study.

Version 1:

Reviewer comments:

Reviewer #1

(Remarks to the Author)

The authors have satisfactorily addressed all my major concerns regarding the manuscript. Indeed the revisions have significantly improved the clarity, methodology, and overall quality of the paper. Given these improvements, I recommend that the manuscript be accepted for publication.

Reviewer #2

(Remarks to the Author)

The authors addressed all the questions very well using complementary techniques. For example, to demonstrate the formation of the FLRT2, NumB, and VE-cadherin complex, they performed expansion microscopy and immunoprecipitation from brain lysates. The new data, after addressing all reviewers' concerns, is of high quality and confirms the overall conclusions of the manuscript.

Reviewer #3

(Remarks to the Author)

Overall, the authors have done a good job at addressing major concerns. The manuscript has been largely improved.

I continue to think that TEM images (Fig 7m & Extended Data Fig. 8b) of tight junctions (TJs) are below quality standards (i.e., imperfect fixation, low quality of ultrastructure, darkness, blurriness, etc). Authors may want to consult few published examples (e.g., PMID 30014540, 38635399, 38951020, 33681208, 37587100) to get a better idea of what would be expected as a quality standard for impactful publications. Moreover, when TJs display short protrusions towards the lumen this is actually not considered abnormal. I suggest authors only keep selected TEM images of higher quality and only with those TJs displaying obvious abnormal features (e.g. vacuoles).

Response point-by-point to reviewers' comments

We thank the reviewers for the positive evaluation of our work and for the detailed analysis of our data. We are convinced that the new data, obtained following the reviewer's suggestions, have served to improve our manuscript. Please see below the point-by-point answers to the reviewer's requests.

Reviewer #1 (Remarks to the Author):

Peguera et al discusses the role of FLRT2 in regulating venous-mediated angiogenic expansion and CNS barrierogenesis. The study utilized genetically modified mice to examine the impact of FLRT2 on vascular development and barrier formation in the central nervous system. The results indicate that FLRT2 plays a significant role in venous zonation and angiogenesis, with preferential localization to the endothelial cell membrane of veins. Additionally, the study suggests that venous endothelial cells contribute to angiogenesis and the formation of the arterial plexus. The research provides insights into the molecular mechanisms underlying vascular development and barrier formation in the CNS, with potential implications for understanding and potentially treating related neurological conditions. The presented study is important and well-conceived but I have identified a few points that, in my opinion, need to be address to enhance its scientific rigor.

Major

Introduction:

1) The introduction exhibits a certain degree of generality and would benefit from improved coherence between paragraphs. Furthermore, the authors have relied on a few reviews to substantiate their descriptions, overlooking crucial recent studies on venous vasculature. Notably, the work by Coelho-Santos et al. PNAS 2021 (PMID: 34172585) demonstrates that veins play a pivotal role in establishing the capillary bed during postnatal brain development. Moreover, findings from Lee et al. Circulation. 2021 (PMID: 34474596) highlight the significance of veins in retinal development and pathophysiology. These important and recent studies should be incorporated into the manuscript instead of relying solely on general reviews.

RESPONSE: We appreciate the reviewer's suggestions. We have revised certain sections of the introduction and have now incorporated original papers as references to support the role of venous endothelial cells to vascular development: (Red-Horse et al., 2010), (Coelho-Santos et al., 2021), (Fruttiger, 2002; Xu et al., 2014), and (Lee et al., 2021).

2) *The reference 2 cited does not really describe the statement: 'quiescent venous endothelium can be reprogrammed to activate migratory behavior in response to angiogenic and mechanical cues'*

RESPONSE: We have changed the sentence to adjust it more precisely to the reference 2, now reference 3 (Xu et al., 2014).

3) *The hypothesis or the main objective of the study is unclear.*

RESPONSE: The main objective of the study was to elucidate whether FLRT2 plays a role in the physiological vascularization of the CNS and if yes which are the mechanisms of its action. We have added a sentence in the introduction: "*However, FLRT2 is also expressed in vessels of the central nervous system (CNS) under physiological conditions, suggesting a potential role in CNS vascularization that has yet to be explored*" (page 4).

Methods:

1) *The rationale for housing the animals individually in ventilated cages remains unspecified, this can impact their behavior. The pups were also alone?*

RESPONSE: The sentence: "*Mice were kept in individually ventilated cages*" specifically referred to the ventilation system used in the cages. Individually ventilated cages have their own ventilation systems to maintain airflow and control factors such as temperature and humidity within each cage. For clarity, in the methods section we changed the sentence to: "*Individually ventilated cages were used to host the mice....*" (page 24).

2) *In this study, fluorescence was quantified using Integrated Density - (Area x Background Mean Gray Value). The primary variable for fluorescence in integrated density is the area. It is suggested to employ Mean Gray Value instead of Integrated Density, as was done for cadaverine. Is there a specific explanation for this change in the approach for measuring fluorescence?*

RESPONSE: Thank you for pointing out these differences. The reason for using the Mean Gray Value to measure the average fluorescence intensity of the Cadaverine signal in the brains of control and mutant animals is due to the consistent ROI size. We consistently used a ROI of the same size, making the area irrelevant to the measurement. Furthermore, the nature of the Cadaverine signal is more homogeneous and is present throughout the entire tissue, making an average measurement the most appropriate.

However, to quantify the total amount of internalized VE-cadherin in the antibody feeding assay, FLRT2, FLAG, and Calpain1/2 expression in cultured endothelial cells, and Calpain2 expression in brain vasculature, we used Corrected Total Fluorescence (Integrated Density - (Area x Background Mean Gray Value)) since the use of just Mean Gray Value would not be accurate. To define the measured area or Region of Interest (ROI), we manually traced each cell. Given the heterogeneous sizes of the cells in these experiments, the Integrated Density is the most appropriate way to measure fluorescence intensity. The Mean Gray Value would not be suitable because it fails to account for differences in cell size; two objects with the same average intensity (Mean Gray Value) but different sizes would have different contents of the signal of interest, which would be lost if using Mean Gray Value alone. Moreover, Integrated Density captures the full range of pixel intensities within the ROI, including both very bright and very dim pixels, whereas the Mean Gray Value diminishes this information. We then corrected the quantified fluorescence by subtracting the background signal (measured outside the cell) to obtain a more accurate measurement.

Results:

1) One of the primary findings asserted by this study is the selective expression of FLRT2 in veins and capillaries. Although this can be truth for retina does not seems translated for brain vascular system. Looking carefully to the images I did not observe a clear reflection of this claim in the presented results. Specifically, in Figure 1b (brain), it is evident that Flrt2 is expressed in arteries (by the morphology) as well. The representation of capillaries was challenging to identify in the provided figures. It is crucial to distinguish arteries by labelling with alpha-smooth muscle actin and veins with EphB4 to accurately dissect this expression pattern.

RESPONSE: We thank the reviewer for their suggestion. To provide evidence of the preferential expression of FLRT2 in veins within the cortex, we characterized the arterio-venous expression of *Flrt2* mRNA in the cerebral cortex using fluorescent in situ hybridization.

We employed *Gkn3* as a specific marker for arteries and *Slc38a5* as a specific marker for veins. These vessel-type specific molecular markers were selected based on a scRNA-seq dataset analysis that identified vascular heterogeneity in brain vasculature (Vanlandewijck et al., 2018), which guided the design and synthesis of the corresponding probes. These probes were combined with Podocalyxin immunostaining (a general vessel marker) and the *Flrt2* in situ probe. Our results confirm that FLRT2 is specifically expressed in venous vessels in the cerebral cortex and is undetectable in arteries. This new data is presented in Extended Data Fig. 2e (page 8).

2) In the figure, Flrt2 is not only staining vessels; it is evidently labelling other cells in both retinas and the brain. Could the author provide explanations for this observation?

RESPONSE: We thank the reviewer for highlighting this oversight. FLRT2 expression has been documented in neurons of both the retina and cerebral cortex across several studies, such as (Li et al., 2021; Prigge et al., 2023; Visser et al., 2015; Yamagishi et al., 2011). In response to the reviewer's point, we have incorporated an explanatory sentence into the main text along with the corresponding references (page 5).

3) The Western blot presented in the manuscript to demonstrate FLRT2 EC-specific deletion was conducted using EC-lung isolates (Extended data Fig. 1e). Why was a similar analysis not performed in the retina or brain tissues?

RESPONSE: The *Cdh5creERT2* promoter is widely utilized in vascular biology. Its specificity and efficiency have been validated across various tissues, including the brain and retina, in numerous studies, including our own previous work (Benedito et al., 2009; Boulday et al., 2011; Boye et al., 2022; Pitulescu et al., 2010; Rama et al., 2015; Segarra et al., 2018; Wang et al., 2010). In our study, we used lungs as a highly vascularized tissue to effectively demonstrate the deletion of FLRT2 protein in the vasculature, given practical limitations in obtaining sufficient protein from isolated primary brain endothelial cells of mouse pups for Western blot analysis. Responding to the reviewer's request for additional controls, we included more data in Extended Data Fig. 1. Specifically, we isolated primary brain endothelial cells from control and FLRT2-induced knockout animals after tamoxifen injection from P1 to P3, plated them directly on coverslips, and conducted immunostaining for FLRT2 expression. Quantification of FLRT2 signal revealed significantly lower levels in endothelial cells from FLRT2 mutant animals compared to controls. Additionally, we extracted mRNA from these cells and performed RT-qPCR, confirming a marked reduction in *Flrt2* mRNA in FLRT2 vascular-

specific mutants compared to control littermates. These findings underscore the efficiency of the *Cdh5creRT2* promoter in inducing *FLRT2* deletion following excision of floxed exon 2, and are now included in Extended Data Fig. 1 (page 5).

4) *'The effectiveness of Tmx treatment was confirmed by evaluating Cre-mediated recombination, which resulted in tdTomato signal in approximately 80% of the vasculature in the retina of Cdh5(PAC)-CreERT2 mice crossed with the Rosa26-Stoplox/lox-tdTomato reporter line'. What about the brain?*

RESPONSE: As requested by the reviewer, we have included in Extended Data Fig. 1e, f the analysis of Cre-mediated recombination in cortical vessels using the *Rosa26-Stoplox/lox-tdTomato* reporter mouse line. Our findings indicate highly efficient Cre recombination in the brain vasculature following tamoxifen treatment, with approximately 96% of the vasculature showing expression of tdTomato (page 5).

1) *In Figure 2d of the Flrt2 knockout (KO), I have observed additional sprouts that were not included in the count. I am uncertain whether this discrepancy is due to issues with the selection of the representative image, but I can identify three more sprouts upon closer examination. The same discrepancy is noticed in Figure 2h; although the cropped images appear distinctly different, upon a comprehensive assessment and counting of sprouts over the representative figures, they seem comparable between the control (ctr) and knockout (KO) groups. This should be clarified.*

RESPONSE: We are sorry for the confusion. As suggested, we have selected representative images which are more adequate to reflect the quantification of angiogenic sprouts along the vasculature for the retina and cortex (new panels in Figure 2 d and h).

2) *Did the animals survive beyond postnatal day 8 (P8)? Were there any observed behavioral alterations, particularly in relation to developmental milestones and the age at which eyes opened?*

RESPONSE: The endothelial-specific *Flrt2* mouse mutants, which underwent tamoxifen-induced deletion from P1 to P3, survived beyond P8. Throughout this period, we conducted health monitoring until weaning and did not observe any gross morphological defects or

evident behavioral alterations. However, specific behavioral and cognitive analyses, although very interesting, were not performed as they were beyond the scope of this study. Additionally, we assessed body weight and brain size at P8, after FLRT2 deletion, to rule out potential growth defects associated with deficient vascularization, but no differences were observed between control and mutant animals. These findings are now presented in Extended Data Fig. 1k-n (page 6).

3) The authors did not measure blood flow, but can infer how blood flow is impacted by FLRT2 based on the EGR results?

RESPONSE: We analysed the polarization of endothelial cells at the leading edge of retinal sprouts by assessing the positioning of Golgi (GM130) relative to endothelial cell nuclei (ERG) (Fig. 4). According to (Barbacena et al., 2022), chemoattractant angiogenic cues play a crucial role in regulating the front-rear polarization of endothelial cells at the forefront of the retina. In contrast, flow-driven forces exert influence closer to the optic nerve, where they regulate endothelial cell polarization and vascular remodeling within the core of the vascular plexus (identifying the transition zone between sprouting and remodeling approximately at 300-400 μm from the sprouting front). Our measurements of cell polarization were confined to the angiogenic front, within 200 μm from the leading edge, suggesting minimal influence from blood flow on our results. This is supported by our findings in cell culture, which demonstrated similar results, indicating that FLRT2's effects on cell polarity at the angiogenic front are likely independent of blood flow. In this context, we have re-phrased the results to clarify better the interpretation of our results (page 14).

4) Since FLRT2 was deleted in EC but not specifically in veins, what outcome does the authors anticipate for the experiment?

RESPONSE: FLRT2 is primarily expressed in veins, but it is also found in capillaries and tip cells. Since venous endothelial cells (ECs) give rise to tip cells and arteries (Coelho-Santos et al., 2021; Lee et al., 2021; Xu et al., 2014), it is plausible that deleting FLRT2 exclusively in veins will replicate the overall effects of a general vascular deletion, particularly in the retina where vascularization occurs postnatally. In the cerebral cortex, the situation might differ. The cortex vascular network undergoes significant development prenatally and early postnatally. By the time FLRT2 is deleted in the venous compartment in early postnatal stages (P1-P4), many capillaries may have already differentiated. Thus, the impact of FLRT2 deletion in veins might not be as pronounced in the cortex as in the retina. The already differentiated capillaries

could maintain some degree of vascular integrity despite the absence of FLRT2 in veins, leading to less severe phenotypic outcomes in cortical vasculature.

5) Regarding the sentence: '*Perinatal endothelial FLRT2 deletion did not have an impact on the total number of arteries and veins present in the retina (Extended data Fig. 2a, b)*'. The tamoxifen-induced deletion of FLRT2 was consistently performed after birth. "Perinatal" typically refers to events occurring before birth. Was this a typographical error, or if not, could you please provide clarification?

RESPONSE: The tamoxifen-induced deletion was consistently performed from P1 to P3, as specified in each figure. We have replaced the term "perinatal" with "postnatal" to avoid any potential confusion (page 7). We thank the reviewer for highlighting this issue.

6) In this manuscript, experiments were conducted to knock down FLRT2 in EC during postnatal CNS development to elucidate the functional role of FLRT2 in vascular development and barrier formation. What potential outcomes could arise if genetic manipulation were employed to overexpress FLRT2 in EC?

RESPONSE: We thank the reviewer for this insightful question. We certainly think it would be interesting to overexpress FLRT2 and contrast the phenotypes with our inducible endothelial-specific deletion. However, there are several significant challenges to consider. Firstly, we would need to file and get approved all the animal experimentation protocols, which would require more time than is available for this revision process. Secondly, overexpressing FLRT2 specifically in the vasculature during the first week of postnatal development is crucial since FLRT2 is highly expressed in neurons. Without an inducible and specific overexpression system, it would be difficult to clearly discern the phenotypes arising from neuronal versus endothelial overexpression.

Nevertheless, we can speculate on the potential outcomes of such an experiment. We hypothesize that accelerated recycling of VE-cadherin could lead to its reduced stable expression at the cell membrane, resulting in a hyper-sprouting endothelium. This hypothesis is supported by previous studies showing that endothelial-specific deletion of VE-cadherin leads to increased angiogenic sprouting at the vascular front of mouse retinas (Gaengel et al., 2012). However, whether hyperactive vessels induced by FLRT2 overexpression would

generate functional angiogenic sprouts or develop vascular malformations remains to be investigated.

A parallel can be drawn to Dll4 mutants, where both gain and loss of protein expression can lead to aberrant angiogenesis. Blockade of Dll4 causes excessive but non-functional vasculature, impairing tumor growth (Noguera-Troise et al., 2006; Ridgway et al., 2006). Conversely, overexpression of Dll4 reduces tumor angiogenesis and halts tumor growth (Segarra et al., 2008; Trindade et al., 2017).

Therefore, while the potential outcomes of FLRT2 overexpression in vivo during physiological and pathological conditions are intriguing, they warrant further investigation beyond the scope of this study.

7) Was sex dimorphism considered in the analysis?

RESPONSE: We included animals of both sexes in our experiments and analyses. We verified that there were no apparent differences in gross morphology between male and female endothelial-specific FLRT2-deleted animals. Additionally, we measured the body weights of both sexes at P7-P8 and found no significant differences (see new panel in Extended Data Fig.1k).

It is crucial to note that our study was conducted during the first postnatal week, a period preceding the peak of sexual hormone activity (Bell, 2018). Studies have shown that canonical manifestations of sexual dimorphism, such as differences in the size of certain brain structures between males and females, typically become evident at P10-P17 (Qiu et al., 2018). Therefore, our analyses were performed outside the critical time window when sexual dimorphism typically exerts a profound impact. We acknowledge that sexual dimorphism is an important consideration that becomes more relevant in juvenile animals, particularly after three postnatal weeks when mice undergo puberty and reach sexual maturity.

8) It would be interesting to investigate the effects of pharmacological agents that modulate FLRT2 activity, specifically employing FLRT2 agonists, to assess their potential in rescuing the impacts on vascular development, particularly in angiogenesis and blood-brain barrier properties.

RESPONSE: We appreciate the comment of the reviewer. While this is indeed a fascinating question, we are unable to undertake pharmacological manipulations of FLRT2 signaling *in*

in vivo within the timeframe of this revision process. Such experiments would require obtaining extensive animal permits and establishing long-term treatment protocols.

One of the challenging aspects of conducting such pharmacological manipulations of FLRT2 activity is the limited availability of commercially applicable FLRT2 agonists or antagonists. Despite these challenges, our study has provided robust data demonstrating the interaction between VE-cadherin and FLRT2, and its effect on sprouting angiogenesis. We believe our findings lay a foundation for future investigations into the structural interaction between FLRT2 and VE-cadherin, potentially leading to the design of blocking peptides or small molecule inhibitors. Such inhibitors could potentially disrupt the binding of FLRT2 and VE-cadherin, offering novel therapeutic approaches for conditions involving abnormal angiogenesis, such as vascular malformations or tumor angiogenesis. However, while these prospects are exciting, they extend beyond the immediate scope of our current work. We hope that our findings will inspire further research efforts to explore the therapeutic potential of targeting FLRT2-mediated pathways in angiogenesis and vascular biology.

9) It was mentioned that single-cell RNA sequencing was conducted to profile expression within the vascular zone, but the results were not presented. This information is crucial for the study. Additionally, revealing the gene expression patterns of endothelial cells in the presence or absence of FLRT2 could offer valuable insights into the molecular pathways and signaling networks regulated by FLRT2 during vascular development in a more unbiased manner.

RESPONSE: Our manuscript focuses on characterizing the role of FLRT2 in vascular development primarily through experimental approaches such as immunostaining, genetic manipulation, and functional assays. The specific expression of FLRT2 in veins that we observed aligns with single-cell transcriptomic data generated by other labs (Sabbagh et al., 2018; Vanlandewijck et al., 2018) and our own ongoing transcriptional projects on postnatal brain endothelial cells. While these transcriptomic studies are valuable for confirming our findings, they are not essential for the current study. To avoid misunderstandings, we have omitted the mention of 'data not shown' and instead referenced relevant published sequencing data.

Regarding the gene expression patterns of endothelial cells in the presence or absence of FLRT2, we agree that such insights will significantly contribute to the broader understanding of FLRT2-mediated vascular biology. These findings will be crucial for guiding future research directions. However, they are beyond the scope of the current study.

10) When examining the BBB at an earlier time point (P4-P5), no alterations were observed. Could this be due to FLRT2-EC deletion not having occurred yet? The results for FLRT2-EC deletion were only presented at P7-P8, and it needs clarification whether, at the time of the analysis at P4-P5, FLRT2 was deleted in EC.

RESPONSE: We thank the reviewer for their suggestion. We have now analysed the efficiency of tamoxifen-induced deletion in *Flrt2*^{iAEC} mice at P5 by isolating primary brain endothelial cells and conducting RT-qPCR to measure the expression of *Flrt2* exon 2, which is flanked by loxP sites in *Flrt2*^{lox/lox} animals. Our results show that *Flrt2* gene expression is significantly knocked down by P5 (new panel in Extended Data Fig. 1h). These findings indicate that *Flrt2* is effectively deleted by this time point and that the preserved blood-brain barrier (BBB) observed at P5 is not due to residual FLRT2 expression. Instead, we hypothesize that the lack of BBB permeability defects at P5 (Extended Data Fig. 6k, l) is due to the delayed onset of the effects of VE-cadherin cleavage and degradation, and subsequent β -catenin nuclear translocation, that we observe only at P7-P8 (see Fig. 7). We have now depicted the temporal sequence of events resulting from FLRT2 deletion in the new Fig. 8a to better illustrate this progression.

11) Consistent with this, Extended Figure 3 demonstrates that FLRT2-EC deletion does not result in capillary density differences at P4-P5; however, there are fewer vein branches and reduced angiogenesis. I tried to comprehend how this discrepancy might arise and looking into the text for some theories, but the authors did not provide an explanation. I posit that this may be crucial to understand and could be linked to the fact that FLRT2 might not be deleted yet or is less expressed at these early time points. Could the issue potentially be related to cell death, or arrested proliferation rather than an angiogenesis problem?

RESPONSE: As mentioned above, we have now shown that the deletion of FLRT2 has already occurred by P5 (new panel in Extended Data Fig. 1h). At P5, our data indicated a defect in vessel sprouting (Extended Data Fig. 3j-m), although changes in vessel density and length were not yet apparent (Extended Data Fig. 3a-g). These changes became evident at P7-P8 (Fig. 1). To further investigate the underlying mechanisms, we examined whether changes in cell cycle or cell death contribute to the angiogenic phenotype. We found no significant differences in endothelial cell death at P5 in retinal and cortical vessels, as indicated by the apoptotic marker cleaved caspase-3 (new panels in Extended Data Fig. 4a-d). This is consistent with our observation of no changes in vessel regression at P7 (new panels in Extended Data Fig. 2a, b).

However, endothelial cells lacking FLRT2 exhibit increased levels of cell cycle arrest, as quantified by p21 expression (new panels in Extended Data Fig. 4e-h). These findings suggest that FLRT2 deletion impairs the expansion of the vascular plexus by disrupting cell cycle progression and angiogenic sprouting, with these effects becoming more pronounced at P7-P8. This temporal progression is detailed now in the manuscript (Fig. 8, page 20). The relation between FLRT2 and cell cycle arrest is also discussed (page 21).

12) Proximity ligation assays for FLRT2 and VE-cadherin were exclusively conducted in vitro, which is essential for initial screenings but does not capture the in vivo complexity, including factors like shear stress from blood flow. Therefore, these assays should be further validated in the retina and brain to account for the physiological conditions in vivo.

RESPONSE: We appreciate the suggestion to validate the PLA for FLRT2 and VE-cadherin *in vivo*. We performed a series of experiments to address this point. We performed PLA on primary brain endothelial cells directly isolated from control and FLRT2-deficient mutant mice. In cells derived from control animals, we observed a cluster signal between FLRT2 and VE-cadherin, validating our findings obtained in HUVECs (see Fig. 3a, b). In contrast, this interaction was significantly reduced in cells from *Flrt2*^{ΔEC} animals, confirming the specificity of the signal (new panels in Extended Data Fig. 5b, c). PLA in tissue sections was intensively tried but technically challenging. Instead, in parallel we tried alternative ways to prove the VE-cadherin/FLRT2 interaction *in vivo*. First, we performed immunoprecipitation on fresh whole brain cell lysates from wild-type animals using an anti-VE-cadherin antibody, ensuring the interactions occurred in the vasculature. We successfully identified FLRT2 pulled down by the VE-cadherin antibody (see new panel in Fig. 3c), indicating that these proteins form a complex *in vivo* in the tissue. Secondly, we leverage the expansion microscopy technique and conducted immunostaining for VE-cadherin, FLRT2, and Numb directly in the tissue. This method allowed us to detect complexes containing these proteins at the cell membrane of the brain vasculature (see new panel in Fig. 6a) as well as the retina vasculature (see new panels in Fig. 6b, c). These combined results confirm our *in vitro* findings and demonstrate the interaction between FLRT2 and VE-cadherin in tissue, thus addressing the physiological relevance of our observations.

13) VE-Cadherin is increased, but the BBB is leaking at P8. The authors referred to internalization and compensation. However, it appears that VE-cadherin, despite the increase, remains at the membrane, and no 'holes' were observed in the monolayer. Can the authors

provide more clarification on these observations? Additionally, if there are fewer new vessels and a high expression of VE-cadherin, were the VE-cadherin quantifications normalized to the number of vessels or the length of the vasculature?

RESPONSE: We appreciate the insightful questions regarding the observations on VE-cadherin expression and the integrity of the BBB. Our data indicate that in the absence of FLRT2, VE-cadherin is not effectively recycled and instead undergoes degradation (Fig. 3f, g). As a compensatory response, there is an upregulation of VE-cadherin transcription (Extended Data Fig. 5f), leading to increased VE-cadherin localization at the cell membrane (Fig. 3 f, h and Extended Data Fig. 5d, e). Importantly, we did not observe any 'holes' between cells either *in vitro* or *in vivo*. In addition, we detected a specific decrease in claudin-5 expression and an abnormal localization of claudin-5 in relation to VE-cadherin (new Fig. 7f-i), suggesting that the subcellular organization of adherens and tight junctions could be impaired (see also a more detailed answer in response to next question). To further investigate the barrier integrity and potential ultrastructural defects, we employed transmission electron microscopy (Fig. 7m, n and new panel in Extended Data Fig. 8b). This technique revealed a significant increase in the proportion of altered tight junctions in the mutant mice (see new quantification in Fig. 7n). Despite the increased VE-cadherin at the membrane, these tight junction alterations suggest a compromised barrier function, which could explain the observed size-selective BBB leakage at P8 (Fig. 7k-l). Regarding the morphology of VE-cadherin, we observed a smoother appearance in FLRT2 mutant animals, indicative of a less active vasculature (see Fig. 5). This morphological change is consistent with a reduced dynamic state of the endothelial junctions.

For the quantification of VE-cadherin, we normalized the measurements to the junctional length and ensured consistency by using the same region of interest (ROI) size in both brain and retina samples (see Methods section). The example ROIs provided in Fig. 5 demonstrate that these measurements are independent of vessel density, as the ROIs are small and specifically focused on the endothelial junctions.

14) The observed effect of FLRT2 on VE-cadherin may represent a compensatory mechanism related to other tight junctions, particularly the decreased expression of Claudin-5, as demonstrated in the study. While it is evident how FLRT2 modulates VE-cadherin, the involvement of FLRT2 in Claudin-5 expression is less clear. It would be interesting to

demonstrate the organization of Claudin-5 by immunohistochemistry and assess whether it correlates with the increased expression of VE-cadherin.

RESPONSE: We appreciate the suggestion to further investigate the relationship between FLRT2, VE-cadherin, and Claudin-5. Following this suggestion, we have conducted additional analyses to examine the correlation of VE-cadherin expression with Claudin-5 in cortical vessels. Our findings reveal a noteworthy phenotype: in control animals, both junctional proteins, VE-cadherin and Claudin-5, overlap and clearly delineate the cell membrane. However, in the absence of FLRT2 in the vasculature, Claudin-5 expression frequently appears split into two distinct bands that encase VE-cadherin. This aberrant organization was quantified by measuring the proportion of double-banded Claudin-5 per unit length of VE-cadherin, and we found a significant increase in this phenotype in vascular FLRT2 mutants (see new Fig. 7h, i).

To our knowledge, this unusual junctional pattern has not been reported before. We hypothesize that the disrupted turnover of VE-cadherin at the membrane in FLRT2-deficient cells impairs the proper positioning of tight junctions, thereby compromising the sealing of the blood-brain barrier (BBB). This split configuration may be an attempt to maintain junctional integrity in the absence of functional FLRT2, albeit not entirely successfully, as evidenced by the compromised BBB. These observations have certain reminiscence to the mislocalization of tight junctions upon JAM-A silencing and consequent E-cadherin overexpression in hepatocytes shown (Konopka et al., 2007) as indicated in our discussion (page 23).

In conclusion, our immunohistochemical analysis of Claudin-5 and the TEM images of vascular junctions support the hypothesis that FLRT2 plays a crucial role in maintaining the proper organization and expression of tight junction proteins. The observed correlation between increased VE-cadherin expression and the aberrant pattern of Claudin-5 underscores the complex interplay between these junctional components and highlights the importance of FLRT2 in vascular biology.

15) Considering the observed increase in BBB leakage over time following FLRT2-EC deletion, it prompts speculation that its expression is changing over time. Therefore, it is important to investigate the expression profile of FLRT2 during postnatal development in both the retina and the brain, with a particular focus on the cortical region, as the study was centered on this area.

RESPONSE: To address the potential modulation of FLRT2 expression during postnatal development, we investigated the expression profile of FLRT2 in primary brain endothelial cells isolated at various postnatal stages: P1, P5, P8, and P13-P15. These time points were chosen to align with the initiation of tamoxifen treatment (P1), the angiogenesis and BBB analysis (P5 and P8), and the period when capillary density stabilizes, and endothelial cell proliferation decreases (P13-P15) as described by (Harb et al., 2013) and reviewed by (Coelho-Santos and Shih, 2020).

Our findings indicate that there are no significant differences in FLRT2 expression levels across these postnatal stages (see Rebuttal letter Fig. 1). This suggests that FLRT2 expression in endothelial cells remains relatively constant during postnatal development in the brain. Therefore, the observed increase in BBB leakage over time following FLRT2-EC deletion is likely due to an aggravation of the phenotype rather than changes in FLRT2 expression levels.

Regarding the retina, due to the technical challenges associated with obtaining sufficient mRNA from isolated retinal endothelial cells from postnatal animals, a similar experiment could not be conducted. However, we do not expect significant differences in FLRT2 expression in this vascular bed compared to the brain. Our findings in brain endothelial cells suggest a consistent expression of FLRT2 during postnatal development, and we anticipate a similar pattern in the retinal vasculature.

16) How does FLRT2 regulate venous-mediated angiogenic expansion and CNS barrier genesis? What type of genes can be regulating?

RESPONSE: As shown in our manuscript, FLRT2 plays a critical role in venous-mediated angiogenic expansion and CNS barriergenesis by regulating the dynamic turnover of VE-cadherin at the endothelial cell membrane. This process is crucial during sprouting angiogenesis, where adherens junctions must disassemble to form new sprouts and subsequently reassemble to allow vascular growth. Our data show that FLRT2 binds to VE-cadherin and promotes its recycling, thereby contributing to the efficient turnover of VE-cadherin (Fig. 3 and Extended Data Fig. 5).

FLRT2 also regulates the expression of Calpains, which are enzymes that cleave the cytoplasmic tail of VE-cadherin, leading to its lysosomal degradation (Su and Kowalczyk, 2017). Additionally, FLRT2 interacts with the endocytic adaptor Numb (Fig. 4), which facilitates the endocytic turnover of cadherins (Miao et al., 2019; Sato et al., 2011). Although

we have not yet identified the specific Rab GTPases involved in FLRT2-mediated VE-cadherin recycling, their role in directing protein cargo in various intracellular vesicular compartments is well-documented (Stenmark, 2009), and therefore Rab GTPases family members could also contribute to VE-cadherin disrupted recycling.

Furthermore, FLRT2 deletion increases the transcription of the VE-cadherin gene as a compensatory response to its degradation (Extended Data Fig. 5). At the cellular level, FLRT2 influences endothelial cell sprouting, cell polarization, and cell division, but it does not affect cell death or vessel regression (Fig. 2, Fig. 4, and Extended Data Figs. 2, 3 and 4).

In terms of CNS barrierogenesis, FLRT2 regulates the tight junction protein Claudin-5 at both transcriptional and translational levels (Fig. 7 and Extended Data Fig. 6). This regulation is likely mediated by VE-cadherin cleavage and the subsequent nuclear translocation of β -catenin (Gavard and Gutkind, 2008; Taddei et al., 2008). The effects on tight junctions appear to be specific to claudin-5, as other tight junction proteins such as JAM-A and ZO-1, as well as other components of the neurovascular unit, remain unaltered (new Extended Data Figs. 6 and 7). This specificity correlates with the observed size-selective permeability of the blood-brain barrier for small molecules in FLRT2 mutants (Fig. 7j-1 and new Extended Data Fig. 6i, j).

17) Cadaverine (950 Da) is relatively small, and it was allowed to circulate for an excessive 90 minutes before fixation. The BBB exhibits permeability to small dyes until postnatal day 12 (P12), and according to previous research (Coelho-Santos et al., PNAS 2021 - PMID: 34172585), a mere 10 minutes after circulation is sufficient for a 10 kDa molecule to exit the circulation. Therefore, the sentence 'indicating that the defective sprouting phenotype observed at this time point is not a consequence of a disrupted barrier' might not be accurate. It could be a conclusion drawn from the experimental time frame used, as all the main vascular angiogenesis processes were already in place. This should be considered.

RESPONSE: We appreciate the insightful comments and understand the concerns regarding the permeability of the blood-brain barrier (BBB) to small molecules like cadaverine, particularly in the context of the experimental timeframe. However, it is important to highlight several key points to address these concerns.

The study by Coelho-Santos et al. (Coelho-Santos et al., 2021) employs 2-photon imaging through a cranial window to assess tracer extravasation in the postnatal neocortex shortly after injection. This technique allows for the immediate detection of tracer leakage from the

vasculature. Our approach, on the other hand, aims to observe tracer accumulation in the parenchyma after a prolonged circulation period, which provides a comprehensive view of BBB permeability over time. This methodology has been successfully used in several other publications, including (Armulik et al., 2010; Boye et al., 2022; Segarra et al., 2018; Yanagida et al., 2017). Therefore, our approach of allowing cadaverine to circulate for 90 minutes before fixation is designed to detect cumulative leakage into the brain parenchyma. This method provides a robust measure of overall BBB integrity over a longer period, rather than capturing immediate leakage events.

We are comparing tracer permeability between control and *Flrt2*^{ΔEC} mutant animals. Even if the BBB exhibits higher permeability at P5 compared to P8, our primary focus is on the relative differences between genotypes at each time point. Our results show no significant difference in cadaverine leakage at P5 between control and mutant animals, suggesting that the defective sprouting phenotype observed at this stage is not due to a disrupted barrier. In contrast, we observe increased cadaverine leakage in *Flrt2*^{ΔEC} brains compared to controls at P7-P8, indicating a progressive BBB impairment linked to FLRT2 deletion.

In conclusion, while the permeability of the BBB to small molecules like cadaverine is indeed a consideration, our methodology and comparative approach across genotypes and time points allow us to discern the specific effects of FLRT2 deletion on vascular and barrier integrity.

18) Pericytes are essential cells for the stability of the vascular system and work in synergy during angiogenesis. Were any alterations after FLRT2 deletion observed in these cells?

RESPONSE: We appreciate the inquiry regarding the role of pericytes in the context of FLRT2 deletion. To address this, we have investigated the coverage of the vasculature by pericytes in *Flrt2*^{ΔEC} animals. Our analysis focused on the presence of PDGFRβ⁺ cells in contact with the vasculature. The results indicate that there is no significant difference in pericyte coverage between control and FLRT2 mutant animals. This suggests that the vascular and barrier defects observed in our study are intrinsic to the endothelial cells and do not stem from alterations in pericyte coverage. These findings have been included in our revised manuscript (new Extended Data Fig. 7a, b) (page 19).

19) After thoroughly reviewing the results and scrutinizing the data, it appears that they do not substantiate the title 'Vascular FLRT2 regulates venous-mediated angiogenic expansion and CNS Barrier genesis.' It would be prudent to consider substituting "CNS" with "retina" in the

title. Given the presented data, it is challenging to draw conclusive parallels between the impact of FLRT2 on vascular development and barrierogenesis in the brain through veins. Although recent papers have highlighted veins as sources of angiogenesis, the labeling of FLRT2 appears to extend across other vascular cells and non-vascular cells in the cortical region.

RESPONSE: We thank the reviewer for their valuable feedback. With all respect, we disagree with the suggestion to limit the title to "retina." Our data, which align with published single-cell RNA sequencing (scRNAseq) data, clearly demonstrate that FLRT2 has a distinct venous zonation in both territories of the CNS, including the retina and cerebral cortex. Therefore, we believe that our title accurately reflects the broader implications of our findings across the CNS and does not need to be limited to the retina. Furthermore, our title does not imply that veins are solely responsible for CNS barrierogenesis. In the sentence title "Vascular FLRT2 regulates venous-mediated angiogenic expansion and CNS barrierogenesis," the subject is "Vascular FLRT2." This subject refers to FLRT2 specifically within the vascular context, implying its role or influence on venous-mediated angiogenic expansion and CNS barrierogenesis. The title suggests that FLRT2, when expressed within the vascular system, plays a regulatory role in these processes. This distinction is crucial and supports the appropriateness of our current title.

Regarding the expression of FLRT2 in other cell types within the cortex, it is indeed irrelevant to the endothelial-specific deletion of FLRT2 that we focus on in this study. The scope of our research and the specificity of our genetic manipulations ensure that our conclusions pertain directly to the endothelial functions of FLRT2.

Thus, we strongly believe that the title 'Vascular FLRT2 regulates venous-mediated angiogenic expansion and CNS barrierogenesis' accurately represents the essence and breadth of our research findings.

20) The references 2,21 and 22 used do not reflect the statements and should be substituted by (PMID: 34172585) and (PMID: 34474596): Veins have been identified as the primary source of ECs for angiogenesis, and recent studies have demonstrated that venous ECs give rise to the endothelial tip cells, which are located at the leading edge of retinal vasculature and subsequently migrate into the arterial plexus 2,21-23.

RESPONSE: We appreciate the suggestions regarding the references. We have now replaced previous references 21 and 22 with the suggested references (PMID: 34172585) and (PMID:

34474596) to accurately reflect the current literature. We have retained reference 2 (now reference 3 in our revised manuscript, (Xu et al., 2014)) because it provided foundational insights into the contribution of vein-derived tip cells to the formation of the arterial retinal bed. This study remains relevant in establishing the context for our discussion on venous endothelial cells and their role in angiogenesis.

21) *This sentence needs to be clearer about the type of vasculature; it begins with "mutant vessels compared to control capillaries." Are the authors comparing capillaries with capillaries, or is the comparison inclusive of all types of vasculature (arteries, veins, and capillary bed)?*

RESPONSE: In our study, we specifically analyzed capillaries within a defined size range of 5.5-8.5 μm diameter, as outlined in the methods section. To address the reviewer's concern, we have revised the sentence to explicitly state that we compared control and mutant vessels within this capillary size range (page 19). We thank the reviewer for the suggestion, which has helped us improve the clarity of our manuscript.

22) *In Figure 7l, the term "abnormal junctions" is used; could the authors provide clarification regarding the criteria used for defining these junctions?*

RESPONSE: We have provided a more specific quantification of the abnormal tight junctions in FLRT2 vascular mutants. We calculated the percentage of normal tight junctions versus abnormal junctions, differentiating between open, undefined or tortuous tight junctions. We also observed a significant increase of tight junctions close to a large vacuole (> 100 nm diameter) in the mutant mice. The new quantification is now found in Fig. 7n. We have included additional clarification in the figure legend outlining the specific morphological criteria used to classify junctions as abnormal. Additionally, to provide further clarity, we have included several example images in new Extended Data Figure 8b, demonstrating various types of junctional defects observed in *Flrt2*^{IAEC} mice. Furthermore, we have revised the methods section to explicitly define the criteria used for identifying abnormal junctions (page 38).

Discussion:

Jumping to conclusions without robust supporting results should be avoided. It is crucial to meticulously discuss the actual data presented in the paper. The author should delve further

into the spatiotemporal events identified in this importante study, emphasising their significance for capillary formation and maturation, rather than engaging in speculation about unexplored diseases in this story. Are there any disorders where FLRT2 plays a particular role? If so, discussing those instances could be important.

RESPONSE: We appreciate the reviewer's feedback and acknowledge the importance of basing conclusions on robust data presented in our study. In response to the suggestion, we have revised the discussion section to highlight the spatiotemporal dynamics identified in our investigation, emphasizing their significance in capillary formation and maturation following FLRT2 deletion. Furthermore, we have incorporated specific references to relevant pathologies where FLRT2 has been shown to play a significant role, thereby contextualizing our study within broader clinical implications.

Minor

1) Introduction:

a. CNS was never described before.

RESPONSE: We apologize for the oversight. We have now clarified that CNS stands for Central Nervous System in the manuscript.

b. The authors refered 'endothelial subtypes' should state what subtypes.

RESPONSE: We have revised the manuscript to specify the different endothelial subtypes derived from vein endothelial cells. Additionally, we have cited the original publication (Lee et al., 2021) that describes these subtypes in detail.

2) Methods:

a. When the authors refer to "1024 x 1024-pixel size maximal intensity projection images (with a minimum of 5 images per animal or condition)," clarification is needed regarding whether this pertains to depth, such as a z-stack, or specific Regions of Interest (ROIs). The methods section should provide explicit details.

RESPONSE: We appreciate the feedback. In the methods section, we have revised the sentence to provide clearer clarification: "For image quantification, at least 5 maximal intensity projection images of size 1024 x 1024 pixels per animal or condition were used, unless

specified otherwise.” We trust that this revised phrasing clarifies the intended approach for image acquisition and analysis.

3) Results

a. In the figures, what is black should be white or grey – just an aesthetic note.

RESPONSE: We have adjusted the colour scheme in the appropriate figures by replacing black with grey where necessary to enhance visual clarity.

b. It would be highly beneficial for readers to have a schematic depicting the timeline of events, facilitating a clearer understanding of the complexity and findings of the work.

RESPONSE: Following the reviewer’s valuable suggestion, we have included a schematic representation of the vascular phenotypes observed upon FLRT2 deletion during postnatal development, along with a diagram illustrating the molecular mechanism uncovered (see new Fig. 8). We appreciate this insightful recommendation.

If possible, the manuscript would benefit from some English editing to achieve a more fluid and harmonious text for improved readability.

RESPONSE: The manuscript has undergone thorough proofreading by a native English speaker.

Throughout the text, some acronyms are explained while others are not. "Blood-brain barrier" and 'endothelial cells' are sometimes abbreviated and sometimes written in full. Please standardize the usage for consistency.

RESPONSE: We have maintained consistent usage of acronyms throughout the manuscript. Specifically, we have used BBB for blood-brain barrier, EC for endothelial cell (or ECs for plural), Tmx for 4OH-hydroxytamoxifen, and CNS for central nervous system consistently to ensure clarity and coherence.

Reviewer #2 (Remarks to the Author):

This interesting manuscript from Peguera and coworkers characterizes the function of FLRT2 in the vessels of the retina and the cerebral cortex. Previous studies from the same group had shown that FLRT3 interacts with Unc5 receptors to regulate retinal vascularization (Seiradake et al., 2014). However, little is known about the functions of other FLRT members in this process or about potential novel interacting partners. In this study, the authors extend this concept to FLRT2, showing that it forms a complex with Numb and VE-cadherin, modulating the recycling of VE-cadherin. This contributes to the modulation of adherens junctions in both the retina and cerebral cortex, and tight junctions that are crucial to blood-brain barrier development.

This manuscript is well-organized, reporting interesting results with an appropriate experimental design and high-quality figures. The data presented in Figures 1, 2, and 4, which show reduced overall retinal vascularization and blood vessel density in the cerebral cortex by P7-P8, reduced branching originating from veins but not arteries, reduced angiogenic sprouts, and altered adherens junctions, are very clear and convincing. The evidence supporting a model in which FLRT2 forms a complex with Numb and VE-cadherin, thereby regulating the dynamics of the latter, relies mainly on in vitro assays using HUVEC and FLRT2 siRNA (as shown in Figure 3 and most of Figure 4; although it is interesting the colocalization between FLRT2 and VE-cadherin in a cerebral cortex vessel in Fig. 6). The FLRT2-dependent regulation of junctional proteins, such as Claudin 5, is primarily demonstrated in bEnd.3 cells using FLRT2 siRNA (Figure 7). These two mechanisms are very intriguing; however, without further in vivo experiments to complement the in vitro work, questions are raised regarding their role as the major contributors to the interesting phenotype reported. For example, can the FLRT2-Numb-VE-cadherin complex be detected in vivo? Could Unc5 receptors be participating to some extent in this phenotype? Indeed, in a previous study, the authors showed that FLRT3 interacts with Unc5 receptors present in blood vessels, thereby regulating retinal vascularization

Below are some questions that should be addressed to further solidify the main conclusions of this study.

1. The authors state that FLRT2 forms a complex with Numb and VE-cadherin; however, most of the data are from in vitro studies. The authors should demonstrate that this complex forms in vivo. For example, they could perform pull-down assays from retina or cerebral cortex lysates using antibodies or recombinant proteins as bait. Another option could be to use expansion microscopy on tissue samples, labeling all three proteins and checking that they are present at the cell surface.

Regarding this, even though the colocalization between FLRT2 and VE-cadherin in 3D reconstructed vessels from the cortex is interesting (Fig. 6), I find that the VE-cadherin staining is very strong, covering almost the entire surface. Would it be possible to obtain images of the three proteins, similar to Figure 4b for FLRT2 and Numb, but with higher resolution?

RESPONSE: We acknowledge the concern regarding the need for *in vivo* demonstration of the FLRT2, Numb, and VE-cadherin complex. Addressing this, we have conducted additional experiments to substantiate our findings.

We first undertook the challenge of performing expansion microscopy on brain tissue to visualize the colocalization of VE-cadherin, FLRT2, and Numb in a multiplexed manner. After troubleshooting, we succeeded in obtaining images that show the three proteins colocalized along the cell junctions of the brain vasculature. These results, now presented in the new Fig. 6a, confirm the *in vivo* tripartite complex of VE-cadherin, FLRT2, and Numb at the vascular wall, supporting the role of FLRT2 as an interacting partner of both VE-cadherin and Numb. Additionally, we have included higher magnification images and separate panels for each channel to ensure clarity and sharpness. Furthermore, we have also visualized the colocalization of these proteins in the retinal vasculature using expansion microscopy (new panels in Fig. 6b, c).

In addition to the experiments using expansion microscopy, we also performed immunoprecipitation from brain lysates using an antibody against VE-cadherin. The results demonstrated that VE-cadherin co-immunoprecipitates with FLRT2, indicating their binding

in vivo (new panel in Fig. 3c). Furthermore, Proximity Ligation Assays (PLA) for VE-cadherin and FLRT2 were also confirmed in primary brain endothelial cells isolated from control and *Flrt2*^{ΔEC} animals. The interaction between VE-cadherin and FLRT2 was significantly reduced in FLRT2-knockout endothelial cells, corroborating the specificity of the PLA signal (new panels in Extended Data Fig. 5b, c).

Moreover, we have now analysed whether the expression of Numb is also regulated in primary mouse brain endothelial cells isolated from control and FLRT2 mutant animals. Similarly to our observations *in vitro* (Fig. 4c), Numb mRNA expression levels in primary brain ECs are downregulated in FLRT2-depleted endothelium (new panel in Fig. 4f).

All these additional experiments collectively support the formation of a complex involving VE-cadherin, FLRT2, and Numb *in vivo*, thereby reinforcing our conclusions. We appreciate the reviewer's suggestions, which have significantly strengthened our manuscript.

2. The authors should perform a rescue experiment in their vascular-specific FLRT2 mutant to support their findings. For example, would only the intracellular domain of FLRT2 attached to the cell surface of vessels be enough to rescue the phenotype, or is the full-length version required? This could indicate whether a FLRT2-ligand is necessary to regulate this process. For example, the authors mention in the discussion that the FLRT3-Rnd1 complex regulates cadherin endocytosis. It has also been shown that Unc5B binds to this complex in cis, thereby regulating cell adhesion (Karaulanov, 2009). Would it be possible to use a mutant version of FLRT2 that does not bind to Unc5 and see whether it can rescue any phenotype?

RESPONSE: Although performing a rescue experiment in our *in vivo* system is certainly a highly interesting and conclusive approach, it is currently not feasible due to several significant challenges. Reexpressing FLRT2 in its full-length or truncated forms in mice requires, first, extensive applications for experimental animal permits, and second, the development of a method to specifically deliver FLRT2 to endothelial cells *in vivo*. These complex and time-consuming processes cannot be completed within the limited time available for revisions.

Regarding the role of Unc5B, its expression is concentrated in arteries and capillaries (Boye et al., 2022; Vanlandewijck et al., 2018), while FLRT2 shows a venous zonation as demonstrated by our data and available single-cell sequencing data (Sabbagh et al., 2018; Vanlandewijck et al., 2018). Fluorescence *in situ* hybridization combining probes for *Flrt2* and *Unc5b* showed

no co-expression of the two molecules in the same vessel type (Rebuttal letter Fig. 2), making it unlikely that Unc5B participates in FLRT2-controlled venous-mediated angiogenesis. Furthermore, Unc5B was found to regulate the BBB in adult vasculature, but endothelial-specific Unc5B deletion did not influence VE-cadherin expression (Boye et al., 2022), suggesting that FLRT2 and Unc5B do not have overlapping roles. FLRT3, identified as the signaling partner of Unc5B and shown to interact with c-cadherin during *Xenopus* gastrulation (Karaulanov et al., 2009), is expressed at lower levels in the mouse postnatal vasculature (see <https://brainrnaseq.org/>). Nonetheless, it is intriguing that FLRT3 was also shown to participate in c-cadherin endocytosis in *Xenopus*, similar to how FLRT2 regulates VE-cadherin trafficking in mouse vasculature, suggesting conserved mechanisms across species.

3. The finding that calpain 1 levels increase in FLRT2 knock-down cells is very interesting. Given that the authors link this protein with both the 95 kDa VE-cadherin being targeted for degradation, and β -catenin nuclear transport and the tightness of the BBB, they should show that the levels of this protein also increase in vivo. One option could be to perform a western blot on FLRT2 conditional mutants, or to use immunofluorescence as demonstrated in Fig. 3K, to avoid any underestimation of the effect.

RESPONSE: Following the suggestion from the reviewer, we investigated the expression levels of Calpain in our mouse models. Calpains are calcium-dependent cysteine proteases with several isoforms ubiquitously expressed throughout the organism (Zhang et al., 2017). Notably, Calpain-1 and Calpain-2 are the only isoforms detected in endothelial cells (Fujitani et al., 1997). Given their broad expression across various cell types in the CNS (Vanlandewijck et al., 2018; Zhang et al., 2014), performing a western blot from whole brain lysates was unsuitable due to potential signal dilution from non-endothelial sources. To address this, we decided to focus on Calpain-2, which is more abundant in brain endothelial cells compared to Calpain-1 (Vanlandewijck et al., 2018; Zhang et al., 2014). We isolated primary brain endothelial cells from both control and FLRT2 mutant animals and analyzed Calpain-2 expression using immunofluorescence after plating the cells. Our results showed a significant increase in Calpain-2 protein levels in cells derived from *Flrt2* ^{Δ E^C} animals, thereby corroborating our initial findings in HUVECs. Moreover, we extended our analysis to the tissue. Immunofluorescence analysis of brain sections revealed a marked increase in Calpain-2 expression in the FLRT2-depleted vasculature. These results very nicely substantiate our initial data showing that FLRT2 regulates VE-cadherin degradation by enhancing its Calpain-

mediated cleavage. We thank the reviewer for their suggestion and have included the new data in Fig. 3k-n (page 12).

4. In line with previous question, Is the 95 kDa VE-cadherin form also increased in the FLRT2-mutant mice? Regarding Claudin-5 (Cldn5), the authors show a slight mRNA reduction in the total brain of mutant mice. What about the protein levels? Perhaps performing immunofluorescence could reveal a stronger effect than analyzing mRNA from the entire brain, as the latter approach may dilute the effect

RESPONSE: We appreciate these suggestions. To address the inquiry regarding Claudin-5 and the cleaved form of VE-cadherin, we have conducted additional analyses. For Claudin-5, we used western blotting on total brain lysates, leveraging the fact that Claudin-5 expression is endothelial-specific. Our results indicated a significant reduction in Claudin-5 protein levels in *Flrt2*^{iAEC} brains, consistent with the observed decrease in mRNA expression. Furthermore, immunofluorescence staining for Claudin-5 and VE-cadherin in the brain vasculature uncovered a notable phenotype. In control animals, Claudin-5 and VE-cadherin co-localize at the junctions of cerebral cortex vessels. However, in the absence of FLRT2, Claudin-5 exhibited an altered morphology, often appearing split and encasing VE-cadherin at the cell junctions. These findings reinforce the necessity of FLRT2 for the proper formation of adherens and tight junctions in the CNS vasculature. The new data are presented in Fig. 7.

As for the 95 kDa cleaved form of VE-cadherin, detection of the cleaved fragment by Western blots was difficult and remained below the limit of detection in whole brain lysates as well as in primary endothelial cells isolated from the brains of *Flrt2*^{iAEC} and control mice. Nonetheless, we have demonstrated that Calpain, the proteolytic enzyme responsible for cleaving VE-cadherin, is increased in endothelial FLRT2-depleted animals (as detailed in our previous response).

5. A previous study has shown that FLRT2 is required for the proper integrity of the basement membrane during heart development (Müller P et al., 2011). Is the basement membrane of FLRT2-mutant blood vessels affected? Maybe comparing control and mutant sections stained with collagen IV could help to address this question.

RESPONSE: We thank the reviewer for this suggestion. Collagen IV is indeed a critical component of the basement membrane, essential for vascular scaffolding (Poschl et al., 2004).

To investigate whether the basement membrane of FLRT2-mutant blood vessels is affected, we conducted a comparative analysis of Collagen IV coverage in the vasculature of both control and mutant animals. Our measurements revealed that Collagen IV levels remained unchanged between control and FLRT2-mutant animals. These new findings have been included in the manuscript in Extended Data Fig. 7e, f. Moreover, we found no association between Collagen IV expression and blood-brain barrier (BBB) leakage. Specifically, as shown in Fig. 7l, cadaverine extravasation occurred regardless of the presence of Collagen IV in the vasculature, indicating that BBB integrity issues were independent of Collagen IV levels. Additionally, we examined vessel regression by analyzing the presence of empty sleeves of Collagen IV, which are devoid of endothelial cells, in retinal vessels. The occurrence of these empty sleeves was similar in both control and mutant samples (new panels in Extended Data Fig. 2a, b), suggesting that vascular growth and remodeling processes were accompanied by Collagen IV synthesis in both groups.

In conclusion, our data indicate that the absence of FLRT2 does not significantly affect Collagen IV deposition or the integrity of the basement membrane in blood vessels. Therefore, while FLRT2 is crucial for basement membrane integrity during heart development, its deletion does not appear to impact the deposition of Collagen IV in the vasculature during CNS development.

6. During heart development, it has been shown that other members, like FLRT3, can compensate for the loss of FLRT2. Is FLRT3 expressed in these blood vessels, and if so, are its levels affected by FLRT2 knockdown?

RESPONSE: We thank the reviewer for this interesting point. To investigate whether FLRT3 could compensate for the loss of FLRT2, we analyzed the transcriptional expression of *Flrt3* in primary mouse brain endothelial cells isolated from both control and endothelial-specific *Flrt2* mutant mice. These analyses were conducted on the same samples used to confirm *Flrt2* deletion. Our results showed that *Flrt3* expression did not significantly change following the abrogation of *Flrt2* expression, indicating that FLRT3 does not compensate for the loss of FLRT2 (new panel in Extended Data Fig. 1h).

Furthermore, it's important to note that FLRT2 is expressed at much higher levels in the brain postnatal vasculature compared to FLRT1 or FLRT3 (see <https://brainrnaseq.org/>). This substantial difference in expression levels further supports the conclusion that FLRT3 cannot fulfil the functional role of FLRT2 in CNS vascular development and maintenance.

Minor questions:

1. Figure 1b displays FLRT2 immunofluorescence in a blood vessel of the cerebral cortex. Is FLRT2 expressed homogeneously across the entire blood vessel throughout the cortical layers? And in rostro-caudal sections? Considering that both FLRTs (including FLRT2) and several of their ligands are specifically expressed in different cortical layers, it would be interesting to determine whether blood vessels show a homogeneous distribution of this receptor, or if there are enrichments specific to certain cortical layers.

RESPONSE: Unfortunately, the high expression levels of FLRT2 in neurons, as demonstrated by (Yamagishi et al., 2011) and depicted in Fig. 1a, b, posed challenges for assessing the comparative expression levels of FLRT2 specifically in different vessels across various cortical layers in tissue sections.

2. Figures 3 and 4 show PLA experiments indicating close proximity between FLRT2-VE-cadherin, Numb-FLRT2, and Numb-VE-cadherin. What percentage of the puncta is present at the cell surface? From the images, it appears that a significant proportion is located in the cytosol. Labeling the plasma membrane of the cells with a specific dye could be useful in measuring this.

RESPONSE: To address this question, we have conducted additional PLA assays using Pecam1 as an endothelial cell membrane marker. Our findings indicate that most PLA puncta are localized along the cell junctions, particularly evident in control cells. However, we also observed a fraction of the signal in the cytosolic compartment, suggesting potential internalization of FLRT2 and VE-cadherin complexes. Given Numb's role in endocytic processes, it is plausible that these complexes are present proximal to the membrane as well as within the cytosol. This data has been included in the manuscript (new panels in Extended data Fig. 5b, c).

3. In the discussion there is a small typo. "GTPase Rdn1" should be "GTPase Rnd1".

RESPONSE: We have corrected the typo. Many thanks for spotting it.

Reviewer #3 (Remarks to the Author):

In this new manuscript by Peguera et al., authors describe the role of protein FLRT2 in the regulation of murine CNS angiogenesis and barrier genesis, with a focus on the venous origin of vascular expansion and maturation during the first postnatal week in the retina and cerebral cortex (up to P8).

The manuscript is very well written, data clearly presented, and the model is interesting. While this study overall is well performed, important concerns remain about the lack of depth in the descriptive and mechanistic aspects, as well as the lack of functional relevance of the pathway beyond the first postnatal week, thus hindering the enthusiasm of the reader. To bring this study to the next level, authors may wish to consider the following points:

Major points:

*1- An important weakness of the manuscript in its present form is the lack of functional relevance to older ages (adulthood). To address this, authors could consider measuring CNS vascular structure and function in adult *Flrt2*^{iDEC} mice. Even more interesting would be to measure neuronal function at P8 and in adults. What does the lack of FLRT2 in vessels cause in terms of cortical lamination? Retinal function? Mouse behaviour? Answers to the questions would bring a unique dimension to this study.*

RESPONSE: We appreciate the reviewer's insightful suggestion regarding extending our study to investigate the role of FLRT2 in adult vascular and neuronal functions. However, conducting experiments in adult *Flrt2*^{iAEC} mice pose several challenges. These experiments are laborious and require multiple applications for conducting animal experimentation, including the need for long-term animal permits and extensive resources.

Developmental angiogenesis is a key component of organ formation and homeostasis, particularly in the CNS. Understanding how endothelial cells dynamically change to promote vascular growth during organ development is essential to discern the mechanisms of physiological vascularization (reviewed in (Potente et al., 2011; Spurgin and Cleaver, 2024)).

Moreover, while signalling pathways guiding developmental angiogenesis become dormant in healthy adult vasculature, they can be reactivated in vascular-dependent pathological scenarios such as tumors, injuries, or vascular malformations (recently reviewed in (Walchli et al., 2023)). Therefore, elucidating how vessels grow and remodel in the CNS is crucial for exploring avenues to manipulate vascularization and gain insights for therapeutic interventions in diseases with vascular and neurovascular components (see reviews (Carmeliet and Jain, 2011; Eelen et al., 2020; Quaegebeur et al., 2011)).

Our current study focuses on elucidating the function of FLRT2 during early postnatal development, a critical period for vascular maturation. By focusing on this developmental window, we aim to strategically explore processes that could potentially affect adult function and behaviour, both physiologically and pathologically. Investigating the consequences of FLRT2 deletion in neuronal function and adulthood would indeed provide a unique dimension to our study. However, such studies require a separate and comprehensive research effort beyond the scope of this current work. For the present study, our goal is to provide a foundational understanding of the role of FLRT2 during early vascular development, laying the groundwork for future exploration of its broader implications in other systems and in adulthood.

2- The effectiveness of Tmx treatment was assessed using Cdh5-CreERT2;Rosa26StopFlox-tdTomato mice. This raises concerns, and a preferable way would be to use a reporter that turns ON upon excision of the floxed Flrt2 allele. I do not think the approach chosen by authors is reliable in assessing the efficiency of Cre to remove Flrt2 from Cdh5-expressing cells. It is more testing Cre activity per se (which is quite different from assessing its action on the target gene).

RESPONSE: While we acknowledge that the expression of the tdTomato reporter does not directly measure the efficiency of Cre-mediated gene excision, it serves to indicate the specificity and extent of Cre promoter activity in Cdh5-expressing cells.

To address concerns about the reliability of our approach, we have implemented multiple complementary methods to verify the efficiency of FLRT2 deletion following tamoxifen induction. Firstly, we conducted western blot analysis, which confirmed the abrogation of FLRT2 expression in primary lung endothelial cells derived from control and *Flrt2*^{ΔEC} mice. Secondly, we measured the reduced expression of *Flrt2* transcript specifically corresponding to exon 2, which is flanked by loxP sites, in primary mouse brain endothelial cells isolated

from *Flrt2*^{ΔEC} mutants and littermate controls. Thirdly, we evaluated the diminished expression of FLRT2 protein through immunostaining in primary mouse brain endothelial cells derived from *Flrt2*^{ΔEC} mice compared to littermate controls.

These additional approaches collectively support the effectiveness of our tamoxifen treatment protocol and demonstrate the efficient deletion of FLRT2 mediated by *Cdh5-CreERT2* in endothelial cells. These results are comprehensively presented in Extended data Fig. 1, providing a thorough validation of our experimental strategy.

3- GLUT-1 and Collagen IV were used to label CNS vessels. However, the expression of GLUT-1 is highly sensitive to metabolic changes, and col. IV labels the vascular basement membrane (for instance, a vascular tract without endothelium may still appear as labeled by Col.IV). Hence, these are not the best options. To address this, authors could add immunostaining and new quantifications using a structural marker like PECAM-1 (and/or a fluorescent perfusion marker injected in the systemic circulation).

RESPONSE: While *Glut1* is indeed sensitive to metabolic changes, it is widely accepted and utilized as a vascular marker in numerous studies, such as (Armulik et al., 2010; Kim et al., 2017; Nikolakopoulou et al., 2021). To address the point raised by the reviewer we performed staining with *Pecam1* and *Glut1* in control and FLRT2-depleted vessels. As shown in Fig. 3 in this rebuttal letter, we did not observe any changes in the levels of expression of *Glut1* in our mutants compared to controls. Moreover, as the reviewer would appreciate, the staining of tip cell filopodia is only possible with *Glut1* and not with *Pecam1*. Therefore, it's important to note that *Glut1* serves a specific purpose in our study, since it allows to evaluate the formation of endothelial tip cells due to its localization to filopodial extensions. *Pecam1* it is only expressed at the interendothelial junctions and cannot identify the endothelial cell filopodia, which would have been a limitation in our study.

Regarding *Collagen IV*, our data in new Extended data Fig. 7e, f demonstrate that *Collagen IV* signal intensity does not differ between control and mutant vessels, and new panels in Extended data Fig. 2a, b confirm that the degree of vessel regression, assessed by *Collagen IV* empty sleeves, remains unchanged between genotypes. *Collagen IV* was chosen in specific instances in our study, such as Fig. 2c for its compatibility with expansion microscopy sample treatment compared to IB4, and in Fig. 7l to differentiate vessels from parenchyma in cadaverine leakage images.

4- *In vitro* experiments (PLA) were done using HUVECs. This raises concerns about the generalizability of mechanisms across species. To address this, authors could use primary mouse CNS endothelial cells (ECs) isolated and cultured. Many protocols have been published to perform such experiments. Authors could then compare primary mouse CNS ECs from their control and mutant mice for qualitative and quantitative assessments.

RESPONSE: We appreciate the suggestion to validate the PLA primary cells. In response to this request, we performed PLA on primary mouse brain endothelial cells directly isolated from control and FLRT2-deficient mutant mice. In cells derived from control animals, we observed a cluster signal between FLRT2 and VE-cadherin, validating our findings obtained in HUVECs (see Fig. 3a, b). In contrast, this interaction was significantly reduced in cells from *Flrt2*^{IAEC} animals, confirming the specificity of the signal (new panels in Extended Data Fig. 5b, c). In parallel, we performed immunoprecipitation on fresh whole brain cell lysates from wild-type animals using an anti-VE-cadherin antibody, ensuring the interactions occurred in the vasculature. Similar to the result we obtained in HUVECs, we have also immunoprecipitated FLRT2 in mouse tissue (see new panel in Fig. 3c). Note that we have also leverage the expansion microscopy technique and conducted immunostaining for VE-cadherin, FLRT2, and Numb directly in the tissue. This method allowed us to detect complexes containing these proteins at the cell membrane of the brain vasculature (see new panel in Fig. 6a) as well as the retina vasculature (see new panels in Fig. 6b, c). These combined results confirm our findings in HUVECS and demonstrate that the interaction between FLRT2 and VE-cadherin occurs *in vivo* and is preserved across species, highlighting the physiological relevance of our observations.

5- *While it may seem a minor point, the quality of the EM images in Figure 7k are below standards. Sample preparation is far from optimal and images not convincing. If authors wish to add EM to the study, they should revisit this part by performing new experiments with better tissue fixation and a deeper level of quantitative analysis.*

RESPONSE: The specific sample preparation for TEM in our study involved fixed thin brain sections (80 μ m) obtained from 2 mm diameter tissue punches that target areas identified by the cadaverine signal under confocal microscopy. This approach was necessary to precisely correlate the ultrastructural data to the vascular leakage. The technical challenge of working with such small tissue dimensions was carefully managed by expert neuropathologists specialized in TEM-based diagnostics, thus ensuring the quality of each sample analyzed. As

outlined in our methods section and depicted in the schematic representation (Extended data Fig. 8a), meticulous steps were taken to maintain tissue integrity and preserve ultrastructural details during preparation. To provide a clearer illustration of tissue quality, we have included a lower magnification view in Extended data Fig. 8a. Acknowledging the critique regarding image quality, we have replaced images with more representative examples (new panel in Fig. 7m). Furthermore, to enhance the comprehensiveness of our findings, additional representative images from both control and mutant animals have been included in Extended data Fig. 8b.

Regarding quantitative analysis, our study adopts a well-established method involving blind estimation of abnormal tight junctions, used by many other studies including recent ones (e.g., (Li et al., 2023)). In response to the suggestion of this reviewer for deeper analysis, we have meticulously refined our classification criteria to discern between various types of junction abnormalities, including open, undefined, tortuous configurations, and the presence of large vacuoles (>100 nm diameter) proximal to the junction. Such vacuoles are observed in conditions of vascular damage, such as those studied in ischemic stroke models (Nahirney et al., 2016). This comprehensive analysis uncovered a notable decrease in the prevalence of normal tight junctions following FLRT2 deletion. Specifically, approximately 30% of junctions were classified as open, and a significant proportion exhibited large vacuoles.

In conclusion, while we acknowledge the initial shortcomings in image quality, we believe these updates and refinements strengthen the robustness of our TEM data.

6- Here, the focus is on the endothelium, however it is well known that developing CNS vessels closely interact with glial cells, particularly in the retina where astrocytes form a carpet guiding vascular growth. Authors could investigate whether lack of FLRT2 in endothelial cells affects glial coverage for instance (using GFAP in retinas and ALDH1L1 in the cerebral cortex).

RESPONSE: In order to investigate whether the phenotypes observed correlate with defects in astrocytic coverage we assessed end-feet coverage through the expression of the water channel Aquaporin 4 (Aqp4). We found similar levels of Aqp4 coverage between control and *Flrt2*^{Δ^{EC}} mutant vessels, suggesting that the observed blood-brain barrier (BBB) defects are not attributable to alterations in astrocytic ensheathment around the vasculature (see results in new Extended data Fig. 7c, d). Importantly, it should be noted that at P7-8, the vasculature is not fully ensheathed by astrocytic end-feet, consistent with previous reports (Daneman et al., 2010).

Minor points:

1- Please define FLRT2 abbreviation when first mentioned in the Introduction.

RESPONSE: We added the description of the abbreviation of FLRT.

2- In the Results section (first paragraph) please justify the use of Podocalyxin in Figure 1. Is it considered a vein marker? Why other known vein markers were not used in this study?

RESPONSE: Podocalyxin is a protein of the glycocalyx and was utilized in our study as a general vascular marker to delineate the vessel lumen, given its widespread expression across both veins and arteries. To specifically investigate the arterio-venous expression pattern of FLRT2 in the cerebral cortex, we employed fluorescent in situ hybridization (FISH) with *Gkn3* as a marker for arteries and *Slc38a5* as a marker for veins. These markers were chosen based on their specificity for arterial and venous endothelial cells, respectively, as identified through scRNA-seq analysis of brain vasculature heterogeneity (Vanlandewijck et al., 2018). The corresponding probes for *Gkn3*, *Slc38a5*, and *Flrt2* were synthesized and used in conjunction with Podocalyxin immunostaining.

It's important to note that Podocalyxin staining is indeed present in both veins and arteries, as clarified now in the text and the figure legend of Extended data Fig. 2. Our newly added FISH results confirm that FLRT2 exhibits specific expression in venous segments of cerebral cortical vessels, while being absent in arteries (new panel in Extended Data Fig. 2e).

3- Differences measured in claudin-5 expression are small. Authors could measure protein levels and distribution of other EC junction for a more thorough description (e.g. ZO-1, occluding). A better description of both junction types (tight and adherent) would solidify the study.

RESPONSE: We acknowledge the reviewer's suggestion to provide a more thorough description of endothelial cell junctions. We have now quantified Claudin-5 protein levels by western blot in total brain lysates and confirmed the decreased expression of this tight junction protein (new Fig. 7f, g). Furthermore, to study the interplay between adherens and tight junctions, we examined Claudin-5 and VE-cadherin distribution by immunostaining the cerebral cortex vasculature of control and FLRT2 mutant mice (Fig. 7h, i). In control animals, Claudin-5 and VE-cadherin co-localize, outlining the cell membrane. However, in the absence of endothelial FLRT2, Claudin-5 often appears split into two distinct lanes flanking VE-

cadherin. We quantified this phenotype by measuring the proportion of split Claudin-5 per unit length of VE-cadherin, finding a significant increase in the mutants. This suggests that disrupted turnover of VE-cadherin at the membrane impairs the proper positioning of Claudin-5 and the sealing of the blood-brain barrier (BBB), as illustrated in Fig. 7.

Additionally, we investigated other tight junction components, such as junctional adhesion molecule-A (JAM-A) and zonula occludens-1 (ZO-1). We found no differences in their total expression levels in brain lysates or in their vascular expression assessed by immunohistochemistry (see Extended data Fig. 6c-h).

These findings suggest that the defects in tight junctions are specific to Claudin-5, which is known to be regulated by VE-cadherin and the associated signaling mechanisms involving β -catenin and FOXO1 nuclear translocation (Taddei et al., 2008). In agreement with this, FLRT2 deficiency also regulates β -catenin and FOXO1 nuclear expression (Fig. 7a-c).

Interestingly, BBB leakage associated with Claudin-5 deficiency is known to be size-selective for small molecules (Nitta et al., 2003; Vazquez-Liebanas et al., 2024). To assess if the Claudin-5 defects in FLRT2 vascular deletion were also size-selective, we injected a fluorescently labeled ovalbumin (45 kDa) BBB tracer. We found no defects in barrier permeability for this larger tracer (Extended Data Fig. 6i, j), confirming that the Claudin-5 specific defects are indeed selective for small molecules such as cadaverine (around 1 kDa).

REFERENCES

- Armulik, A., Genove, G., Mae, M., Nisancioglu, M.H., Wallgard, E., Niaudet, C., He, L., Norlin, J., Lindblom, P., Strittmatter, K., *et al.* (2010). Pericytes regulate the blood-brain barrier. *Nature* 468, 557-561.
- Barbacena, P., Dominguez-Cejudo, M., Fonseca, C.G., Gomez-Gonzalez, M., Faure, L.M., Zarkada, G., Pena, A., Pezzarossa, A., Ramalho, D., Giarratano, Y., *et al.* (2022). Competition for endothelial cell polarity drives vascular morphogenesis in the mouse retina. *Dev Cell* 57, 2321-2333 e2329.
- Bell, M.R. (2018). Comparing Postnatal Development of Gonadal Hormones and Associated Social Behaviors in Rats, Mice, and Humans. *Endocrinology* 159, 2596-2613.

- Benedito, R., Roca, C., Sorensen, I., Adams, S., Gossler, A., Fruttiger, M., and Adams, R.H. (2009). The notch ligands Dll4 and Jagged1 have opposing effects on angiogenesis. *Cell* *137*, 1124-1135.
- Boulday, G., Rudini, N., Maddaluno, L., Blecon, A., Arnould, M., Gaudric, A., Chapon, F., Adams, R.H., Dejana, E., and Tournier-Lasserre, E. (2011). Developmental timing of CCM2 loss influences cerebral cavernous malformations in mice. *J Exp Med* *208*, 1835-1847.
- Boye, K., Geraldo, L.H., Furtado, J., Pibouin-Fragner, L., Poulet, M., Kim, D., Nelson, B., Xu, Y., Jacob, L., Maissa, N., *et al.* (2022). Endothelial Unc5B controls blood-brain barrier integrity. *Nat Commun* *13*, 1169.
- Carmeliet, P., and Jain, R.K. (2011). Molecular mechanisms and clinical applications of angiogenesis. *Nature* *473*, 298-307.
- Coelho-Santos, V., Berthiaume, A.A., Ornelas, S., Stuhlmann, H., and Shih, A.Y. (2021). Imaging the construction of capillary networks in the neonatal mouse brain. *Proc Natl Acad Sci U S A* *118*.
- Coelho-Santos, V., and Shih, A.Y. (2020). Postnatal development of cerebrovascular structure and the neurogliovascular unit. *Wiley Interdiscip Rev Dev Biol* *9*, e363.
- Daneman, R., Zhou, L., Kebede, A.A., and Barres, B.A. (2010). Pericytes are required for blood-brain barrier integrity during embryogenesis. *Nature* *468*, 562-566.
- Eelen, G., Treps, L., Li, X., and Carmeliet, P. (2020). Basic and Therapeutic Aspects of Angiogenesis Updated. *Circ Res* *127*, 310-329.
- Fruttiger, M. (2002). Development of the mouse retinal vasculature: angiogenesis versus vasculogenesis. *Invest Ophthalmol Vis Sci* *43*, 522-527.
- Fujitani, K., Kambayashi, J., Sakon, M., Ohmi, S.I., Kawashima, S., Yukawa, M., Yano, Y., Miyoshi, H., Ikeda, M., Shinoki, N., *et al.* (1997). Identification of mu-, m-calpains and calpastatin and capture of mu-calpain activation in endothelial cells. *J Cell Biochem* *66*, 197-209.
- Gaengel, K., Niaudet, C., Hagikura, K., Lavina, B., Muhl, L., Hofmann, J.J., Ebarasi, L., Nystrom, S., Rymo, S., Chen, L.L., *et al.* (2012). The sphingosine-1-phosphate receptor S1PR1 restricts sprouting angiogenesis by regulating the interplay between VE-cadherin and VEGFR2. *Dev Cell* *23*, 587-599.
- Gavard, J., and Gutkind, J.S. (2008). VE-cadherin and claudin-5: it takes two to tango. *Nat Cell Biol* *10*, 883-885.

- Harb, R., Whiteus, C., Freitas, C., and Grutzendler, J. (2013). In vivo imaging of cerebral microvascular plasticity from birth to death. *J Cereb Blood Flow Metab* *33*, 146-156.
- Karaulanov, E., Bottcher, R.T., Stannek, P., Wu, W., Rau, M., Ogata, S., Cho, K.W., and Niehrs, C. (2009). Unc5B interacts with FLRT3 and Rnd1 to modulate cell adhesion in *Xenopus* embryos. *PLoS One* *4*, e5742.
- Kim, J., Kim, Y.H., Kim, J., Park, D.Y., Bae, H., Lee, D.H., Kim, K.H., Hong, S.P., Jang, S.P., Kubota, Y., *et al.* (2017). YAP/TAZ regulates sprouting angiogenesis and vascular barrier maturation. *J Clin Invest* *127*, 3441-3461.
- Konopka, G., Tekiela, J., Iverson, M., Wells, C., and Duncan, S.A. (2007). Junctional adhesion molecule-A is critical for the formation of pseudocanaliculi and modulates E-cadherin expression in hepatic cells. *J Biol Chem* *282*, 28137-28148.
- Lee, H.W., Xu, Y., He, L., Choi, W., Gonzalez, D., Jin, S.W., and Simons, M. (2021). Role of Venous Endothelial Cells in Developmental and Pathologic Angiogenesis. *Circulation* *144*, 1308-1322.
- Li, C., Chen, S., Siedhoff, H.R., Grant, D., Liu, P., Balderrama, A., Jackson, M., Zuckerman, A., Greenlief, C.M., Kobeissy, F., *et al.* (2023). Low-intensity open-field blast exposure effects on neurovascular unit ultrastructure in mice. *Acta Neuropathol Commun* *11*, 144.
- Li, J., Shinoda, Y., Ogawa, S., Ikegaya, S., Li, S., Matsuyama, Y., Sato, K., and Yamagishi, S. (2021). Expression of FLRT2 in Postnatal Central Nervous System Development and After Spinal Cord Injury. *Front Mol Neurosci* *14*, 756264.
- Miao, L., Li, J., Li, J., Lu, Y., Shieh, D., Mazurkiewicz, J.E., Barroso, M., Schwarz, J.J., Xin, H.B., Singer, H.A., *et al.* (2019). Cardiomyocyte orientation modulated by the Numb family proteins-N-cadherin axis is essential for ventricular wall morphogenesis. *Proc Natl Acad Sci U S A* *116*, 15560-15569.
- Nahirney, P.C., Reeson, P., and Brown, C.E. (2016). Ultrastructural analysis of blood-brain barrier breakdown in the peri-infarct zone in young adult and aged mice. *J Cereb Blood Flow Metab* *36*, 413-425.
- Nikolakopoulou, A.M., Wang, Y., Ma, Q., Sagare, A.P., Montagne, A., Huuskonen, M.T., Rege, S.V., Kisler, K., Dai, Z., Korbelin, J., *et al.* (2021). Endothelial LRP1 protects against neurodegeneration by blocking cyclophilin A. *J Exp Med* *218*.
- Nitta, T., Hata, M., Gotoh, S., Seo, Y., Sasaki, H., Hashimoto, N., Furuse, M., and Tsukita, S. (2003). Size-selective loosening of the blood-brain barrier in claudin-5-deficient mice. *J Cell Biol* *161*, 653-660.

- Noguera-Troise, I., Daly, C., Papadopoulos, N.J., Coetzee, S., Boland, P., Gale, N.W., Lin, H.C., Yancopoulos, G.D., and Thurston, G. (2006). Blockade of Dll4 inhibits tumour growth by promoting non-productive angiogenesis. *Nature* *444*, 1032-1037.
- Pitulescu, M.E., Schmidt, I., Benedito, R., and Adams, R.H. (2010). Inducible gene targeting in the neonatal vasculature and analysis of retinal angiogenesis in mice. *Nat Protoc* *5*, 1518-1534.
- Poschl, E., Schlotzer-Schrehardt, U., Brachvogel, B., Saito, K., Ninomiya, Y., and Mayer, U. (2004). Collagen IV is essential for basement membrane stability but dispensable for initiation of its assembly during early development. *Development* *131*, 1619-1628.
- Potente, M., Gerhardt, H., and Carmeliet, P. (2011). Basic and therapeutic aspects of angiogenesis. *Cell* *146*, 873-887.
- Prigge, C.L., Dembla, M., Sharma, A., El-Quessny, M., Kozlowski, C., Paisley, C.E., Miltner, A.M., Johnson, T.M., Della Santina, L., Feller, M.B., *et al.* (2023). Rejection of inappropriate synaptic partners in mouse retina mediated by transcellular FLRT2-UNC5 signaling. *Dev Cell* *58*, 2080-2096 e2087.
- Qiu, L.R., Fernandes, D.J., Szulc-Lerch, K.U., Dazai, J., Nieman, B.J., Turnbull, D.H., Foster, J.A., Palmert, M.R., and Lerch, J.P. (2018). Mouse MRI shows brain areas relatively larger in males emerge before those larger in females. *Nat Commun* *9*, 2615.
- Quaegebeur, A., Lange, C., and Carmeliet, P. (2011). The neurovascular link in health and disease: molecular mechanisms and therapeutic implications. *Neuron* *71*, 406-424.
- Rama, N., Dubrac, A., Mathivet, T., Ni Charthaigh, R.A., Genet, G., Cristofaro, B., Pibouin-Fragner, L., Ma, L., Eichmann, A., and Chedotal, A. (2015). Slit2 signaling through Robo1 and Robo2 is required for retinal neovascularization. *Nat Med* *21*, 483-491.
- Red-Horse, K., Ueno, H., Weissman, I.L., and Krasnow, M.A. (2010). Coronary arteries form by developmental reprogramming of venous cells. *Nature* *464*, 549-553.
- Ridgway, J., Zhang, G., Wu, Y., Stawicki, S., Liang, W.C., Chanthery, Y., Kowalski, J., Watts, R.J., Callahan, C., Kasman, I., *et al.* (2006). Inhibition of Dll4 signalling inhibits tumour growth by deregulating angiogenesis. *Nature* *444*, 1083-1087.
- Sabbagh, M.F., Heng, J.S., Luo, C., Castanon, R.G., Nery, J.R., Rattner, A., Goff, L.A., Ecker, J.R., and Nathans, J. (2018). Transcriptional and epigenomic landscapes of CNS and non-CNS vascular endothelial cells. *Elife* *7*.
- Sato, K., Watanabe, T., Wang, S., Kakeno, M., Matsuzawa, K., Matsui, T., Yokoi, K., Murase, K., Sugiyama, I., Ozawa, M., *et al.* (2011). Numb controls E-cadherin endocytosis through p120 catenin with aPKC. *Mol Biol Cell* *22*, 3103-3119.

- Segarra, M., Aburto, M.R., Cop, F., Llao-Cid, C., Hartl, R., Damm, M., Bethani, I., Parrilla, M., Husainie, D., Schanzer, A., *et al.* (2018). Endothelial Dab1 signaling orchestrates neuroglia-vessel communication in the central nervous system. *Science* *361*.
- Segarra, M., Williams, C.K., Sierra Mde, L., Bernardo, M., McCormick, P.J., Maric, D., Regino, C., Choyke, P., and Tosato, G. (2008). Dll4 activation of Notch signaling reduces tumor vascularity and inhibits tumor growth. *Blood* *112*, 1904-1911.
- Spurgin, S., and Cleaver, O. (2024). *Vascular Organization: Lessons from Development and Disease*. Cold Spring Harb Perspect Med.
- Stenmark, H. (2009). Rab GTPases as coordinators of vesicle traffic. *Nat Rev Mol Cell Biol* *10*, 513-525.
- Su, W., and Kowalczyk, A.P. (2017). The VE-cadherin cytoplasmic domain undergoes proteolytic processing during endocytosis. *Mol Biol Cell* *28*, 76-84.
- Taddei, A., Giampietro, C., Conti, A., Orsenigo, F., Breviario, F., Pirazzoli, V., Potente, M., Daly, C., Dimmeler, S., and Dejana, E. (2008). Endothelial adherens junctions control tight junctions by VE-cadherin-mediated upregulation of claudin-5. *Nat Cell Biol* *10*, 923-934.
- Trindade, A., Djokovic, D., Gigante, J., Mendonca, L., and Duarte, A. (2017). Endothelial Dll4 overexpression reduces vascular response and inhibits tumor growth and metastasization in vivo. *BMC Cancer* *17*, 189.
- Vanlandewijck, M., He, L., Mae, M.A., Andrae, J., Ando, K., Del Gaudio, F., Nahar, K., Lebouvier, T., Lavina, B., Gouveia, L., *et al.* (2018). A molecular atlas of cell types and zonation in the brain vasculature. *Nature* *554*, 475-480.
- Vazquez-Liebanas, E., Mocci, G., Li, W., Lavina, B., Reddy, A., O'Connor, C., Hudson, N., Elbeck, Z., Nikoloudis, I., Gaengel, K., *et al.* (2024). Mosaic deletion of claudin-5 reveals rapid non-cell-autonomous consequences of blood-brain barrier leakage. *Cell Rep* *43*, 113911.
- Visser, J.J., Cheng, Y., Perry, S.C., Chastain, A.B., Parsa, B., Masri, S.S., Ray, T.A., Kay, J.N., and Wojtowicz, W.M. (2015). An extracellular biochemical screen reveals that FLRTs and Unc5s mediate neuronal subtype recognition in the retina. *Elife* *4*, e08149.
- Walchli, T., Bisschop, J., Carmeliet, P., Zadeh, G., Monnier, P.P., De Bock, K., and Radovanovic, I. (2023). Shaping the brain vasculature in development and disease in the single-cell era. *Nat Rev Neurosci* *24*, 271-298.
- Wang, Y., Nakayama, M., Pitulescu, M.E., Schmidt, T.S., Bochenek, M.L., Sakakibara, A., Adams, S., Davy, A., Deutsch, U., Luthi, U., *et al.* (2010). Ephrin-B2 controls VEGF-induced angiogenesis and lymphangiogenesis. *Nature* *465*, 483-486.

- Xu, C., Hasan, S.S., Schmidt, I., Rocha, S.F., Pitulescu, M.E., Busmann, J., Meyen, D., Raz, E., Adams, R.H., and Siekmann, A.F. (2014). Arteries are formed by vein-derived endothelial tip cells. *Nat Commun* 5, 5758.
- Yamagishi, S., Hampel, F., Hata, K., Del Toro, D., Schwark, M., Kvachnina, E., Bastmeyer, M., Yamashita, T., Tarabykin, V., Klein, R., *et al.* (2011). FLRT2 and FLRT3 act as repulsive guidance cues for Unc5-positive neurons. *EMBO J* 30, 2920-2933.
- Yanagida, K., Liu, C.H., Faraco, G., Galvani, S., Smith, H.K., Burg, N., Anrather, J., Sanchez, T., Iadecola, C., and Hla, T. (2017). Size-selective opening of the blood-brain barrier by targeting endothelial sphingosine 1-phosphate receptor 1. *Proc Natl Acad Sci U S A* 114, 4531-4536.
- Zhang, Y., Chen, K., Sloan, S.A., Bennett, M.L., Scholze, A.R., O'Keeffe, S., Phatnani, H.P., Guarnieri, P., Caneda, C., Ruderisch, N., *et al.* (2014). An RNA-sequencing transcriptome and splicing database of glia, neurons, and vascular cells of the cerebral cortex. *J Neurosci* 34, 11929-11947.
- Zhang, Y., Liu, N.M., Wang, Y., Youn, J.Y., and Cai, H. (2017). Endothelial cell calpain as a critical modulator of angiogenesis. *Biochim Biophys Acta Mol Basis Dis* 1863, 1326-1335.

Rebuttal letter Fig. 1 Endothelial cell *Flrt2* mRNA levels during postnatal development.

RT-qPCR quantification of *Flrt2* mRNA levels obtained from primary mouse brain endothelial cells isolated from WT mice at P1, P5, P8 and P13-15. n = 3-4 animals per age.

Rebuttal letter Fig. 2 *Unc5b* and *Flrt2* are not co-expressed in the same neocortical vessels.

Fluorescent in situ hybridization in P8 wild-type mouse cerebral cortex (upper cortical layers and pial vasculature) detecting *Unc5b* and *Flrt2* mRNA expression. Blood vessels were labelled with Podocalyxin (Podxl) immunostaining. *Unc5b* positive ECs (arrowheads upper panels) are negative for *Flrt2* mRNA signal, while positive *Flrt2* ECs (arrowheads lower panels) are negative for *Unc5b*. Scale bar: 20 μ m.

Rebuttal letter Fig. 3 Comparison of Glut1 and Pecam1 as vascular markers.

Neocortical blood vessels from P7-8 *Flrt2*^{ΔEC} and control mice stained for Glut1 and Pecam1. Note the possible visualization of filopodia structures with the Glut1 staining in contrast to the Pecam1 junctional staining. No differences have been observed in Glut1 levels between control and *Flrt2*^{ΔEC} mice. Scale bar: 10 μm.

NCOMMS-23-62704A

Response point-by-point to reviewers' comments

Reviewer #1 (Remarks to the Author):

The authors have satisfactorily addressed all my major concerns regarding the manuscript. Indeed the revisions have significantly improved the clarity, methodology, and overall quality of the paper. Given these improvements, I recommend that the manuscript be accepted for publication.

Reviewer #2 (Remarks to the Author):

The authors addressed all the questions very well using complementary techniques. For example, to demonstrate the formation of the FLRT2, NumB, and VE-cadherin complex, they performed expansion microscopy and immunoprecipitation from brain lysates. The new data, after addressing all reviewers' concerns, is of high quality and confirms the overall conclusions of the manuscript.

Reviewer #3 (Remarks to the Author):

Overall, the authors have done a good job at addressing major concerns. The manuscript has been largely improved.

I continue to think that TEM images (Fig 7m & Extended Data Fig. 8b) of tight junctions (TJs) are below quality standards (i.e., imperfect fixation, low quality of ultrastructure, darkness, blurriness, etc). Authors may want to consult few published examples (e.g., PMID 30014540, 38635399, 38951020, 33681208, 37587100) to get a better idea of what would be expected as a quality standard for impactful publications. Moreover, when TJs display short protrusions towards the lumen this is actually not considered abnormal. I suggest authors only keep selected TEM images of higher quality and only with those TJs displaying obvious abnormal features (e.g. vacuoles).

RESPONSE: We would like to express our sincere gratitude to all three reviewers for their thoughtful feedback and constructive comments. We appreciate the recognition of the improvements made to our manuscript and are pleased that the revisions have significantly enhanced the quality of our work. In particular, in response to Reviewer #3's observations, we

have carefully revised Figure 7 and Extended Figure 8. As recommended, we have selectively included only those TEM images that clearly exhibit the most obvious abnormal features.